# Risk Bounds for Mixture Density Estimation on Compact Domains via the $h$-Lifted Kullback–Leibler Divergence

**Mark Chiu Chong**
*School of Mathematics and Physics*
*The University of Queensland*
*St Lucia, QLD 4072, Australia*

*mark.chiuchong@gmail.com*

**Hien Duy Nguyen**
*Institute of Mathematics for Industry*
*Kyushu University*
*Nishi Ward, Fukuoka 819-0395, Japan;*

*hien@imi.kyushu-u.ac.jp*

*School of Computing, Engineering, and Mathematical Sciences*
*La Trobe University*
*Bundoora, VIC 3086, Australia;*

**TrungTin Nguyen**
*School of Mathematics and Physics*
*The University of Queensland*
*St Lucia, QLD 4072, Australia*

*trungtin.nguyen@uq.edu.au*

**Reviewed on OpenReview:** *https: // openreview. net/ forum? id= lAKvQO4vHj*

## Abstract

We consider the problem of estimating probability density functions based on sample data, using a finite mixture of densities from some component class. To this end, we introduce the $h$-lifted Kullback–Leibler (KL) divergence as a generalization of the standard KL divergence and a criterion for conducting risk minimization. Under a compact support assumption, we prove an $\mathcal{O}(1/\sqrt{n})$ bound on the expected estimation error when using the $h$-lifted KL divergence, which extends the results of Rakhlin et al. (2005, ESAIM: Probability and Statistics, Vol. 9) and Li & Barron (1999, Advances in Neural Information Processing Systems, Vol. 12) to permit the risk bounding of density functions that are not strictly positive. We develop a procedure for the computation of the corresponding maximum $h$-lifted likelihood estimators ($h$-MLLEs) using the Majorization-Maximization framework and provide experimental results in support of our theoretical bounds.

## 1 Introduction

Let $(\Omega, \mathfrak{A}, \mathbf{P})$ be an abstract probability space and let $X : \Omega \to \mathcal{X}$ be a random variable taking values in the measurable space $(\mathcal{X}, \mathfrak{F})$, where $\mathcal{X}$ is a compact metric space equipped with its Borel $\sigma$-algebra $\mathfrak{F}$. Suppose that we observe an independent and identically distributed (i.i.d.) sample of random variables $\mathbf{X}_n = (X_i)_{i \in [n]}$, where $[n] = \{1, \ldots, n\}$, and that each $X_i$ arises from the same data generating process as $X$, characterized by the probability measure $F \ll \mu$ on $(\mathcal{X}, \mathfrak{F})$, with density function $f = \mathrm{d}F/\mathrm{d}\mu$, for some $\sigma$-finite $\mu$. In this work, we are concerned with the estimating $f$ via a data dependent double-index sequence of estimators $(f_{k,n})_{k,n \in \mathbb{N}}$, where

$$f_{k,n} \in \mathcal{C}_k = \mathrm{co}_k(\mathcal{P}) = \left\{ f_k(\cdot; \psi_k) = \sum_{j=1}^{k} \pi_j \varphi(\cdot; \theta_j) \mid \varphi(\cdot; \theta_j) \in \mathcal{P}, \ \pi_j \geq 0, \ j \in [k], \sum_{j=1}^{k} \pi_j = 1 \right\},$$

for each $k, n \in \mathbb{N}$, and where

$$\mathcal{P} = \left\{ \varphi\left(\cdot; \theta\right) : \mathcal{X} \to \mathbb{R}_{\geq 0} \mid \theta \in \Theta \subset \mathbb{R}^d \right\}, \tag{1}$$

$\psi_k = (\pi_1, \ldots, \pi_k, \theta_1, \ldots, \theta_k)$, and $d \in \mathbb{N}$. To ensure the measurability and existence of various optima, we shall assume that $\varphi$ is *Carathéodory* in the sense that $\varphi\left(\cdot; \theta\right)$ is $(\mathcal{X}, \mathfrak{F})$-measurable for each $\theta \in \Theta$, and $\varphi\left(X; \cdot\right)$ is continuous for each $X \in \mathcal{X}$.

In the definition above, we can identify the set $\mathcal{C}_k = \mathrm{co}_k\left(\mathcal{P}\right)$ as the set of density functions that can be written as a convex combination of $k$ elements of $\mathcal{P}$, where $\mathcal{P}$ is often called the space of component density functions. We then interpret $\mathcal{C}_k$ as the class of $k$-component finite mixtures of densities of class $\mathcal{P}$, as studied, for example, by McLachlan & Peel (2004); Nguyen et al. (2020; 2022b).

## 1.1 Risk bounds for mixture density estimation

We are particularly interested in oracle bounds of the form

$$\mathbf{E}\left\{\ell\left(f, f_{k,n}\right)\right\} - \ell\left(f, \mathcal{C}\right) \leq \rho\left(k, n\right), \tag{2}$$

where $(p, q) \mapsto \ell\left(p, q\right) \in \mathbb{R}_{\geq 0}$ is a loss function on pairs of density functions. We define the density-to-class loss

$$\ell\left(f, \mathcal{C}\right) = \inf_{q \in \mathcal{C}} \ell\left(f, q\right), \quad \mathcal{C} = \mathrm{cl}\left(\bigcup_{k \in \mathbb{N}} \mathrm{co}_k\left(\mathcal{P}\right)\right),$$

where $\mathrm{cl}(\cdot)$ is the closure. Here, we identify $(k, n) \mapsto \rho\left(k, n\right)$ as a characterization of the rate at which the left-hand side of (2) converges to zero as $k$ and $n$ increase. Our present work follows the research of Li & Barron (1999), Rakhlin et al. (2005) and Klemelä (2007) (see also Klemelä 2009, Ch. 19). In Li & Barron (1999) and Rakhlin et al. (2005), the authors consider the case where $\ell\left(p, q\right)$ is taken to be the Kullback–Leibler (KL) divergence

$$\mathrm{KL}\left(p \,\|\, q\right) = \int p \log \frac{p}{q} \mathrm{d}\mu$$

and $f_{k,n} = f_k\left(\cdot; \psi_{k,n}\right)$ is a maximum likelihood estimator (MLE), where

$$\psi_{k,n} \in \underset{\psi_k \in \mathcal{S}_k \times \Theta^k}{\arg\max} \frac{1}{n} \sum_{i=1}^{n} \log f_k\left(X_i; \psi_k\right),$$

is a function of $\mathbf{X}_n$, with $\mathcal{S}_k$ denoting the probability simplex in $\mathbb{R}^k$.

Under the assumption that $f, f_k \geq a$, for some $a > 0$ and each $k \in [n]$ (i.e., strict positivity), Li & Barron (1999) obtained the bound

$$\mathbf{E}\left\{\mathrm{KL}\left(f \,\|\, f_{k,n}\right)\right\} - \mathrm{KL}\left(f \,\|\, \mathcal{C}\right) \leq c_1 \frac{1}{k} + c_2 \frac{k \log\left(c_3 n\right)}{n},$$

for constants $c_1, c_2, c_3 > 0$, which was then improved by Rakhlin et al. (2005) who obtained the bound

$$\mathbf{E}\left\{\mathrm{KL}\left(f \,\|\, f_{k,n}\right)\right\} - \mathrm{KL}\left(f \,\|\, \mathcal{C}\right) \leq c_1 \frac{1}{k} + c_2 \frac{1}{\sqrt{n}},$$

for constants $c_1, c_2 > 0$ (constants $(c_j)_{j \in \mathbb{N}}$ are typically different between expressions).

Alternatively, Klemelä (2007) takes $\ell\left(p, q\right)$ to be the squared $L_2\left(\mu\right)$ norm distance (i.e., the least-squares loss):

$$\ell\left(p, q\right) = \|p - q\|_{2,\mu}^2,$$

where $\|p\|_{2,\mu}^2 = \int_{\mathcal{X}} |p|^2 \mathrm{d}\mu$, for each $p \in L_2\left(\mu\right)$, and choose $f_{k,n}$ as minimizers of the $L_2\left(\mu\right)$ empirical risk, i.e., $f_{k,n} = f_k\left(\cdot; \psi_{k,n}\right)$, where

$$\psi_{k,n} \in \underset{\psi_k \in \mathcal{S}_k \times \Theta^k}{\arg\min} -\frac{2}{n} \sum_{i=1}^{n} f_k\left(\cdot; \psi_k\right) + \|f_k\left(\cdot; \psi_k\right)\|_{2,\mu}^2. \tag{3}$$

Here, Klemelä (2007) establish the bound

$$\mathbf{E}\left\|f - f_{k,n}\right\|_{2,\mu}^2 - \inf_{q \in \mathcal{C}} \left\|f - q\right\|_{2,\mu}^2 \leq c_1 \frac{1}{k} + c_2 \frac{1}{\sqrt{n}},$$

$c_1, c_2 > 0$, without the lower bound assumption on $f, f_k$ above, even permitting $\mathcal{X}$ to be unbounded. Via the main results of Naito & Eguchi (2013), the bound above can be generalized to the $U$-divergences, which includes the special $L_2(\mu)$ norm distance as a special case.

On the one hand, the sequence of MLEs required for the results of Li & Barron (1999) and Rakhlin et al. (2005) are typically computable, for example, via the usual expectation–maximization approach (cf. McLachlan & Peel 2004, Ch. 2). This contrasts with the computation of least-squares density estimators of form (3), which requires evaluations of the typically intractable integral expressions $\left\|f_k\left(\cdot; \psi_k\right)\right\|_2^2$. However, the least-squares approach of Klemelä (2007) permits the analysis using families $\mathcal{P}$ of usual interest, such as normal distributions and beta distributions, the latter of which being compactly supported but having densities that cannot be bounded away from zero without restrictions, and thus do not satisfy the regularity conditions of Li & Barron (1999) and Rakhlin et al. (2005).

## 1.2 Main contributions

We propose the following $h$-lifted KL divergence, as a generalization of the standard KL divergence to address the computationally tractable estimation of density functions which do not satisfy the regularity conditions of Li & Barron (1999) and Rakhlin et al. (2005). The use of the $h$-lifted KL divergence has the possibility to advance theories based on the standard KL divergence in statistical machine learning. To this end, let $h : \mathcal{X} \to \mathbb{R}_{\geq 0}$ be a function in $L_1(\mu)$, and define the $h$-lifted KL divergence by:

$$\mathrm{KL}_h\left(p \,\|\, q\right) = \int_{\mathcal{X}} \{p + h\} \log \frac{p + h}{q + h} \mathrm{d}\mu. \tag{4}$$

In the sequel, we shall show that $\mathrm{KL}_h$ is a Bregman divergence on the space of probability density functions, as per Csiszár (1995).

Assume that $h$ is a probability density function, and let $\mathbf{Y}_n = (Y_i)_{i \in [n]}$ be a an i.i.d. sample, independent of $\mathbf{X}_n$, where each $Y_i : \Omega \to \mathcal{X}$ is a random variable with probability measure on $(\mathcal{X}, \mathfrak{F})$, characterized by the density $h$ with respect to $\mu$. Then, for each $k$ and $n$, let $f_{k,n}$ be defined via the maximum $h$-lifted likelihood estimator ($h$-MLLE; see Appendix B for further discussion) $f_{k,n} = f_k\left(\cdot; \psi_{k,n}\right)$, where

$$\psi_{k,n} \in \operatorname*{arg\,max}_{\psi_k \in \mathcal{S}_k \times \Theta^k} \frac{1}{n} \sum_{i=1}^n \left(\log\left\{f_k\left(X_i; \psi_k\right) + h\left(X_i\right)\right\} + \log\left\{f_k\left(Y_i; \psi_k\right) + h\left(Y_i\right)\right\}\right). \tag{5}$$

The primary aim of this work is to show that

$$\mathbf{E}\left\{\mathrm{KL}_h\left(f \,\|\, f_{k,n}\right)\right\} - \mathrm{KL}_h\left(f \,\|\, \mathcal{C}\right) \leq c_1 \frac{1}{k} + c_2 \frac{1}{\sqrt{n}} \tag{6}$$

for some constants $c_1, c_2 > 0$, without requiring the strict positivity assumption that $f, f_k \geq a > 0$.

This result is a compromise between the works of Li & Barron (1999) and Rakhlin et al. (2005), and Klemelä (2007), as it applies to a broader space of component densities $\mathcal{P}$, and because the required $h$-MLLEs (5) can be efficiently computed via minorization–maximization (MM) algorithms (see e.g., Lange 2016). We shall discuss this assertion in Section 4.

## 1.3 Relevant literature

Our work largely follows the approach of Li & Barron (1999), which was extended upon by Rakhlin et al. (2005) and Klemelä (2007). All three texts use approaches based on the availability of greedy algorithms for maximizing convex functions with convex functional domains. In this work, we shall make use of the

proof techniques of Zhang (2003). Related results in this direction can be found in DeVore & Temlyakov (2016) and Temlyakov (2016). Making the same boundedness assumption as Rakhlin et al. (2005), Dalalyan & Sebbar (2018) obtain refined oracle inequalities under the additional assumption that the class $\mathcal{P}$ is finite. Numerical implementations of greedy algorithms for estimating finite mixtures of Gaussian densities were studied by Vlassis & Likas (2002) and Verbeek et al. (2003).

The $h$-MLLE as an optimization objective can be compared to other similar modified likelihood estimators, such as the $L_q$ likelihood of Ferrari & Yang (2010) and Qin & Priebe (2013), the $\beta$-likelihood of Basu et al. (1998) and Fujisawa & Eguchi (2006), penalized likelihood estimators, such as maximum a posteriori estimators of Bayesian models, or $f$-separable Bregman distortion measures of Kobayashi & Watanabe (2024; 2021).

The practical computation of the $h$-MLLEs, (5), is made possible via the MM algorithm framework of Lange (2016), see also Hunter & Lange (2004), Wu & Lange (2010), and Nguyen (2017) for further details. Such algorithms have well-studied global convergence properties and can be modified for mini-batch and stochastic settings (see, e.g., Razaviyayn et al., 2013 and Nguyen et al., 2022a).

A related and popular setting of investigations is that of model selection, where the objects of interest are single-index sequences $(f_{k_n,n})_{n \in \mathbb{N}}$, and where the aim is to obtain finite-sample bounds for losses of the form $\ell(f_{k_n,n}, f)$, where each $k_n \in \mathbb{N}$ is a data dependent function, often obtained by optimizing some penalized loss criterion, as described in Massart (2007), Koltchinskii (2011, Ch. 6), and Giraud (2021, Ch. 2). In the context of finite mixtures, examples of such analyses can be found in the works of Maugis & Michel (2011) and Maugis-Rabusseau & Michel (2013). A comprehensive bibliography of model selection results for finite mixtures and related statistical models can be found in Nguyen et al. (2022c).

### 1.4 Organization of paper

The manuscript is organized as follows. In Section 2, we formally define the $h$-lifted KL divergence as a Bregman divergence and establish several of its properties. In Section 3, we present new risk bounds for the $h$-lifted KL divergence of the form (2). In Section 4, we discuss the computation of the $h$-lifted likelihood estimator in the form of (5), followed by empirical results illustrating the convergence of (2) with respect to both $k$ and $n$. Additional insights and technical results are provided in the Appendices at the end of the manuscript.

## 2 The $h$-lifted KL divergence and its properties

In this section we formally define the $h$-lifted KL divergence on the space of density functions and establish some of its properties.

**Definition 1** ($h$-lifted KL divergence). *Let $f, g,$ and $h$ be probability density functions on the space $\mathcal{X}$, where $h > 0$. The $h$-lifted* KL *divergence from $g$ to $f$ is defined as follows:*

$$\mathrm{KL}_h(f \,||\, g) = \int_{\mathcal{X}} \{f + h\} \log \frac{f+h}{g+h} \mathrm{d}\mu = \mathbf{E}_f \left\{ \log \frac{f+h}{g+h} \right\} + \mathbf{E}_h \left\{ \log \frac{f+h}{g+h} \right\}.$$

### 2.1 $\mathrm{KL}_h$ as a Bregman divergence

Let $\phi : \mathcal{I} \to \mathbb{R}$, $\mathcal{I} = (0, \infty)$ be a strictly convex function that is continuously differentiable. The Bregman divergence between scalars $d_\phi : \mathcal{I} \times \mathcal{I} \to \mathbb{R}_{\geq 0}$ generated by the function $\phi$ is given by:

$$d_\phi(p, q) = \phi(p) - \phi(q) - \phi'(q)(p - q),$$

where $\phi'(q)$ denotes the derivative of $\phi$ at $q$.

Bregman divergences possess several useful properties, including the following list:

1. Non-negativity: $d_\phi(p, q) \geq 0$ for all $p, q \in \mathcal{I}$ with equality if and only if $p = q$;

2. Asymmetry: $d_\phi(p, q) \neq d_\phi(q, p)$ in general;

3. Convexity: $d_\phi(p, q)$ is convex in $p$ for every fixed $q \in \mathcal{I}$.

4. Linearity: $d_{c_1\phi_1 + c_2\phi_2}(p, q) = c_1 d_{\phi_1}(p, q) + c_2 d_{\phi_2}(p, q)$ for $c_1, c_2 \geq 0$.

The properties for Bregman divergences between scalars can be extended to density functions and other functional spaces, as established in Frigyik et al. (2008) and Stummer & Vajda (2012), for example. We also direct the interested reader to the works of Pardo (2006), Basu et al. (2011), and Amari (2016).

The class of $h$-lifted KL divergences constitute a generalization of the usual KL divergence and are a subset of the Bregman divergences over the space of density functions that are considered by Csiszár (1995). Namely, let $\mathcal{P}$ be a convex set of probability densities with respect to the measure $\mu$ on $\mathcal{X}$. The Bregman divergence $D_\phi : \mathcal{P} \times \mathcal{P} \to [0, \infty)$ between densities $p, q \in \mathcal{P}$ can be constructed as follows:

$$D_\phi(p \,||\, q) = \int_{\mathcal{X}} d_\phi\left(p(x), q(x)\right) \mathrm{d}\mu(x).$$

The $h$-lifted KL divergence $\mathrm{KL}_h$ as a Bregman divergence is generated by the function $\phi(u) = (u+h)\log(u+h) - (u+h) + 1$. This assertion is demonstrated in Appendix C.1.

## 2.2 Advantages of the $h$-lifted KL divergence

When the standard KL divergence is employed in the density estimation problem, it is common to restrict consideration of density functions to those bounded away from zero by some positive constant. That is, one typically considers the smaller class of so-called *admissible* target densities $\mathcal{P}_\alpha \subset \mathcal{P}$ (cf. Meir & Zeevi, 1997), where

$$\mathcal{P}_\alpha = \{\varphi(\cdot; \theta) \in \mathcal{P} \mid \varphi(\cdot; \theta) \geq \alpha > 0\}.$$

Without this restriction, the standard KL divergence can be unbounded, even for functions with bounded $L_1$ norms. For example, let $p$ and $q$ be densities of beta distributions on the support $\mathcal{X} = [0, 1]$. That is, suppose that $p, q \in \mathcal{P}_{\mathrm{beta}}$, respectively characterized by parameters $\theta_p = (a_p, b_p)$ and $\theta_q = (a_q, b_q)$, where

$$\mathcal{P}_{\mathrm{beta}} = \left\{x \mapsto \beta\left(x; \theta\right) = \frac{\Gamma\left(a + b\right)}{\Gamma\left(a\right)\Gamma\left(b\right)} x^{a-1}\left(1 - x\right)^{b-1}, \theta = (a, b) \in \mathbb{R}_{>0}^2\right\}. \tag{7}$$

Then, from Gil et al. (2013), the KL divergence between $p$ and $q$ is given by:

$$\mathrm{KL}\left(p \,||\, q\right) = \log\left\{\frac{\Gamma\left(a_q\right)\Gamma\left(b_q\right)}{\Gamma\left(a_q + b_q\right)}\right\} - \log\left\{\frac{\Gamma\left(a_p\right)\Gamma\left(b_p\right)}{\Gamma\left(a_p + b_p\right)}\right\}$$
$$+ (a_p - a_q)\left\{\psi\left(a_p\right) - \psi\left(a_p + b_p\right)\right\} + (b_p - b_q)\left\{\psi\left(b_p\right) - \psi\left(a_p + b_p\right)\right\},$$

where $\psi : \mathbb{R}_{>0} \to \mathbb{R}$ is the digamma function. Next, suppose that $a_p = b_q$ and $a_q = b_p = 1$, which leads to the simplification

$$\mathrm{KL}\left(p \,||\, q\right) = (a_p - 1)\left\{\psi\left(a_p\right) - \psi(1)\right\}.$$

Since $\psi$ is strictly increasing, we observe that the right-hand side diverges as $a_p \to \infty$. Thus, the KL divergence between beta distributions is unbounded. The $h$-lifted KL divergence in contrast does not suffer from this problem, and does not require the restriction to $\mathcal{P}_\alpha$. This allows us to consider cases where $p, q \in \mathcal{P}$ are not bounded away from 0, as per the following result.

**Proposition 2.** *Let $\mathcal{P}$ be defined as in (1). $\mathrm{KL}_h\left(f \,||\, g\right)$ is bounded for all continuous densities $f, g \in \mathcal{P}$.*

*Proof.* See Appendix C.2. $\qquad\qquad\square$

Let $L_p(f, g)$ denote the standard $L_p$-norm, $L_p(f, g) = \left\{\int_{\mathcal{X}} |f(x) - g(x)|^p \mathrm{d}\mu(x)\right\}^{1/p}$. As remarked previously, Klemelä (2007) established empirical risk bounds in terms of the $L_2$-norm distance. Following results from Meir & Zeevi (1997) characterizing the relationship between the KL divergence in terms of the $L_2$-norm

distance, in Proposition 3 we establish the corresponding relationship between the $h$-lifted KL divergence and the $L_2$-norm distance, along with a relationship between the $h$-lifted KL divergence and the $L_1$-norm distance.

**Proposition 3.** *For probability density functions $f$, $g$, and $h$, where $h$ is such that $h(x) \geq \gamma > 0$ for all $x \in \mathcal{X}$, the following inequalities hold:*

$$\frac{1}{4} L_1^2 (f, g) \leq \mathrm{KL}_h (f \,\|\, g) \leq \gamma^{-1} L_2^2 (f, g).$$

*Proof.* See Appendix C.3. $\qquad\square$

**Remark 4.** *Proposition 2 highlights the benefit of the h-lifted KL divergence being bounded for all continuous densities, unlike the standard KL divergence, while satisfying a relationship similar to that between the KL divergence and the $L_2$ norm distance. Moreover, the first inequality of Proposition 3 is a Pinsker-like relationship between the h-lifted KL divergence and the total variation distance $\mathrm{TV}(f, g) = \frac{1}{2} L_1(f, g)$.*

## 3 Main results

Here we provide explicit statements regarding the convergence rates claimed in (6) via Theorem 5 and Corollary 6, which are proved in Appendix A.2. We assume that $f$ is bounded above by some constant $c$ and that the lifting function $h$ is bounded above and below by constants $a$ and $b$, respectively.

**Theorem 5.** *Let $h$ be a positive density satisfying $0 < a \leq h(x) \leq b$, for all $x \in \mathcal{X}$. For any target density $f$ satisfying $0 \leq f(x) \leq c$, for all $x \in \mathcal{X}$ and where $f_{k,n}$ is the minimizer of $\mathrm{KL}_h$ over $k$-component mixtures, the following inequality holds:*

$$\mathbf{E} \{\mathrm{KL}_h (f \,\|\, f_{k,n})\} - \mathrm{KL}_h (f \,\|\, \mathcal{C}) \leq \frac{u_1}{k+2} + \frac{u_2}{\sqrt{n}} \int_0^c \log^{1/2} N(\mathcal{P}, \varepsilon/2, \|\cdot\|_\infty) \mathrm{d}\varepsilon + \frac{u_3}{\sqrt{n}},$$

*where $u_1$, $u_2$, and $u_3$ are positive constants that depend on some or all of $a$, $b$, and $c$.*

**Corollary 6.** *Let $\mathcal{X}$ and $\Theta$ be compact and assume the following Lipschitz condition holds: for each $x \in \mathcal{X}$, and for each $\theta, \tau \in \Theta$,*

$$|\varphi(x; \theta) - \varphi(x; \tau)| \leq \Phi(x) \|\theta - \tau\|_1, \tag{8}$$

*for some function $\Phi : \mathcal{X} \to \mathbb{R}_{\geq 0}$, where $\|\Phi\|_\infty = \sup_{x \in \mathcal{X}} |\Phi(x)| < \infty$. Then the bound in Theorem 5 becomes*

$$\mathbf{E} \{\mathrm{KL}_h (f \,\|\, f_{k,n})\} - \mathrm{KL}_h (f \,\|\, \mathcal{C}) \leq \frac{c_1}{k+2} + \frac{c_2}{\sqrt{n}},$$

*where $c_1$ and $c_2$ are positive constants.*

**Remark 7.** *Our results are applicable to any compact metric space $\mathcal{X}$, with $[0, 1]$ used in the experimental setup in Section 4.2 as a simple and tractable example to illustrate key aspects of our theory. There is no issue in generalizing to $\mathcal{X} = [-m, m]^d$ for $m > 0$ and $d \in \mathbb{N}$, or more abstractly, to any compact subset $\mathcal{X} \subset \mathbb{R}^d$. Additionally, $\mathcal{X}$ could even be taken as a functional compact space, though establishing compactness and constructing appropriate component classes $\mathcal{P}$ over such spaces to achieve small approximation errors $\mathrm{KL}_h (f \,\|\, \mathcal{C})$ is an approximation theoretic task that falls outside the scope of our work.*

*From the proof of Theorem 5 in Appendix A.2, it is clear that the dimensionality of $\mathcal{X}$ only influences our bound through the complexity of the class $\mathcal{P}$, specifically, the constant $\int_0^c \log^{1/2} N(\mathcal{P}, \varepsilon/2, \|\cdot\|_\infty) \mathrm{d}\varepsilon$, which remains independent of both $n$ and $k$. Here, $N(\mathcal{P}, \varepsilon, \|\cdot\|)$ is the $\varepsilon$-covering number of $\mathcal{P}$. In fact, the constant with respect to $k$ ($u_1$ in Theorem 5) is entirely unaffected by the dimensionality of $\mathcal{X}$. Thus, the rates of our bound on the expected h-lifted KL divergence are dimension-independent and hold even when $\mathcal{X}$ is infinite-dimensional, as long as there exists a class $\mathcal{P}$ such that $\int_0^c \log^{1/2} N(\mathcal{P}, \varepsilon/2, \|\cdot\|_\infty) \mathrm{d}\varepsilon$ is finite.*

*Corollary 6 provides a method for obtaining such a bound when the elements of $\mathcal{P}$ satisfy a Lipschitz condition.*

# 4 Numerical experiments

Here we discuss the computability and computation of $\mathrm{KL}_h$ estimation problems and provide empirical evidence towards the rates obtained in Theorem 5. Specifically, we seek to develop a methodology for computing $h$-MLLEs, and to use numerical experiments to demonstrate that the sequence of expected $h$-lifted KL divergences between some density $f$ and a sequence of $k$-component mixture densities from a suitable class $\mathcal{P}$, estimated using $n$ observations does indeed decrease at rates proportional to $1/k$ and $1/\sqrt{n}$, as $k$ and $n$ increase.

The code for all simulations and analyses in Experiments 1 and 2 is available in both the `R` and `Python` programming languages. The code repository is available here: `https://github.com/hiendn/LiftedLikelihood`.

## 4.1 Minorization–Maximization algorithm

One solution for computing (5) is to employ an MM algorithm. To do so, we first write the objective of (5) as

$$L_{h,n}\left(\psi_k\right) = \frac{1}{n}\sum_{i=1}^{n}\left(\log\left\{\sum_{j=1}^{k}\pi_j\varphi\left(X_i;\theta_j\right) + h\left(X_i\right)\right\} + \log\left\{\sum_{j=1}^{k}\pi_j\varphi\left(Y_i;\theta_j\right) + h\left(Y_i\right)\right\}\right),$$

where $\psi_k \in \Psi_k = \mathcal{S}_k \times \Theta^k$. We then require the definition of a minorizer $Q_n$ for $L_{h,n}$ on the space $\Psi_k$, where $Q_n : \Psi_k \times \Psi_k \to \mathbb{R}$ is a function with the properties:

(i) $Q_n\left(\psi_k, \psi_k\right) = L_{h,n}\left(\psi_k\right)$, and

(ii) $Q_n\left(\psi_k, \chi_k\right) \le L_{h,n}\left(\psi_k\right)$,

for each $\psi_k, \chi_k \in \Psi_k$. In this context, given a fixed $\chi_k$, the minorizer $Q_n\left(\cdot, \chi_k\right)$ should possess properties that simplify it compared to the original objective $L_{h,n}$. These properties should make the minorizer more tractable and might include features such as parametric separability, differentiability, convexity, among others.

In order to build an appropriate minorizer for $L_{h,n}$, we make use of the so-called Jensen's inequality minorizer, as detailed in Lange (2016, Sec. 4.3), applied to the logarithm function. This construction results in a minorizer of the form

$$
\begin{aligned}
Q_n\left(\psi_k, \chi_k\right) &= \frac{1}{n}\sum_{i=1}^{n}\sum_{j=1}^{k}\left\{\tau_j\left(X_i;\chi_k\right)\log\pi_j + \tau_j\left(X_i;\chi_k\right)\log\varphi\left(X_i;\theta_j\right)\right\} \\
&+ \frac{1}{n}\sum_{i=1}^{n}\sum_{j=1}^{k}\left\{\tau_j\left(Y_i;\chi_k\right)\log\pi_j + \tau_j\left(Y_i;\chi_k\right)\log\varphi\left(Y_i;\theta_j\right)\right\} \\
&+ \frac{1}{n}\sum_{i=1}^{n}\left\{\gamma\left(X_i;\chi_k\right)\log h\left(X_i\right) + \gamma\left(Y_i;\chi_k\right)\log h\left(Y_i\right)\right\} \\
&- \frac{1}{n}\sum_{i=1}^{n}\sum_{j=1}^{k}\left\{\tau_j\left(X_i;\chi_k\right)\log\tau_j\left(X_i;\chi_k\right) + \tau_j\left(Y_i;\chi_k\right)\log\tau_j\left(Y_i;\chi_k\right)\right\} \\
&- \frac{1}{n}\sum_{i=1}^{n}\left\{\gamma\left(X_i;\chi_k\right)\log\gamma\left(X_i;\chi_k\right) + \gamma\left(Y_i;\chi_k\right)\log\gamma\left(Y_i;\chi_k\right)\right\}
\end{aligned}
$$

where

$$\gamma\left(X_i;\psi_k\right) = h\left(X_i\right) \Big/ \left\{\sum_{j=1}^{k}\pi_j\varphi\left(X_i;\theta_j\right) + h\left(X_i\right)\right\}$$

and

$$\tau_j\left(X_i; \psi_k\right) = \pi_j \varphi\left(X_i; \theta_j\right) / \left\{\sum_{j=1}^{k} \pi_j \varphi\left(X_i; \theta_j\right) + h\left(X_i\right)\right\}.$$

Observe that $Q_n\left(\cdot, \chi_k\right)$ now takes the form of a sum-of-logarithms, as opposed to the more challenging log-of-sum form of $L_{h,n}$. This change produces a functional separation of the elements of $\psi_k$.

Using $Q_n$, we then define the MM algorithm via the parameter sequence $\left(\psi_k^{(s)}\right)_{s\in\mathbb{N}}$, where

$$\psi_k^{(s)} = \underset{\psi_k \in \Psi_k}{\arg\max}\ Q_n\left(\psi_k, \psi_k^{(s-1)}\right), \tag{9}$$

for each $s > 0$, and where $\psi_k^{(0)}$ is user chosen and is typically referred to as the initialization of the algorithm. Notice that for each $s$, (9) is a simpler optimization problem than (5). Writing $\psi_k^{(s)} = \left(\pi_1^{(s)}, \ldots, \pi_k^{(s)}, \theta_1^{(s)}, \ldots, \theta_k^{(s)}\right)$, we observe that (9) simplifies to the separated expressions:

$$\pi_j^{(s)} = \frac{\sum_{i=1}^{n}\left\{\tau_j\left(X_i; \psi_k^{(s-1)}\right) + \tau_j\left(Y_i; \psi_k^{(s-1)}\right)\right\}}{\sum_{i=1}^{n}\sum_{l=1}^{k}\left\{\tau_l\left(X_i; \psi_k^{(s-1)}\right) + \tau_l\left(Y_i; \psi_k^{(s-1)}\right)\right\}}$$

and

$$\theta_j^{(s)} = \underset{\theta_j \in \Theta}{\arg\max}\ \frac{1}{n}\sum_{i=1}^{n}\left\{\tau_j\left(X_i; \psi_k^{(s-1)}\right) \log \varphi\left(X_i; \theta_j\right) + \tau_j\left(Y_i; \psi_k^{(s-1)}\right) \log \varphi\left(Y_i; \theta_j\right)\right\},$$

for each $j \in [k]$.

A noteworthy property of the MM sequence $\left(\psi_k^{(s)}\right)_{s\in\mathbb{N}}$ is that it generates an increasing sequence of objective values, due to the chain of inequalities

$$L_{h,n}\left(\psi_k^{(s-1)}\right) = Q_n\left(\psi_k^{(s-1)}, \psi_k^{(s-1)}\right) \leq Q_n\left(\psi_k^{(s)}, \psi_k^{(s-1)}\right) \leq L_{h,n}\left(\psi_k^{(s)}\right),$$

where the equality is due to property (i) of $Q_n$, the first in equality is due to the definition of $\psi_k^{(s)}$, and the second inequality is due to property (ii) of $Q_n$. This provides a kind of stability and regularity to the sequence $\left(L_{h,n}\left(\psi_k^{(s)}\right)\right)_{s\in\mathbb{N}}$.

Of course, we can provide stronger guarantees under additional assumptions. Namely, assume that (iii) $\Psi_k \subset \bar{\Psi}_k$, where $\bar{\Psi}_k$ is an open set in a finite dimensional Euclidean space on which $L_{h,n}$ and $Q_n\left(\cdot, \chi_k\right)$ is differentiable, for each $\chi_k \in \Psi_k$. Then, under assumptions (i)–(iii) regarding $L_{h,n}$ and $Q_n$, and due to the compactness of $\Psi_k$ and the continuity of $Q_n$ on $\Psi_k \times \Psi_k$, Razaviyayn et al. (2013, Cor. 1) implies that $\left(\psi_k^{(s)}\right)_{s\in\mathbb{N}}$ converges to the set of stationary points of $L_{h,n}$ in the sense that

$$\lim_{s\to\infty} \inf_{\psi_k^* \in \Psi_k^*} \left\|\psi_k^{(s)} - \psi_k^*\right\|_2 = 0,\ \text{where } \Psi_k^* = \left\{\psi_k^* \in \Psi_k:\ \left.\frac{\partial L_{h,n}}{\partial \psi_k}\right|_{\psi_k = \psi_k^*} = 0\right\}.$$

More concisely, we say that the sequence $\left(\psi_k^{(s)}\right)_{s\in\mathbb{N}}$ globally converges to the set of stationary points $\Psi_k^*$.

## 4.2 Experimental setup

Towards the task of demonstrating empirical evidence of the rates in Theorem 5, we consider the family of beta distributions on the unit interval $\mathcal{X} = [0,1]$ as our base class (i.e., (7)) to estimate a pair of target densities

$$f_1\left(x\right) = \frac{1}{2}\chi_{[0,2/5]}\left(x\right) + \frac{1}{2}\chi_{[3/5,1]}\left(x\right),$$

and

$$f_2\left(x\right) = \chi_{[0,1]}\left(x\right)\begin{cases} 2 - 4x & \text{if } x \leq 1/2, \\ -2 + 4x & \text{if } x > 1/2, \end{cases}$$

where $\chi_{\mathcal{A}}$ is the characteristic function that takes value 1 if $x \in \mathcal{A}$ and 0, otherwise. Note that neither $f_1$ nor $f_2$ are in $\mathcal{C}$. In particular, $f_1\left(x\right) = 0$ when $x \in \left(\frac{2}{5}, \frac{3}{5}\right)$, and $f_2(x) = 0$ when $x = 1/2$, and hence neither densities are bounded away from 0, on $\mathcal{X}$. Thus, the theory of Rakhlin et al. (2005) cannot be applied to provide bounds for the expected KL divergence between MLEs of beta mixtures and the pair of targets. We visualize $f_1$ and $f_2$ in Figure 1.

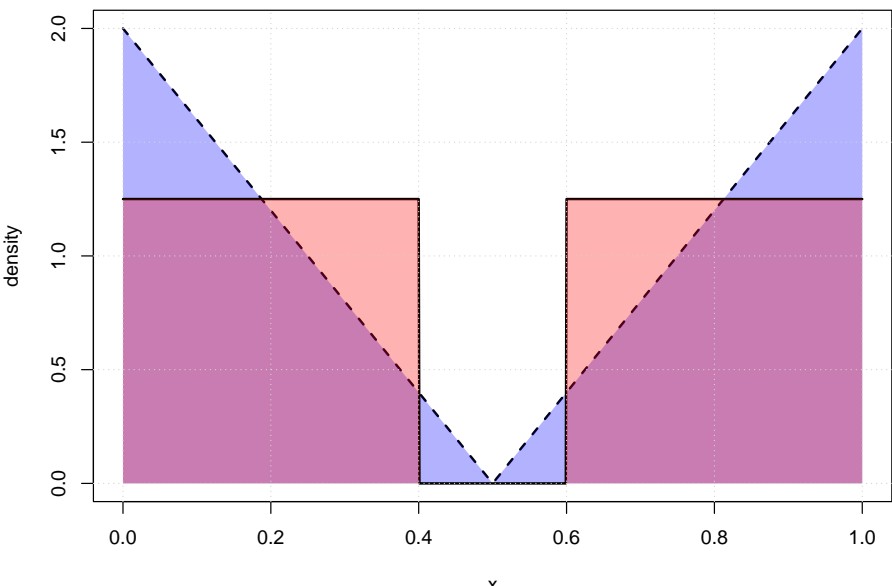

Figure 1: Simulation target densities $f_1$ (solid line) and $f_2$ (dashed line).

To observe the rate of decrease of the $h$-lifted KL divergence between the targets and respective sequences of $h$-MLLEs, we conduct two experiments **E1** and **E2**. In **E1**, our target density is set to $f_1$ and $h_1 = \beta\left(\cdot; 1/2, 1/2\right)$. For each $n \in \left\{2^{10}, \ldots, 2^{15}\right\}$ and $k \in \left\{2, \ldots, 8\right\}$, we independently simulate $\mathbf{X}_n$ and $\mathbf{Y}_n$ with each $X_i$ and $Y_i$ ($i \in [n]$), i.i.d., from the distributions characterized by $f_1$ and $h_1$, respectively. In **E2**, we target $f_2$ with $h$-MLLEs over the same ranges of $k$ and $n$, but with $h_2 = \beta\left(\cdot; 1, 1\right)$–the density of the uniform distribution. For each $k$ and $n$, we simulate $\mathbf{X}_n$ and $\mathbf{Y}_n$, respectively, from distributions characterized by $f_2$ and $h_2$.

In both experiments, we simulate $r = 50$ replicates of each $(k, n)$-scenario and compute the corresponding $h$-MLLEs, $(f_{k,n,l})_{l \in [r]}$, using the previously described MM algorithm. For each $l \in [r]$, we compute the corresponding negative log $h$-lifted likelihood between the target $f$ and $f_{k,n,l}$:

$$K_{k,n,l} = -\int_{\mathcal{X}} \left(f + h\right)\log\left(f_{k,n,l} + h\right)\mathrm{d}\mu$$

to assess the rates, and note that

$$\mathrm{KL}_h\left(f \,\|\, f_{k,n,l}\right) = \int_{\mathcal{X}} \left(f + h\right)\log\left(f + h\right)\mathrm{d}\mu + K_{k,n,l},$$

where the prior term is a constant with respect to $k$ and $n$.

To analyze the sample of $7 \times 6 \times 50 = 2100$ observations of relationship between the values $(K_{k,n,l})_{l \in [r]}$ and the corresponding values of $k$ and $n$, we use non-linear least squares (Amemiya, 1985, Sec. 4.3) to fit the regression relationship

$$\mathbf{E}\left[K_{k,n,l}\right] = a_0 + \frac{a_1}{(k+2)^{b_1}} + \frac{a_2}{n^{b_2}}. \tag{10}$$

We obtain 95% asymptotic confidence intervals for the estimates of the regression parameters $a_0$, $a_1$, $a_2$, $b_1$, $b_2 \in \mathbb{R}$, under the assumption of potential mis-specification of (10), by using the sandwich estimator for the asymptotic covariance matrix (cf. White 1982).

### 4.3 Results

We report the estimates along with 95% asymptotic confidence intervals for the parameters of (10) for **E1** and **E2** in Table 1. Plots of the average negative log $h$-lifted likelihood values by sample sizes $n$ and numbers of components $k$ are provided in Figure 2.

Table 1:  Estimates of parameters for fitted relationships (with 95% confidence intervals) between negative log $h$-lifted likelihood values, sample size and number of mixture components for experiments **E1** and **E2**.

| **E1** | $a_0$ | $a_1$ | $a_2$ | $b_1$ | $b_2$ |
|---|---|---|---|---|---|
| Est. | $-1.68$ | $0.73$ | $6.80$ | $1.87$ | $0.99$ |
| 95% CI | $(-1.68, -1.67)$ | $(0.68, 0.78)$ | $(1.24, 12.36)$ | $(1.81, 1.93)$ | $(0.87, 1.11)$ |
| **E2** | $a_0$ | $a_1$ | $a_2$ | $b_1$ | $b_2$ |
| Est. | $-1.47$ | $1.49$ | $6.75$ | $4.36$ | $1.07$ |
| 95% CI | $(-1.48, -1.47)$ | $(0.58, 2.41)$ | $(2.17, 11.32)$ | $(3.91, 4.81)$ | $(0.97, 1.16)$ |

From Table 1, we observe that $\mathbf{E}\left[K_{k,n,l}\right]$ decreases with both $n$ and $k$ in both simulations, and that the rates at which the averages decrease are faster than anticipated by Theorem 5, with respect to both $n$ and $k$. We can visually confirm the decreases in the estimate of $\mathbf{E}\left[K_{k,n,l}\right]$ via Figure 2. In both **E1** and **E2**, the rate of decrease over the assessed range of $n$ is approximately proportional to $1/n$, as opposed to the anticipated rate of $1/\sqrt{n}$, whereas the rate of decrease in $k$ is far larger, at approximately $1/k^{1.87}$ for **E1** and $1/k^{4.36}$ for **E2**.

These observations provide empirical evidence towards the fact that the rate of decrease of $\mathbf{E}\left[K_{k,n,l}\right]$ is at least $1/k$ and $1/\sqrt{n}$, respectively, for $k$ and $n$, at least over the simulation scenarios. These fast rates of fit over small values of $n$ and $k$ may be indicative of a diminishing returns of fit phenomenon, as discussed in Cadez & Smyth (2000) or the so-called elbow phenomenon (see, e.g., Ritter 2014, Sec. 4.2), whereupon the rate of decrease in average loss for small values of $k$ is fast and becomes slower as $k$ increases, converging to some asymptotic rate. This is also the reason why we do not include the outcomes when $k = 1$, as the drop in $\mathbf{E}\left[K_{k,n,l}\right]$ between $k = 1$ and $k = 2$ is so dramatic that it makes our simulated data ill-fitted by any model of form (10). As such, we do not view Theorem 5 as being pessimistic in light of these phenomena, as it applies uniformly over all values of $k$ and $n$.

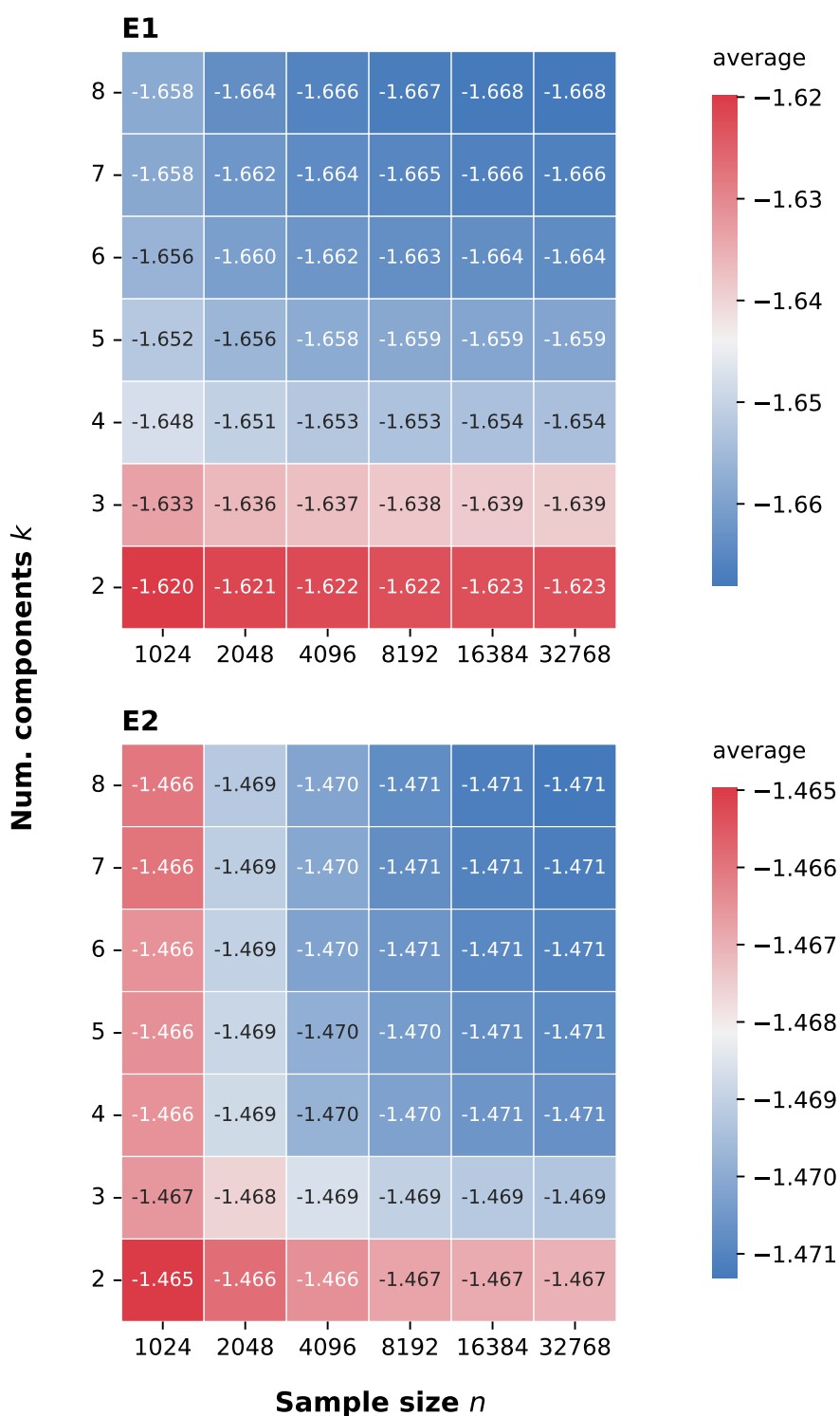

Figure 2: Average negative log $h$-lifted likelihood values by sample sizes $n$ and numbers of components $k$ for experiments **E1** and **E2**.

## 5 Conclusion

The estimation of probability densities using finite mixtures from some base class $\mathcal{P}$ appears often in machine learning and statistical inference as a natural method for modelling underlying data generating processes. In this work, we pursue novel generalization bounds for such mixture estimators. To this end, we introduce the family of $h$-lifted KL divergences for densities on compact supports, within the family of Bregman divergences, which correspond to risk functions that can be bounded, even when densities in the class $\mathcal{P}$ are not bounded away from zero, unlike the standard KL divergence.

Unlike the least-squares loss, the corresponding maximum $h$-likelihood estimation problem can be computed via an MM algorithm, mirroring the availability of EM algorithms for the maximum likelihood problem corresponding to the KL divergence. Along with our derivations of generalization bounds that achieve the same rates as the best-known bounds for the KL divergence and least square loss, we also provide numerical evidence towards the correctness of these bounds in the case when $\mathcal{P}$ corresponds to beta densities.

Aside from beta distributions, mixture densities on compact supports that can be analysed under our framework appear frequently in the literature. For supports on compact Euclidean subset, examples include mixtures of Dirichlet distributions (Fan et al., 2012) and bivariate binomial distributions (Papageorgiou & David, 1994). Alternatively, one can consider distributions on compact Euclidean manifolds, such as mixtures of Kent (Peel et al., 2001) distributions and von Mises–Fisher distributions (Banerjee et al., 2005, Ng & Kwong, 2022). We defer investigating the practical performance of the maximum $h$-lifted likelihood estimators and accompanying theory for such models to future work.

### Acknowledgments

We express sincere gratitude to the Reviewers and Action Editor for their valuable feedback, which has helped to improve the quality of this paper. Hien Duy Nguyen and TrungTin Nguyen acknowledge funding from the Australian Research Council grant DP230100905.

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

# A Proofs of main results

The following section is devoted to establishing some technical definitions and instrumental results which are used to prove Theorem 5 and Corollary 6, and also includes the proofs of these results themselves.

## A.1 Preliminaries

Recall that we are interested in bounds of the form (2). Note that $\mathcal{P}$ is a subset of the linear space

$$
\mathcal{V} = \mathrm{cl}\left(\bigcup_{k\in\mathbb{N}}\left\{\sum_{j=1}^{k}\varpi_j\varphi\left(\cdot;\theta_j\right) \mid \varphi\left(\cdot;\theta_j\right)\in\mathcal{P}, \varpi_j\in\mathbb{R}, j\in[k]\right\}\right),
$$

and hence we can apply the following result, paraphrased from Zhang (2003, Thm. II.1).

**Lemma 8.** *Let $\kappa$ be a differentiable and convex function on $\mathcal{V}$, and let $\left(\bar{f}_k\right)_{k\in\mathbb{N}}$ be a sequence of functions obtained by Algorithm 1. If*

$$
\sup_{p,q\in\mathcal{C},\pi\in(0,1)}\frac{\mathrm{d}^2}{\mathrm{d}\pi^2}\kappa\left((1-\pi)\,p+\pi q\right)\leq\mathfrak{M}<\infty,
$$

*then, for each $k\in\mathbb{N}$,*

$$
\kappa\left(\bar{f}_k\right)-\inf_{p\in\mathcal{C}}\kappa\left(p\right)\leq\frac{2\mathfrak{M}}{k+2}.
$$

---

**Algorithm 1** Algorithm for computing a greedy approximation sequence.

---

**Require:** $\bar{f}_0\in\mathcal{P}$
1: **for** $k\in\mathbb{N}$ **do**
2:      Compute $\left(\bar{\pi}_k,\bar{\theta}_k\right)=\underset{(\pi,\theta)\in[0,1]\times\Theta}{\arg\min}\ \kappa\left((1-\pi)\,\bar{f}_{k-1}+\pi\varphi\left(\cdot;\theta\right)\right)$
3:      Define $\bar{f}_k=(1-\bar{\pi}_k)\,\bar{f}_{k-1}+\bar{\pi}_k\varphi\left(\cdot;\bar{\theta}_k\right)$
4: **end for**

---

We are interested in two choices for $\kappa$:

$$
\kappa\left(p\right)=\mathrm{KL}_h\left(f\,||\,p\right) \tag{11}
$$

and its sample counterpart,

$$
\kappa_n\left(p\right)=\frac{1}{n}\sum_{i=1}^{n}\log\frac{f\left(X_i\right)+h\left(X_i\right)}{p\left(X_i\right)+h\left(X_i\right)}+\frac{1}{n}\sum_{i=1}^{n}\log\frac{f\left(Y_i\right)+h\left(Y_i\right)}{p\left(Y_i\right)+h\left(Y_i\right)}, \tag{12}
$$

where $(X_i)_{i\in[n]}$ and $(Y_i)_{i\in[n]}$ are realisations of $X$ and $Y$, respectively. We obtain the following important results.

**Proposition 9.** *Let $\varkappa$ denote either $\kappa$, the $\mathrm{KL}_h$ divergence (11), or $\kappa_n$, the sample $\mathrm{KL}_h$ divergence (12), and assume that $h\geq a$ and $\varphi\left(\cdot;\theta\right)\leq c$, for each $\theta\in\Theta$. Then,*

$$
\varkappa\left(\bar{f}_k\right)-\inf_{p\in\mathcal{C}}\varkappa\left(p\right)\leq\frac{4a^{-2}c^2}{k+2},
$$

*for each $k\in\mathbb{N}$, where $\left(\bar{f}_k\right)_{k\in\mathbb{N}}$ is obtained as per Algorithm 1.*

*Proof.* See Appendix C.4. □

Notice that sequences $\left(\bar{f}_k\right)_{k\in\mathbb{N}}$ obtained via Algorithm 1 are greedy approximation sequences, and that $\bar{f}_k\in\mathcal{C}_k$, for each $k\in\mathbb{N}$. Let $(f_k)_{k\in\mathbb{N}}$ be the sequence of minimizers defined by

$$
f_k=\underset{\psi_k\in\mathcal{S}_k\times\Theta^k}{\arg\min}\ \mathrm{KL}_h\left(f\,||\,f_k\left(\cdot;\psi_k\right)\right), \tag{13}
$$

and let $(f_{k,n})_{k \in \mathbb{N}}$ be the sequence of $h$-MLLEs, as per (5). Then, by definition, we have the fact that $\kappa(f_k) \leq \kappa(\bar{f}_k)$ and $\kappa(f_{k,n}) \leq \kappa(\bar{f}_k)$, for $\kappa$ set as (11) or (12), respectively. Thus, we have the following result.

**Proposition 10.** *For the* $\mathrm{KL}_h$ *divergence (11), under the assumption that* $h \geq a$ *and* $\varphi(\cdot; \theta) \leq c$, *for each* $\theta \in \Theta$, *we have*

$$\kappa(f_k) - \inf_{p \in \mathcal{C}} \kappa(p) \leq \frac{4a^{-2}c^2}{k+2} \tag{14}$$

*for each* $k \in \mathbb{N}$, *where* $(f_k)_{k \in \mathbb{N}}$ *is the sequence of minimizers defined via (13). Furthermore, for the sample* $\mathrm{KL}_h$ *divergence (12), under the same assumptions as above, we have*

$$\kappa_n(f_{k,n}) - \inf_{p \in \mathcal{C}} \kappa_n(p) \leq \frac{4a^{-2}c^2}{k+2}, \tag{15}$$

*for each* $k \in \mathbb{N}$, *where* $(f_{k,n})_{k \in \mathbb{N}}$ *are* $h$-*MLLEs defined via (5).*

As is common in many statistical learning/uniform convergence results (e.g., Bartlett & Mendelson, 2002, Koltchinskii & Panchenko, 2004), we employ the use of Rademacher processes and associated bounds. Let $(\varepsilon_i)_{i \in [n]}$ be i.i.d. Rademacher random variables, that is $\mathbf{P}(\varepsilon_i = -1) = \mathbf{P}(\varepsilon_i = 1) = 1/2$, that are independent of $(X_i)_{i \in [n]}$. The Rademacher process, indexed by a class of real measurable functions $\mathcal{S}$, is defined as the quantity

$$R_n(s) = \frac{1}{n} \sum_{i=1}^{n} s(X_i)\varepsilon_i,$$

for $s \in \mathcal{S}$. The Rademacher complexity of the class $\mathcal{S}$ is given by $\mathcal{R}_n(\mathcal{S}) = \mathbf{E} \sup_{s \in \mathcal{S}} |R_n(s)|$.

In the subsequent section, we make use of the following result regarding the supremum of convex functions:

**Lemma 11** (Rockafellar, 1997, Thm. 32.2). *Let* $\eta$ *be a convex function on a linear space* $\mathcal{T}$, *and let* $\mathcal{S} \subset \mathcal{T}$ *be an arbitrary subset. Then,*

$$\sup_{p \in \mathcal{S}} \eta(p) = \sup_{p \in \mathrm{co}(\mathcal{S})} \eta(p).$$

In particular, we use the fact that since a linear functional of convex combinations achieves its maximum value at vertices, the Rademacher complexity of $\mathcal{S}$ is equal to the Rademacher complexity of $\mathrm{co}(\mathcal{S})$ (see Lemma 21). We consequently obtain the following result.

**Lemma 12.** *Let* $(\varepsilon_i)_{i \in [n]}$ *be i.i.d. Rademacher random variables, independent of* $(X_i)_{i \in [n]}$ *and* $\mathcal{P}$ *be defined as above. The sets* $\mathcal{C}$ *and* $\mathcal{P}$ *will have equal complexity,* $\mathcal{R}_n(\mathcal{C}) = \mathcal{R}_n(\mathcal{P})$, *and the supremum of the Rademacher process indexed by* $\mathcal{C}$ *is equal to the supremum on the basis functions of* $\mathcal{P}$:

$$\mathbf{E}_\varepsilon \sup_{g \in \mathcal{C}} \left| \frac{1}{n} \sum_{i=1}^{n} g(X_i)\varepsilon_i \right| = \mathbf{E}_\varepsilon \sup_{\theta \in \Theta} \left| \frac{1}{n} \sum_{i=1}^{n} \varphi(X_i; \theta)\varepsilon_i \right|.$$

*Proof.* Follows immediately from Lemma 11. $\square$

### A.2 Proofs

We first present a result establishing a uniform concentration bound for the $h$-lifted log-likelihood ratios, which is instrumental in the proof of Theorem 5. Our proofs broadly follow the structure of Rakhlin et al. (2005), modified as needed for the use of $\mathrm{KL}_h$.

Assume that $0 \leq \varphi(\cdot; \theta) < c$ for some $c \in \mathbb{R}_{>0}$. For brevity, we adopt the notation: $\|T(g)\|_{\mathcal{C}} = \sup_{g \in \mathcal{C}} |T(g)|$.

**Theorem 13.** *Let* $X_1, \ldots, X_n$ *be an i.i.d. sample of size* $n$ *drawn from a fixed density* $f$ *such that* $0 \leq f(x) \leq c$ *for all* $x \in \mathcal{X}$, *and let* $h$ *be a positive density with* $0 < a \leq h(x) \leq b$ *for all* $x \in \mathcal{X}$. *Then, for each* $t > 0$, *with probability at least* $1 - \mathrm{e}^{-t}$,

$$\left\| \frac{1}{n} \sum_{i=1}^{n} \log \frac{g(X_i) + h(X_i)}{f(X_i) + h(X_i)} - \mathbf{E}_f \log \frac{g+h}{f+h} \right\|_{\mathcal{C}} \leq \frac{w_1}{\sqrt{n}} \mathbf{E} \int_0^c \log^{1/2} N(\mathcal{P}, \varepsilon, d_{n,x}) \mathrm{d}\varepsilon + \frac{w_2}{\sqrt{n}} + w_3 \sqrt{\frac{t}{n}},$$

where $w_1$, $w_2$, and $w_3$ are constants that each depend on some or all of $a$, $b$, and $c$, and $N(\mathcal{P}, \varepsilon, d_{n,x})$ is the $\varepsilon$-covering number of $\mathcal{P}$ with respect to the following empirical $L_2$ metric

$$d_{n,x}^2(\varphi_1, \varphi_2) = \frac{1}{n}\sum_{i=1}^n (\varphi_1(X_i) - \varphi_2(X_i))^2.$$

**Remark 14.** *The bound on the term*

$$\left\| \frac{1}{n}\sum_{i=1}^n \log \frac{g(Y_i) + h(Y_i)}{f(Y_i) + h(Y_i)} - \mathbf{E}_h \log \frac{g+h}{f+h} \right\|_{\mathcal{C}}$$

*is the same as the above, except where the empirical distance $d_{n,x}$ is replaced by $d_{n,y}$, defined in the same way as $d_{n,x}$ but with $Y_i$ replacing $X_i$.*

*Proof of Theorem 13.* Fix $h$ and define the following quantities: $\tilde{g} = g + h$, $\tilde{f} = f + h$, $\tilde{\mathcal{C}} = \mathcal{C} + h$,

$$m_i = \log \frac{\tilde{g}(X_i)}{\tilde{f}(X_i)}, \quad m_i' = \log \frac{\tilde{g}(X_i')}{\tilde{f}(X_i')}, \quad Z(x_1, \ldots, x_n) = \left\| \frac{1}{n}\sum_{i=1}^n \log \frac{\tilde{g}(X_i)}{\tilde{f}(X_i)} - \mathbf{E}\log \frac{\tilde{g}}{\tilde{f}} \right\|_{\tilde{\mathcal{C}}}.$$

We first apply McDiarmid's inequality (Lemma 23) to the random variable $Z$. The bound on the martingale difference is given by

$$
\begin{aligned}
|Z(X_1, \ldots, X_i, \ldots, X_n) - Z(X_1, \ldots, X_i', \ldots, X_n)| &= \left| \left\| \mathbf{E}\log \frac{\tilde{g}}{\tilde{f}} - \frac{1}{n}(m_1 + \ldots + m_i + \ldots + m_n) \right\|_{\tilde{\mathcal{C}}} \right. \\
&\quad \left. - \left\| \mathbf{E}\log \frac{\tilde{g}}{\tilde{f}} - \frac{1}{n}(m_1 + \ldots + m_i' + \ldots + m_n) \right\|_{\tilde{\mathcal{C}}} \right| \\
&\leq \frac{1}{n}\left\| \log \frac{\tilde{g}(X_i')}{\tilde{f}(X_i')} - \log \frac{\tilde{g}(X_i)}{\tilde{f}(X_i)} \right\|_{\tilde{\mathcal{C}}} \\
&\leq \frac{1}{n}\left( \log \frac{c+b}{a} - \log \frac{a}{c+b} \right) = \frac{1}{n}2\log\frac{c+b}{a} = c_i.
\end{aligned}
$$

The chain of inequalities holds because of the triangle inequality and the properties of the supremum. By Lemma 23, we have

$$\mathbf{P}(Z - \mathbf{E}\,Z > \varepsilon) \leq \exp\left\{ -\frac{n\varepsilon^2}{(\sqrt{2}\log\frac{c+b}{a})^2} \right\},$$

so

$$\mathbf{P}(Z \leq \varepsilon + \mathbf{E}\,Z) \geq 1 - \exp\left\{ -\frac{n\varepsilon^2}{(\sqrt{2}\log\frac{c+b}{a})^2} \right\},$$

where it follows from $t = n\varepsilon^2/(\sqrt{2}\log\frac{c+b}{a})^2$ that $\varepsilon = \sqrt{2}\log\left(\frac{c+b}{a}\right)\sqrt{\frac{t}{n}}$. Therefore with probability at least $1 - \mathrm{e}^{-t}$,

$$\left\| \frac{1}{n}\sum_{i=1}^n \log \frac{\tilde{g}(X_i)}{\tilde{f}(X_i)} - \mathbf{E}_f \log \frac{\tilde{g}}{\tilde{f}} \right\|_{\tilde{\mathcal{C}}} \leq \mathbf{E}\left\| \frac{1}{n}\sum_{i=1}^n \log \frac{\tilde{g}(X_i)}{\tilde{f}(X_i)} - \mathbf{E}_f \log \frac{\tilde{g}}{\tilde{f}} \right\|_{\tilde{\mathcal{C}}} + \sqrt{2}\log\left(\frac{c+b}{a}\right)\sqrt{\frac{t}{n}}.$$

Let $(\varepsilon_i)_{i\in[n]}$ be i.i.d. Rademacher random variables, independent of $(X_i)_{i\in[n]}$. By Lemma 24,

$$\mathbf{E}\left\| \frac{1}{n}\sum_{i=1}^n \log \frac{\tilde{g}(X_i)}{\tilde{f}(X_i)} - \mathbf{E}_f \log \frac{\tilde{g}}{\tilde{f}} \right\|_{\tilde{\mathcal{C}}} \leq 2\,\mathbf{E}\left\| \frac{1}{n}\sum_{i=1}^n \log \frac{\tilde{g}(X_i)}{\tilde{f}(X_i)}\varepsilon_i \right\|_{\tilde{\mathcal{C}}}.$$

By combining the results above, the following inequality holds with probability at least $1 - \mathrm{e}^{-t}$

$$\left\| \frac{1}{n}\sum_{i=1}^n \log \frac{\tilde{g}(X_i)}{\tilde{f}(X_i)} - \mathbf{E}_f \log \frac{\tilde{g}}{\tilde{f}} \right\|_{\tilde{\mathcal{C}}} \leq 2\,\mathbf{E}\left\| \frac{1}{n}\sum_{i=1}^n \log \frac{\tilde{g}(X_i)}{\tilde{f}(X_i)}\varepsilon_i \right\|_{\tilde{\mathcal{C}}} + \sqrt{2}\log\left(\frac{c+b}{a}\right)\sqrt{\frac{t}{n}}.$$

Now let $p_i = \frac{\tilde{g}(X_i)}{\tilde{f}(X_i)} - 1$, such that $\frac{a}{c+b} \leq p_i + 1 \leq \frac{c+b}{a}$ holds for all $i \in [n]$. Additionally, let $\eta(p) = \log(p+1)$ so that $\eta(0) = 0$ and note that for $p \in \left[\frac{a}{c+b} - 1, \frac{c+b}{a} - 1\right]$, the derivative of $\eta(p)$ is maximal at $p^* = \frac{a}{c+b} - 1$, and equal to $\eta'(p^*) = (c+b)/a$. Therefore, $\frac{a}{b+c}\log(p+1)$ is 1-Lipschitz. By Lemma 22 applied to $\eta(p)$,

$$
2\,\mathbf{E}\left\|\frac{1}{n}\sum_{i=1}^{n}\log\frac{\tilde{g}(X_i)}{\tilde{f}(X_i)}\varepsilon_i\right\|_{\tilde{\mathcal{C}}} = 2\,\mathbf{E}\left\|\frac{1}{n}\sum_{i=1}^{n}\eta(p_i)\varepsilon_i\right\|_{\tilde{\mathcal{C}}} \leq \frac{4(c+b)}{a}\,\mathbf{E}\left\|\frac{1}{n}\sum_{i=1}^{n}\frac{\tilde{g}(X_i)}{\tilde{f}(X_i)}\varepsilon_i - \frac{1}{n}\sum_{i=1}^{n}\varepsilon_i\right\|_{\tilde{\mathcal{C}}}
$$

$$
\leq \frac{4(c+b)}{a}\,\mathbf{E}\left\|\frac{1}{n}\sum_{i=1}^{n}\frac{\tilde{g}(X_i)}{\tilde{f}(X_i)}\varepsilon_i\right\|_{\tilde{\mathcal{C}}} + \frac{4(c+b)}{a}\mathbf{E}_\varepsilon\left|\frac{1}{n}\sum_{i=1}^{n}\varepsilon_i\right|
$$

$$
\leq \frac{4(c+b)}{a}\,\mathbf{E}\left\|\frac{1}{n}\sum_{i=1}^{n}\frac{\tilde{g}(X_i)}{\tilde{f}(X_i)}\varepsilon_i\right\|_{\tilde{\mathcal{C}}} + \frac{4(c+b)}{a}\frac{1}{\sqrt{n}},
$$

where the final inequality follows from the following result, proved in Haagerup (1981):

$$
\mathbf{E}_\varepsilon\left|\frac{1}{n}\sum_{i=1}^{n}\varepsilon_i\right| \leq \left(\mathbf{E}_\varepsilon\left\{\frac{1}{n}\sum_{i=1}^{n}\varepsilon_i\right\}^2\right)^{1/2} = \frac{1}{\sqrt{n}}.
$$

Now, let $\xi_i(\tilde{g}_i) = a \cdot \tilde{g}(X_i)/\tilde{f}(X_i)$, and note that

$$
|\xi_i(u_i) - \xi_i(v_i)| = \frac{a}{|\tilde{f}(X_i)|}|u(X_i) - v(X_i)| \leq |u(X_i) - v(X_i)|.
$$

By again applying Lemma 22, we have

$$
\frac{4(c+b)}{a}\,\mathbf{E}\left\|\frac{1}{n}\sum_{i=1}^{n}\frac{\tilde{g}(X_i)}{\tilde{f}(X_i)}\varepsilon_i\right\|_{\tilde{\mathcal{C}}} \leq \frac{8(c+b)}{a^2}\,\mathbf{E}\left\|\frac{1}{n}\sum_{i=1}^{n}\tilde{g}(X_i)\varepsilon_i\right\|_{\tilde{\mathcal{C}}}
$$

$$
\leq \frac{8(c+b)}{a^2}\,\mathbf{E}\left\|\frac{1}{n}\sum_{i=1}^{n}g(X_i)\varepsilon_i\right\|_{\mathcal{C}} + \frac{8(c+b)}{a^2}\,\mathbf{E}\left|\frac{1}{n}\sum_{i=1}^{n}h(X_i)\varepsilon_i\right|
$$

$$
\leq \frac{8(c+b)}{a^2}\,\mathbf{E}\left\|\frac{1}{n}\sum_{i=1}^{n}g(X_i)\varepsilon_i\right\|_{\mathcal{C}} + \frac{8(c+b)}{a^2}\frac{b}{\sqrt{n}}.
$$

Applying Lemmas 12 and 25, the following inequality holds for some constant $K > 0$:

$$
\mathbf{E}_\varepsilon\sup_{g\in\mathcal{C}}\left|\frac{1}{n}\sum_{i=1}^{n}g(X_i)\varepsilon_i\right| = \mathbf{E}_\varepsilon\sup_{\theta\in\Theta}\left|\frac{1}{n}\sum_{i=1}^{n}\varphi(X_i;\theta)\varepsilon_i\right| \leq \frac{K}{\sqrt{n}}\,\mathbf{E}\int_0^c \log^{1/2}N(\mathcal{P},\varepsilon,d_{n,x})\mathrm{d}\varepsilon, \tag{16}
$$

and combining the results together, the following inequality holds with probability at least $1 - \mathrm{e}^{-t}$:

$$
\left\|\frac{1}{n}\sum_{i=1}^{n}\log\frac{\tilde{g}(X_i)}{\tilde{f}(X_i)} - \mathbf{E}_f\log\frac{\tilde{g}}{\tilde{f}}\right\| \leq \frac{8(c+b)K}{a^2\sqrt{n}}\,\mathbf{E}\int_0^c\log^{1/2}N(\mathcal{P},\varepsilon,d_{n,x})\mathrm{d}\varepsilon + \frac{(8b+4a)(c+b)}{a^2\sqrt{n}} + \sqrt{2}\log\left(\frac{c+b}{a}\right)\sqrt{\frac{t}{n}}
$$

$$
= \frac{w_1}{\sqrt{n}}\,\mathbf{E}\int_0^c\log^{1/2}N(\mathcal{P},\varepsilon,d_{n,x})\mathrm{d}\varepsilon + \frac{w_2}{\sqrt{n}} + w_3\sqrt{\frac{t}{n}}, \tag{17}
$$

where $w_1$, $w_2$, and $w_3$ are constants that each depend on some or all of $a$, $b$, and $c$. $\qquad\square$

**Remark 15.** *From Lemma 25 we have that $\sigma_n^2 := \sup_{f\in\mathscr{F}} P_n f^2$. To make explicit why $2\sigma_n = \left(\sup_{g\in\mathcal{C}} P_n g^2\right)^{1/2} = 2c$, let $\mathscr{F} = \mathcal{C}$ and observe*

$$
\sigma_n^2 = \sup_{g\in\mathcal{C}}P_n g^2 = \sup_{g\in\mathcal{C}}\frac{1}{n}\sum_{i=1}^{n}g(X_i)^2 \leq \frac{1}{n}\sum_{i=1}^{n}c^2 = c^2.
$$

*Since our basis functions $\varphi(\cdot, \theta)$ are bounded by $c$, everything greater than $c$ will have value $0$ and hence the change from $2c$ to $c$ is inconsequential. However, it can also be motivated by the fact that $\varphi(\cdot, \theta)$ are positive functions.*

As highlighted in Remark 14, the full result of Theorem 13 relies on the empirical $L_2$ distances $d_{n,x}$ and $d_{n,y}$. In the result of Theorem 5, we make use of the following result to bound $d_{n,x}$ and $d_{n,y}$.

**Proposition 16.** *By combining Lemmas 18 and 19, the following inequalities holds:*

$$\log N(\mathcal{P}, \varepsilon, \|\cdot\|) \leq \log N_{[]}(\mathcal{P}, \varepsilon, \|\cdot\|) \leq \log N(\mathcal{P}, \varepsilon/2, \|\cdot\|_\infty),$$

*where $N_{[]}(\mathcal{P}, \varepsilon, \|\cdot\|)$ is the $\varepsilon$-bracketing number of $\mathcal{P}$. Therefore, we have that*

$$\log N(\mathcal{P}, \varepsilon, d_{n,x}) \leq \log N(\mathcal{P}, \varepsilon/2, \|\cdot\|_\infty), \text{ and } \log N(\mathcal{P}, \varepsilon, d_{n,y}) \leq \log N(\mathcal{P}, \varepsilon/2, \|\cdot\|_\infty).$$

With this result, we can now prove Theorem 5.

*Proof (of Theorem 5).* The notation is the same as in the proof of Theorem 13. The values of the constants may change from line to line.

$$\mathrm{KL}_h\left(f \,\|\, f_{k,n}\right) - \mathrm{KL}_h\left(f \,\|\, f_k\right) = \mathbf{E}_f \log \frac{\tilde{f}}{\tilde{f}_{k,n}} + \mathbf{E}_h \log \frac{\tilde{f}}{\tilde{f}_{k,n}} - \mathbf{E}_f \log \frac{\tilde{f}}{\tilde{f}_k} - \mathbf{E}_h \log \frac{\tilde{f}}{\tilde{f}_k}$$

$$= \mathbf{E}_f \log \frac{\tilde{f}}{\tilde{f}_{k,n}} - \frac{1}{n}\sum_{i=1}^n \log \frac{\tilde{f}(X_i)}{\tilde{f}_{k,n}(X_i)} + \frac{1}{n}\sum_{i=1}^n \log \frac{\tilde{f}(X_i)}{\tilde{f}_{k,n}(X_i)} + \mathbf{E}_h \log \frac{\tilde{f}}{\tilde{f}_{k,n}} - \frac{1}{n}\sum_{i=1}^n \log \frac{\tilde{f}(Y_i)}{\tilde{f}_{k,n}(Y_i)} + \frac{1}{n}\sum_{i=1}^n \log \frac{\tilde{f}(Y_i)}{\tilde{f}_{k,n}(Y_i)}$$

$$- \mathbf{E}_f \log \frac{\tilde{f}}{\tilde{f}_k} + \frac{1}{n}\sum_{i=1}^n \log \frac{\tilde{f}(X_i)}{\tilde{f}_k(X_i)} - \frac{1}{n}\sum_{i=1}^n \log \frac{\tilde{f}(X_i)}{\tilde{f}_k(X_i)} - \mathbf{E}_h \log \frac{\tilde{f}}{\tilde{f}_k} + \frac{1}{n}\sum_{i=1}^n \log \frac{\tilde{f}(Y_i)}{\tilde{f}_k(Y_i)} - \frac{1}{n}\sum_{i=1}^n \log \frac{\tilde{f}(Y_i)}{\tilde{f}_k(Y_i)}$$

$$= \left( \mathbf{E}_f \log \frac{\tilde{f}}{\tilde{f}_{k,n}} - \frac{1}{n}\sum_{i=1}^n \log \frac{\tilde{f}(X_i)}{\tilde{f}_{k,n}(X_i)} \right) + \left( \mathbf{E}_h \log \frac{\tilde{f}}{\tilde{f}_{k,n}} - \frac{1}{n}\sum_{i=1}^n \log \frac{\tilde{f}(Y_i)}{\tilde{f}_{k,n}(Y_i)} \right)$$

$$+ \left( \frac{1}{n}\sum_{i=1}^n \log \frac{\tilde{f}(X_i)}{\tilde{f}_k(X_i)} - \mathbf{E}_f \log \frac{\tilde{f}}{\tilde{f}_k} \right) + \left( \frac{1}{n}\sum_{i=1}^n \log \frac{\tilde{f}(Y_i)}{\tilde{f}_k(Y_i)} - \mathbf{E}_h \log \frac{\tilde{f}}{\tilde{f}_k} \right)$$

$$+ \left( \frac{1}{n}\sum_{i=1}^n \log \frac{\tilde{f}(X_i)}{\tilde{f}_{k,n}(X_i)} - \frac{1}{n}\sum_{i=1}^n \log \frac{\tilde{f}(X_i)}{\tilde{f}_k(X_i)} \right) + \left( \frac{1}{n}\sum_{i=1}^n \log \frac{\tilde{f}(Y_i)}{\tilde{f}_{k,n}(Y_i)} - \frac{1}{n}\sum_{i=1}^n \log \frac{\tilde{f}(Y_i)}{\tilde{f}_k(Y_i)} \right)$$

$$\leq 2\sup_{\tilde{g}\in\tilde{\mathcal{C}}} \left| \frac{1}{n}\sum_{i=1}^n \log \frac{\tilde{g}(X_i)}{\tilde{f}(X_i)} - \mathbf{E}_f \log \frac{\tilde{g}}{\tilde{f}} \right| + 2\sup_{\tilde{g}\in\tilde{\mathcal{C}}} \left| \frac{1}{n}\sum_{i=1}^n \log \frac{\tilde{g}(Y_i)}{\tilde{f}(Y_i)} - \mathbf{E}_h \log \frac{\tilde{g}}{\tilde{f}} \right|$$

$$+ \left( \frac{1}{n}\sum_{i=1}^n \log \frac{\tilde{f}(X_i)}{\tilde{f}_{k,n}(X_i)} - \frac{1}{n}\sum_{i=1}^n \log \frac{\tilde{f}(X_i)}{\tilde{f}_k(X_i)} \right) + \left( \frac{1}{n}\sum_{i=1}^n \log \frac{\tilde{f}(Y_i)}{\tilde{f}_{k,n}(Y_i)} - \frac{1}{n}\sum_{i=1}^n \log \frac{\tilde{f}(Y_i)}{\tilde{f}_k(Y_i)} \right)$$

$$\leq 2\mathbf{E}\left\{ \frac{w_1^x}{\sqrt{n}} \int_0^c \log^{1/2} N(\mathcal{P}, \varepsilon, d_{n,x})\mathrm{d}\varepsilon \right\} + \frac{w_2^x}{\sqrt{n}} + w_3^x \sqrt{\frac{t}{n}} + \frac{1}{n}\sum_{i=1}^n \log \frac{\tilde{f}_k(X_i)}{\tilde{f}_{k,n}(X_i)}$$

$$+ 2\mathbf{E}\left\{ \frac{w_1^y}{\sqrt{n}} \int_0^c \log^{1/2} N(\mathcal{P}, \varepsilon, d_{n,y})\mathrm{d}\varepsilon \right\} + \frac{w_2^y}{\sqrt{n}} + w_3^y \sqrt{\frac{t}{n}} + \frac{1}{n}\sum_{i=1}^n \log \frac{\tilde{f}_k(Y_i)}{\tilde{f}_{k,n}(Y_i)}$$

$$\leq \frac{w_1}{\sqrt{n}} \int_0^c \log^{1/2} N(\mathcal{P}, \varepsilon/2, \|\cdot\|_\infty)\mathrm{d}\varepsilon + \frac{w_2}{\sqrt{n}} + w_3 \sqrt{\frac{t}{n}} + \frac{1}{n}\sum_{i=1}^n \log \frac{\tilde{f}_k(X_i)}{\tilde{f}_{k,n}(X_i)} + \frac{1}{n}\sum_{i=1}^n \log \frac{\tilde{f}_k(Y_i)}{\tilde{f}_{k,n}(Y_i)},$$

with probability at least $1 - \mathrm{e}^{-t}$, by Theorem 13. Now, we can use (15) from Proposition 10 applied to the target density $f_k$, obtaining the following:

$$\mathrm{KL}_h\left(f_k \,\|\, f_{k,n}\right) = \frac{1}{n}\sum_{i=1}^n \log \frac{\tilde{f}_k(X_i)}{\tilde{f}_{k,n}(X_i)} + \frac{1}{n}\sum_{i=1}^n \log \frac{\tilde{f}_k(Y_i)}{\tilde{f}_{k,n}(Y_i)} \leq \frac{4a^{-2}c^2}{k+2} + \inf_{p\in\mathcal{C}} \mathrm{KL}_h\left(f_k \,\|\, p\right).$$

Since by definition we have that $f_k \in \mathcal{C}$, $\inf_{p \in \mathcal{C}} \mathrm{KL}_h(f_k \| p) = 0$, and so with probability at least $1 - \mathrm{e}^{-t}$ we have:

$$\mathrm{KL}_h(f \| f_{k,n}) - \mathrm{KL}_h(f \| f_k) \le \frac{w_1}{\sqrt{n}} \int_0^c \log^{1/2} N(\mathcal{P}, \varepsilon/2, \| \cdot \|_\infty) \mathrm{d}\varepsilon + \frac{w_2}{\sqrt{n}} + w_3 \sqrt{\frac{t}{n}} + \frac{w_4}{k+2}. \tag{18}$$

We can write the overall error as the sum of the approximation and estimation errors as follows. The former is bounded by (14), and the latter is bounded as above in (18). Therefore, with probability at least $1 - \mathrm{e}^{-t}$,

$$\mathrm{KL}_h(f \| f_{k,n}) - \mathrm{KL}_h(f \| \mathcal{C}) = [\mathrm{KL}_h(f \| f_k) - \mathrm{KL}_h(f \| \mathcal{C})] + [\mathrm{KL}_h(f \| f_{k,n}) - \mathrm{KL}_h(f \| f_k)]$$

$$\le \frac{w_4}{k+2} + \frac{w_1}{\sqrt{n}} \int_0^c \log^{1/2} N(\mathcal{P}, \varepsilon/2, \| \cdot \|_\infty) \mathrm{d}\varepsilon + \frac{w_2}{\sqrt{n}} + w_3 \sqrt{\frac{t}{n}}. \tag{19}$$

As in Rakhlin et al. (2005), we rewrite the above probabilistic statement as a statement in terms of expectations. To this end, let

$$\mathcal{A} := \frac{w_4}{k+2} + \frac{w_1}{\sqrt{n}} \int_0^c \log^{1/2} N(\mathcal{P}, \varepsilon/2, \| \cdot \|_\infty) \mathrm{d}\varepsilon + \frac{w_2}{\sqrt{n}},$$

and $\mathcal{Z} := \mathrm{KL}_h(f \| f_{k,n}) - \mathrm{KL}_h(f \| \mathcal{C})$. We have shown $\mathbf{P}\left(\mathcal{Z} \ge \mathcal{A} + w_3 \sqrt{\frac{t}{n}}\right) \le \mathrm{e}^{-t}$. Since $\mathcal{Z} \ge 0$,

$$\mathbf{E}\{\mathcal{Z}\} = \int_0^{\mathcal{A}} \mathbf{P}(\mathcal{Z} > s) \mathrm{d}s + \int_{\mathcal{A}}^\infty \mathbf{P}(\mathcal{Z} > s) \mathrm{d}s \le \mathcal{A} + \int_0^\infty \mathbf{P}(\mathcal{Z} > \mathcal{A} + s) \mathrm{d}s.$$

Setting $s = w_3 \sqrt{\frac{t}{n}}$, we have $t = w_5 n s^2$ and $\mathbf{E}\{\mathcal{Z}\} \le \mathcal{A} + \int_0^\infty e^{-w_5 n s^2} \mathrm{d}s \le \mathcal{A} + \frac{w}{\sqrt{n}}$. Hence,

$$\mathbf{E}\{\mathrm{KL}_h(f \| f_{k,n})\} - \mathrm{KL}_h(f \| \mathcal{C}) \le \frac{c_1}{k+2} + \frac{c_2}{\sqrt{n}} \int_0^c \log^{1/2} N(\mathcal{P}, \varepsilon/2, \| \cdot \|_\infty) \mathrm{d}\varepsilon + \frac{c_3}{\sqrt{n}},$$

where $c_1$, $c_2$, and $c_3$ are constants that depend on some or all of $a$, $b$, and $c$. $\qquad \square$

**Remark 17.** *The approximation error characterises the suitability of the class $\mathcal{C}$, i.e., how well functions in $\mathcal{C}$ are able to estimate a target $f$ which does not necessarily lie in $\mathcal{C}$. The estimation error characterises the error arising from the estimation of the target $f$ on the basis of the finite sample of size $n$.*

*Proof of Corollary 6.* Let $\mathcal{X}$ and $\Theta$ be compact and assume the Lipshitz condition given in (8). If $\varphi(x; \cdot)$ is continuously differentiable, then

$$|\varphi(x; \theta) - \varphi(x; \tau)| \le \sum_{k=1}^d \left| \frac{\partial \varphi(x; \cdot)}{\partial \theta_k}(\theta_k^*) \right| |\theta_k - \tau_k| \le \sup_{\theta^* \in \Theta} \left\| \frac{\partial \varphi(x; \cdot)}{\partial \theta}(\theta^*) \right\|_1 \|\theta - \tau\|_1.$$

Setting

$$\Phi(x) = \sup_{\theta^* \in \Theta} \left\| \frac{\partial \varphi(x; \cdot)}{\partial \theta}(\theta^*) \right\|_1,$$

we have $\|\Phi\|_\infty < \infty$. From Lemma 20, we obtain the fact that

$$\log N_{[]}(\mathcal{P}, 2\varepsilon \|\Phi\|_\infty, \| \cdot \|_\infty) \le \log N(\Theta, \varepsilon, \| \cdot \|_\infty),$$

which by the change of variable $\delta = 2\varepsilon \|\Phi\|_\infty \implies \varepsilon = \delta/2\|\Phi\|_\infty$ implies

$$\log N_{[]}(\mathcal{P}, \varepsilon/2, \| \cdot \|_\infty) \le \log N\left(\Theta, \frac{\varepsilon}{4\|\Phi\|_\infty}, \| \cdot \|_1\right).$$

Since $\Theta \subset \mathbb{R}^d$, using the fact that a Euclidean set of radius $r$ has covering number $N(r, \varepsilon) \le \left(\frac{3r}{\varepsilon}\right)^d$, we have

$$\log N\left(\Theta, \frac{\varepsilon}{4\|\Phi\|_\infty}, \| \cdot \|_1\right) \le d \log\left[\frac{12\|\Phi\|_\infty \mathrm{diam}(\Theta)}{\varepsilon}\right].$$

So

$$\int_0^c \sqrt{\log N\left(\Theta, \frac{\varepsilon}{4\|\Phi\|_\infty}, \|\cdot\|_1\right)} \mathrm{d}\varepsilon \leq \int_0^c \sqrt{d\log\left[\frac{12\|\Phi\|_\infty \operatorname{diam}(\Theta)}{\varepsilon}\right]} \mathrm{d}\varepsilon,$$

and since $c < \infty$, the integral is finite, as required. $\qquad\square$

## B  Discussions and remarks regarding $h$-MLLEs

In this section, we share some commentary on the derivation of the $h$-lifted KL divergence, its advantages and drawbacks, and some thoughts on the selection of the lifting function $h$. We also discuss the suitability of the MM algorithms in contrast to other approaches (e.g., EM algorithms).

### B.1  Elementary derivation

From Equation 4, we observe that if $X$ arises from a measure with density $f$, and if we aim to approximate $f$ with a density $g \in \operatorname{co}_k(\mathcal{P})$ that minimizes the $h$-lifted KL divergence $\mathrm{KL}_h$ with respect to $f$, then we can define an approximator (referred to as the minimum $h$-lifted KL approximator) as

$$\begin{aligned}
f_k &= \underset{g \in \operatorname{co}_k(\mathcal{P})}{\arg\min} \left[\int_{\mathcal{X}} \{f+h\}\log\{f+h\}\mathrm{d}\mu - \int_{\mathcal{X}} \{f+h\}\log\{g+h\}\mathrm{d}\mu\right] \\
&= \underset{g \in \operatorname{co}_k(\mathcal{P})}{\arg\min} -\int_{\mathcal{X}} \{f+h\}\log\{g+h\}\mathrm{d}\mu = \underset{g \in \operatorname{co}_k(\mathcal{P})}{\arg\max} \int_{\mathcal{X}} \{f+h\}\log\{g+h\}\mathrm{d}\mu,
\end{aligned}$$

noting that $\int_{\mathcal{X}} \{f+h\}\log\{f+h\}\mathrm{d}\mu$ is a constant that does not depend on the argument $g$. Now, observe that

$$\int_{\mathcal{X}} f\log\{g+h\}\mathrm{d}\mu = \mathbf{E}_f \log\{g+h\} \text{ and } \int_{\mathcal{X}} f\log\{g+h\}\mathrm{d}\mu = \mathbf{E}_f \log\{g+h\},$$

since both $f$ and $h$ are densities on $\mathcal{X}$ with respect to the dominating measure $\mu$. If a sample $\mathbf{X}_n = (X_i)_{i\in[n]}$ is available, we can estimate the expectation $\mathbf{E}_f \log\{g+h\}$ by the sample average functional

$$\frac{1}{n}\sum_{i=1}^n \log\{g(X_i)+h(X_i)\},$$

resulting in the sample estimator for $f_k$:

$$f'_{k,n} = \underset{g \in \operatorname{co}_k(\mathcal{P})}{\arg\max} \left[\frac{1}{n}\sum_{i=1}^n \log\{g(X_i)+h(X_i)\} + \mathbf{E}_h \log\{g+h\}\right],$$

which serves as an alternative to Equation 5. However, the expectation $\mathbf{E}_h \log\{g+h\}$ is intractable, making the optimization problem computationally infeasible, especially when $\mathcal{X}$ is multivariate (i.e., $\mathcal{X} \subset \mathbb{R}^d$ for $d > 1$), as integral evaluations may be challenging to compute accurately. Thus, we approximate the intractable integral $\mathbf{E}_h \log\{g+h\}$ using the sample average approximation (SAA) from stochastic programming (cf. Shapiro et al., 2021, Chapter 5), yielding the Monte Carlo approximation

$$\frac{1}{n_1}\sum_{i=1}^{n_1} \log\{g(Y_i)+h(Y_i)\}$$

for a sufficiently large $n_1 \in \mathbb{N}$, where each $Y_i$ is an independent and identically distributed random variable from the measure on $\mathcal{X}$ with density $h$. This approach provides an estimator for $f_k$ of the form

$$f_{k,n,n_1} = \underset{g \in \operatorname{co}_k(\mathcal{P})}{\arg\max} \left[\frac{1}{n}\sum_{i=1}^n \log\{g(X_i)+h(X_i)\} + \frac{1}{n_1}\sum_{i=1}^{n_1} \log\{g(Y_i)+h(Y_i)\}\right],$$

which is exactly the $h$-MLLE defined by Equation (5) when we take $n_1 = n$. Notably, the additional samples $\mathbf{Y}_n = (Y_i)_{i \in [n]}$ provide no information regarding $\mathbf{E}_f \log\{g + h\}$, which is the component of the objective function coupling the estimator $g$ with the target $f$. However, it offers a feasible mechanism for approximating the otherwise intractable integral $\mathbf{E}_h \log\{g + h\}$.

By setting $n_1 = n$ for the SAA approximation of $\mathbf{E}_h \log\{g + h\}$, the convergence rate in Theorem 5 remains unaffected. Specifically, for any $t > 0$,

$$\left\| \frac{1}{n} \sum_{i=1}^{n} \log \frac{g(X_i) + h(X_i)}{f(X_i) + h(X_i)} - \mathbf{E}_f \log \frac{g + h}{f + h} \right\|_{\mathcal{C}} \leq \frac{w_1}{\sqrt{n}} \mathbf{E} \log^{1/2} N(\mathcal{P}, \varepsilon, d_{n,x}) \mathrm{d}\varepsilon + \frac{w_2}{\sqrt{n}} + w_3 \sqrt{\frac{t}{n}}$$

and

$$\left\| \frac{1}{n} \sum_{i=1}^{n} \log \frac{g(Y_i) + h(Y_i)}{f(Y_i) + h(Y_i)} - \mathbf{E}_h \log \frac{g + h}{f + h} \right\|_{\mathcal{C}} \leq \frac{w_1}{\sqrt{n}} \mathbf{E} \log^{1/2} N(\mathcal{P}, \varepsilon, d_{n,y}) \mathrm{d}\varepsilon + \frac{w_2}{\sqrt{n}} + w_3 \sqrt{\frac{t}{n}},$$

with probability at least $1 - e^{-t}$, as noted in Remark 14. Given that both upper bounds are of order $\mathcal{O}(1/\sqrt{n}) + \mathcal{O}(\sqrt{t/n})$, the combined bound in the proof of Theorem 5 in Appendix A.2 is also of this order, as required.

Finally, to obtain the additional samples $\mathbf{Y}_n = (Y_i)_{i \in [n]}$, we simply simulate $\mathbf{Y}_n$ from the data-generating process defined by $h$. Since we can choose $h$ freely, selecting an $h$ that facilitates easy simulation (e.g., $h$ uniform over $\mathcal{X}$, which remains bounded away from zero on a compact set) is advisable for satisfying the requirements of our theorems.

## B.2 Advantages and limitations

As discussed extensively in Sections 1 and 2, our two primary benchmarks are the MLE and the least $L_2$ estimator. Indeed, the MLE is simpler than the $h$-MLLE estimator, as it takes the reduced form

$$\hat{f}_{k,n} = \arg\max_{g \in \mathrm{co}_k(\mathcal{P})} \frac{1}{n} \sum_{i=1}^{n} \log g(X_i),$$

and does not require a sample average approximation for intractable integrals. It is well established that the MLE estimates the minimum KL divergence approximation to the target $f$

$$f_k = \arg\min_{g \in \mathrm{co}_k(\mathcal{P})} \int_{\mathcal{X}} f \log \frac{f}{g} \mathrm{d}\mu = \mathrm{KL}_h (f \,\|\, g).$$

However, as highlighted in the foundational works of Li & Barron (1999) and Rakhlin et al. (2005), controlling the expected risk

$$\mathbf{E} \left\{ \mathrm{KL} \left( f \,\|\, \hat{f}_{k,n} \right) \right\} - \mathrm{KL} (f \,\|\, \mathcal{C}).$$

requires that $f \geq \gamma$ for some strictly positive constant $\gamma > 0$. This requirement excludes many interesting density functions as targets, including those that vanish at the boundaries of $\mathcal{X}$, such as the $\beta(\cdot; 1/2, 1/2)$ distribution, or those that vanish in the interior of $\mathcal{X}$, such as examples $f_1$ and $f_2$ in Section 4.2. Consequently, the condition $f \geq \gamma$ is restrictive and often impractical in many data analysis settings.

Alternatively, one could consider targeting the minimum $L_2$ estimator:

$$f_k = \arg\min_{g \in \mathrm{co}_k(\mathcal{P})} \int_{\mathcal{X}} (f - g)^2 \mathrm{d}\mu = \arg\min_{g \in \mathrm{co}_k(\mathcal{P})} \int_{\mathcal{X}} f^2 - 2fg + g^2 \mathrm{d}\mu = \arg\min_{g \in \mathrm{co}_k(\mathcal{P})} \left[ -2 \int_{\mathcal{X}} fg \mathrm{d}\mu + \int_{\mathcal{X}} g^2 \mathrm{d}\mu \right].$$

Using a sample $\mathbf{X}_n$ generated from the distribution given by $f$, the first term of the objective can be approximated by $-\frac{1}{n} \sum_{i=1}^{n} g(X_i)$, which is relatively simple. However, the second term involves an intractable integral that cannot be approximated by Monte Carlo sampling from a fixed generative distribution, as it depends on the optimization argument $g$. Thus, unlike the $h$-MLLE, it is not feasible to reduce this intractable integral to a sample average, which implies the need for a numerical approximation in practice. This can be

computationally complex, particularly when $g$ is intricate and $\mathcal{X}$ is high-dimensional. Hence, the minimum $L_2$-norm estimator of the form

$$\hat{f}_{k,n} = \underset{g \in \mathrm{co}_k(\mathcal{P})}{\arg\min} \left[ -\frac{2}{n} \sum_{i=1}^{n} g(X_i) + \int_{\mathcal{X}} g^2 \mathrm{d}\mu \right]$$

is often computationally infeasible, though its risk

$$\mathbf{E} \| f - \hat{f}_{k,n} \|_2 - \inf_{g \in \mathcal{C}} \| f - g \|_2$$

can be bounded, as shown in the works of Klemelä (2007) and Klemelä (2009), even when $\min_{\mathcal{X}} f = 0$. In comparison with the minimum $L_2$ estimator and the MLE, we observe that the $h$-MLLE allows risk bounding for targets $f$ not bounded from below (i.e., $\min_{\mathcal{X}} f = 0$), without requiring intractable integral expressions. The $h$-MLLE achieves the beneficial properties of both the MLE and minimum $L_2$ estimators, which is the focus of our work.

Other divergences and risk minimization schemes for estimating $f$, such as $\beta$-likelihoods and $L_q$ likelihood, could also be considered. The $L_q$ likelihood, for instance, provides a maximizing estimator with a simple sample average expression, similar to the MLE and $h$-MLLE. However, it lacks a characterization in terms of a proper divergence function, such as the KL divergence, $h$-lifted KL divergence, or $L_2$ norm. Consequently, this estimator is often inconsistent, as observed in Ferrari & Yang (2010) and Qin & Priebe (2013). These studies show that the $L_q$ likelihood estimator may not converge meaningfully to $f$, even when $f \in \mathrm{co}_k(\mathcal{P})$, for any fixed $q \in \mathbb{R}_{>0} \setminus \{1\}$, unless a sequence of maximum $L_q$ likelihood estimators is constructed with $q$ depending on $n$ and approaching 1 to approximate the MLE. Thus, the maximum $L_q$ likelihood estimator does not yield the type of risk bound we require.

Similarly, with the $\beta$-likelihood (or density power divergence), the situation is comparable to that of the minimum $L_2$ norm estimator, where the sample-based estimator involves an intractable integral that cannot be approximated through SAA. Specifically, the minimum $\beta$-likelihood estimator is defined as (cf. Basu et al., 1998):

$$\hat{f}_{k,n} = \underset{g \in \mathrm{co}_k(\mathcal{P})}{\arg\min} \left[ -\frac{1}{n} \left( 1 + \frac{1}{\beta} \right) \sum_{i=1}^{n} g^\beta(X_i) + \int_{\mathcal{X}} g^{1+\beta} \mathrm{d}\mu \right]$$

for $\beta > 0$, which closely resembles the form of the minimum $L_2$ estimator. Hence, the limitations of the minimum $L_2$ estimator apply here as well, although a risk bound with respect to the $\beta$-likelihood divergence could theoretically be obtained if the computational challenges are disregarded. In Section 1.3, we cite additional estimators based on various divergences and modified likelihoods. Nevertheless, in each case, one of the limitations discussed here will apply.

### B.3   Selection of the lifting density function $h$

The choice of $h$ is entirely independent of the data. In fact, $h$ can be any density with respect to $\mu$, satisfying $0 < a \le h(x) \le b < \infty$ for every $x \in \mathcal{X}$. Beyond this requirement, our theoretical framework remains unaffected by the specific choice of $h$. In Section 4, we explore cases where $h$ is uniform and non-uniform, demonstrating convergence in both $k$ and $n$ that aligns with the predictions of Theorem 5. For practical implementation, as discussed in Appendix B.1, $h$ serves as the sampling distribution for the sample average approximation (SAA) of the intractable integral $\mathbf{E}_h \log\{g + h\}$. Given its role as a sampling distribution, it is advantageous to select a form for $h$ that is easy to sample from. In many applications, we find that the uniform distribution over $\mathcal{X}$ is an optimal choice for $h$, as it meets the bounding conditions.

We observe that although calibrating $h$ does not improve the rate, it does influence the constants in the upper bound of the final equation of Equation 19. Specifically, for each $t > 0$, with probability at least $1 - \mathrm{e}^{-t}$,

$$\mathrm{KL}_h \left( f \, \| \, f_{k,n} \right) - \mathrm{KL}_h \left( f \, \| \, \mathcal{C} \right) \le \frac{w_1}{\sqrt{n}} \int_0^c \log^{1/2} N(\mathcal{P}, \varepsilon/2, \| \cdot \|_\infty) \mathrm{d}\varepsilon + \frac{w_2}{\sqrt{n}} + w_3 \sqrt{\frac{t}{n}} + \frac{w_4}{k+2},$$

which contributes to the constants in Theorem 5. Letting $c$ denote the upper bound of the target $f$ (i.e., $f(x) \leq c < \infty$ for every $x \in \mathcal{X}$), we make the following observations regarding the constants:

$$w_1 \propto \frac{c+b}{a^2}, \quad w_2 \propto \frac{(8b+4a)(c+b)}{a^2}, \quad w_3 \propto \log\left(\frac{c+b}{a}\right), \quad w_4 \propto \frac{c^2}{a^2}.$$

Here, $w_1$, $w_2$, and $w_3$ are per the final bound in Equation 17 in the proof of Theorem 13, while $w_4$ arises from the bound in Equation 15.

When $h$ is uniform, it takes the form $h(x) = z$, where $z = 1/\int_{\mathcal{X}} \mathrm{d}\mu$, making $a = z = b$. If $h$ is non-uniform, then necessarily $a < z < b$, as there would exist a region $\mathcal{Z}$ of positive measure where $h(x) > z$, which implies that $h(x) < z$ for some $x \in \mathcal{X} \setminus \mathcal{Z}$; otherwise,

$$\int_{\mathcal{X}} h\mathrm{d}\mu = \int_{\mathcal{Z}} h\mathrm{d}\mu + \int_{\mathcal{X}\setminus\mathcal{Z}} h\mathrm{d}\mu > \frac{\mu(\mathcal{Z})}{\mu(\mathcal{X})} + \frac{\mu(\mathcal{X}\setminus\mathcal{Z})}{\mu(\mathcal{X})} = 1,$$

contradicting $h$ being a density function. Although we cannot control $c$, we can choose $h$ to control $a$ and $b$. Setting $h = z$ minimizes the numerators in $w_1$, as deviations from uniformity increase the numerators of $w_1$ and $w_4$ while decreasing the denominators. The same reasoning applies to $w_2$:

$$w_2 \propto \frac{(8b+4a)(c+b)}{a^2} = \left\{ \frac{8bc}{a^2} + \frac{4c}{a} + \frac{8b^2}{a^2} + \frac{4b}{a} \right\}.$$

Since $c > 0$, any deviation from uniformity in $h$ either increases or maintains the numerators while decreasing the denominators, minimizing $w_2$ when $h$ is uniform. The same logic applies to $w_3$, as the logarithmic function is increasing, so $w_3$ is minimized when $h$ is uniform.

Consequently, we conclude that the smallest constants in Theorem 5 are achieved when $h$ is chosen as the uniform distribution on $\mathcal{X}$. This suggests that a uniform $h$ is optimal from both practical and theoretical perspectives.

### B.4  Discussions regarding the sharpness of the obtained risk bound

Similar to the role of Gaussian mixtures as the archetypal class of mixture models for Euclidean spaces, beta mixtures represent the archetypal class of mixture models on the compact interval $[0, 1]$, as established in the studies by Ghosal (2001); Petrone (1999). Just as Gaussian mixtures can approximate any continuous density on $\mathcal{X} = \mathbb{R}^d$ to an arbitrary level of accuracy in the $L_p$-norm (Nguyen et al., 2020; 2021; 2022b), mixtures of beta distributions can similarly approximate any continuous density on $\mathcal{X} = [0, 1]$ with respect to the supremum norm (Ghosal, 2001; Petrone, 1999; Petrone & Wasserman, 2002). We will leverage this property in the following discussion.

Assuming the target $f$ is within the closure of our mixture class $\mathcal{C}$ (i.e., $\mathrm{KL}_h(f\,||\,\mathcal{C}) = 0$), setting $k_n = \mathcal{O}(\sqrt{n})$ achieves a convergence rate in expected $\mathrm{KL}_h$ of $\mathcal{O}(1/\sqrt{n})$ for the mixture maximum $h$-lifted likelihood estimator ($h$-MLLE) $f_{k_n,n}$. An interesting question is whether this rate is tight and not overly conservative, given the observed rates in Table 1. We aim to investigate this question by discussing a lower bound for the estimation problem.

To approach this, we use Proposition 3 to observe that $\mathrm{KL}_h$ satisfies a Pinsker-like inequality:

$$\sqrt{\mathrm{KL}_h(f\,||\,g)} \geq \mathrm{TV}(f, g),$$

where $\mathrm{TV}(f, g) = \frac{1}{2} \int_{\mathcal{X}} |f - g| \mathrm{d}\mu$. Using this inequality along with Corollary 6 and the convexity of $f \mapsto f^2$, we find that the $h$-MLLE satisfies the following total variation bound:

$$\mathbf{E}\left\{\mathrm{TV}(f, f_{k_n,n})\right\} \leq \sqrt{\frac{w_{1,f}}{k_n} + \frac{w_{2,f}}{\sqrt{n}}} \leq \frac{w_{1,f}}{k_n^{1/2}} + \frac{w_{2,f}}{n^{1/4}} \leq \frac{w_f}{n^{1/4}},$$

for some positive constants $w_{1,f}$, $w_{2,f}$, $w_f$ depending on $f$, by taking $k_n = \sqrt{n}$. Now, consider the specific case when $\mathcal{X} = [0, 1]$, and the component class $\mathcal{P}$ consists of beta distributions. In this case, we have (cf.

Petrone & Wasserman, 2002, Eq. 5), for any continuous density function $f : [0, 1] \to \mathbb{R}_{\geq 0}$:

$$\inf_{g \in \mathcal{C}} \sup_{x \in [0,1]} |f(x) - g(x)| = 0, \text{ which implies that } \inf_{g \in \mathcal{C}} \mathrm{KL}_h (f \,\|\, g) = 0,$$

since

$$\sup_{x \in [0,1]} |f(x) - g(x)| \geq L_2(f, g) \geq \sqrt{\gamma} \, \mathrm{KL}_h (f \,\|\, g),$$

for any $0 < \gamma \leq h$, with the second inequality due to Proposition 3. Thus, for a compact parameter space $\Theta$ defining $\mathcal{P}$, we assume $\mathrm{KL}_h (f \,\|\, \mathcal{C}) = 0$. Consequently, the rate of $\mathcal{O}(n^{-1/4})$ for expected total variation distance is achievable in the beta mixture model setting. This convergence is uniform in the sense that

$$\mathbf{E} \{\mathrm{TV}(f, f_{k_n, n})\} \leq \frac{w}{n^{1/4}},$$

where $w$ depends only on the maximum $c \geq f$, the diameter of $\Theta$, and the condition $\mathrm{KL}_h (f \,\|\, \mathcal{C}) = 0$, with component distributions in $\mathcal{P}$ restricted to parameter values in $\Theta$.

In the context of minimum total variation density estimation on $[0, 1]$, Exercise 15.14 of Devroye & Lugosi (2001) states that for every estimator $\hat{f}$ and every Lipschitz continuous density $f$ (with a sufficiently large Lipschitz constant),

$$\sup_{f \in \mathrm{Lip}} \mathbf{E}_f \left\{ \mathrm{TV}(\hat{f}, f) \right\} \geq \frac{W}{n^{1/3}},$$

for some universal constant $W$ depending only on the Lipschitz constant. This lower bound is faster than our achieved rate of $\mathcal{O}(n^{-1/4})$, but it applies only to the smaller class of Lipschitz targets, a subset of the continuous targets satisfying $\mathrm{KL}_h (f \,\|\, \mathcal{C}) = 0$.

The target $f_2$ from our simulations in Section 4 belongs to the class of Lipschitz targets, and thus the improved lower bound rate of $\mathcal{O}(n^{-1/3})$ from Devroye & Lugosi (2001) applies. We can compare this with $\sqrt{n^{-b_2}}$ for Experiment 2 in Table 1, yielding an empirical rate in $n$ of $\mathcal{O}(n^{-1.03})$, with an exponent between $-1.07$ and $-0.98$ (95% confidence), over the range $n \in \{2^{10}, \ldots, 2^{15}\}$. Clearly, this observed rate is faster than the lower bound rate of $\mathcal{O}(n^{-1/3})$, indicating that the faster rates observed in Table 1 are due to small values of $n$ and $k$. As $n$ increases, the rate must eventually decelerate to at least $\mathcal{O}(n^{-1/3})$ when the target $f$ is Lipschitz on $\mathcal{X}$, which is only marginally faster than our guaranteed rate of $\mathcal{O}(n^{-1/4})$. Demonstrating that $\mathcal{O}(n^{-1/4})$ is minimax optimal for certain target classes $f$ is a complex task, left for future exploration.

Lastly, we note that our discussions in this section implies that the $h$-MLLE provides an effective and genetic method for obtaining estimators with total variation guarantees, which complements the comprehensive studies on the topic presented in Devroye & Györfi (1985) and Devroye & Lugosi (2001).

### B.5 The KL divergence and the MLE

For any probability densities $f$ and $g$ with respect to a dominant measure $\mu$ on $\mathcal{X}$, the $h$-lifted KL divergence is defined as

$$\mathrm{KL}_h (f \,\|\, g) = \int_{\mathcal{X}} \{f + h\} \log \left( \frac{f + h}{g + h} \right) \mathrm{d}\mu,$$

which we establish as a Bregman divergence on the space of probability densities dominated by $\mu$ on $\mathcal{X}$ in Appendix C.1.

We previously demonstrated a relationship between $\mathrm{KL}_h$ and the $L_2$ distance (Proposition 3), showing that if $h(x) \geq \gamma > 0$ for all $x \in \mathcal{X}$, then

$$\mathrm{KL}_h (f \,\|\, g) \leq \frac{1}{\gamma} L_2^2(f, g), \text{ where } L_2^2(f, g) = \|f - g\|_2^2 = \int_{\mathcal{X}} (f - g)^2 \mathrm{d}\mu$$

is the square of the $L_2$ distance between the densities. Given that we can always select $h(x) \geq \gamma$, this bound is always enforceable. This relationship is stronger than that between the standard KL divergence and the $L_2$ distance, which similarly satisfies

$$\mathrm{KL} (f \,\|\, g) \leq \frac{1}{\gamma} L_2^2(f, g),$$

but with the more restrictive requirement that $f(x) \geq \gamma > 0$ for every $x \in \mathcal{X}$, limiting its applicability to densities that do not vanish. In the proof of Proposition 3 in Appendix C.2, we show that one can write

$$\mathrm{KL}_h \left( f \,\|\, g \right) = 2\mathrm{KL} \left( \frac{f + h}{2}, \frac{g + h}{2} \right),$$

which allows the application of the theory from Rakhlin et al. (2005) by considering the mixture density $(f + h)/2$ as the target and using $g + h$ as the approximand, where $g \in \mathrm{co}_k(\mathcal{P})$. Under this framework, the maximum likelihood estimator can be formulated as

$$f_{k,n} \in \underset{g \in \mathrm{co}_k(\mathcal{P})}{\arg\min} \ -\frac{1}{n} \sum_{i=1}^n \log \left( \frac{g(Z_i) + h(Z_i)}{2} \right),$$

where $(Z_i)_{i \in [n]}$ are independent and identically distributed samples from a distribution with density $(f+h)/2$. This sampling can be performed by choosing $X_i$ with probability $1/2$ or $Y_i$ with probability $1/2$ for each $i \in [n]$, where $X_i$ is an observation from the generative model $f$ and $Y_i$ is an independent sample from the auxiliary density $h$. Although the modified estimator, based on the bound from Rakhlin et al. (2005), attains equivalent convergence rates, it inefficiently utilizes observed data, as 50% of the data is replaced by simulated samples $Y_i$. In contrast, our $h$-MLLE estimator maximally utilizes all available data while achieving the same bound.

### B.6 Comparison of the MM algorithm and the EM algorithm

Since the risk functional is not a log-likelihood, a straightforward EM approach cannot be used to compute the $h$-MLLE. However, by interpreting $\mathrm{KL}_h$ as a loss between the target mixture $(f+h)/2$ and the estimator $(f_{k,n} + h)/2$, an EM algorithm can be constructed using the standard admixture framework (see Lange, 2013, Section 9.5). Remarkably, the EM algorithm for estimating $(f_{k,n} + h)/2$, has the same form as our MM algorithm, which leverages Jensen's inequality (cf. Lange, 2013, Section 8.3). In fact, the majorizer in any EM algorithm results directly from Jensen's inequality (see Lange, 2013, Section 9.2), making our MM algorithm in Section 4.1 no more complex than an EM approach for mixture models.

Beyond the EM and MM methods, no other standard algorithms typically address the generic estimation of a $k$-component mixture model in $\mathrm{co}_k(\mathcal{P})$ for a given parametric class $\mathcal{P}$. Since our MM algorithm follows a structure nearly identical to the EM algorithm for the MLE of this problem, it has comparable iterative complexity. Notably, per iteration, the MM approach requires additional evaluations for both $\mathbf{X}_n$ and $\mathbf{Y}_n$, and for $g(X_i)$ and $h(X_i)$, so it requires a constant multiple of evaluations compared to EM, depending on whether $h$ is a uniform distribution or otherwise (typically by a factor of 2 or 4).

### B.7 Non-convex optimization

We note that the $h$-MLLE problem (and the corresponding MLE) are non-convex optimization problems. This implies that, aside from global optimization methods, no iterative algorithm–whether gradient-based methods like gradient descent, coordinate descent, mirror descent, or momentum-based variants–can be guaranteed to find a global optimum. Likewise, second-order techniques such as Newton and quasi-Newton methods also cannot be expected to locate the global solution. In non-convex scenarios, the primary assurance that can be offered is convergence to a critical point of the objective function. In our case, this assurance is achieved by applying Corollary 1 from Razaviyayn et al. (2013), as discussed in Section 4.1. Notably, this convergence guarantee is consistent with that provided by other iterative approaches, such as EM, gradient descent, or Newton's method.

Additionally, it may be valuable to examine whether specific convergence rates can be ensured when the algorithm's iterates approach a neighborhood around a critical value. In the context of the MM algorithm, we can affirmatively answer this question: since the $h$-MLLE objective is twice continuously differentiable with respect to the parameter $\psi_k$, it satisfies the local convergence conditions outlined in Lange (2016, Proposition 7.2.2). This result implies that if $\psi_k^{(s)}$ lies within a sufficiently close neighborhood of a local minimizer $\psi_k^*$, the MM algorithm converges linearly to $\psi_k^*$. This behavior aligns with the convergence guarantees offered

by other iterative methods, such as gradient descent or line-search based quasi-Newton methods. Quadratic convergence rates near $\psi_k^*$ can be achieved with a Newton method, though this forfeits the monotonicity (or stability) of the MM algorithm, as it is well-known that even in convex settings, Newton's method can diverge if the initialization is not properly handled.

An additional advantage of the MM algorithm over Newton's method is its capacity to decompose the original objective into a sum of functions where each component of $\psi_k = (\pi_1, \ldots, \pi_k, \theta_1, \ldots, \theta_k)$ is separable within the summation. In other words, we can independently optimize functions that depend only on subsets of parameters, either $(\pi_1, \ldots, \pi_k)$ or each $\theta_j$ for $j = 1, \ldots, k$, thereby simplifying the iterative computation. This characteristic is noted after Equation 9 in the main text. Such decomposition can lead to computational efficiency by avoiding the need to compute the Hessian matrix for Newton's method or approximations required by quasi-Newton methods. Specifically, in cases involving mixtures of exponential family distributions such as the beta distributions discussed in Section 4.2, each parameter-separated problem becomes a strictly concave maximization problem, which can be efficiently solved (see Proposition 3.10 in Sundberg, 2019).

## C   Auxiliary proofs

In this section, we include other proofs of claims made in the main text that are not included in Appendix A.

### C.1   The $h$-lifted KL divergence as a Bregman divergence

Let $\tilde{u} = u + h$, so that $\phi(u) = \tilde{u}\log(\tilde{u}) - \tilde{u} + 1$. Then $\phi'(u) = \log(\tilde{u})$, and

$$
\begin{aligned}
D_\phi(f \,\|\, g) &= \int_X \{\tilde{f}\log(\tilde{f}) - \tilde{f} - 1\} - \{\tilde{g}\log(\tilde{g}) - \tilde{g} - 1\} - \log(\tilde{g})(f - g)\mathrm{d}\mu \\
&= \int_\mathcal{X} \tilde{f}\log(\tilde{f}) - \tilde{g}\log(\tilde{g}) - f\log(\tilde{g}) + g\log(\tilde{g})\mathrm{d}\mu \\
&= \int_\mathcal{X} \{f + h\}\log(\tilde{f}) - \{g + h\}\log(\tilde{g}) - f\log(\tilde{g}) + g\log(\tilde{g})\mathrm{d}\mu \\
&= \int_\mathcal{X} \{f + h\}\log\frac{f + h}{g + h}\mathrm{d}\mu = \mathrm{KL}_h\left(f \,\|\, g\right).
\end{aligned}
$$

### C.2   Proof of Proposition 2

Let $\tilde{f} = f + h$ and $\tilde{g} = g + h$. Since $h$ is positive, there exists some $\tilde{g}_*$ such that $\tilde{g}_* = \inf_{x \in \mathcal{X}}\{g(x) + h(x)\} > 0$. Similarly, since $\mathcal{X}$ is compact, there exists some positive $\tilde{f}^*$ such that $0 < \tilde{f}^* = \sup_{x \in \mathcal{X}}\{f(x) + h(x)\} < \infty$. Define $M = \sup_{x \in \mathcal{X}}\log\{\tilde{f}(x)/\tilde{g}(x)\}$. Then $M < \infty$, and

$$
\mathrm{KL}_h\left(f \,\|\, g\right) = \int_\mathcal{X} \tilde{f}\log\frac{\tilde{f}}{\tilde{g}}\mathrm{d}\mu \leq \sup_{x \in \mathcal{X}}\log\frac{\tilde{f}}{\tilde{g}}\int_\mathcal{X}\tilde{f}\mathrm{d}\mu = 2M < \infty.
$$

### C.3   Proof of Proposition 3

Defining $\tilde{f}$ and $\tilde{g}$ as above, we have

$$
\mathrm{KL}_h\left(f \,\|\, g\right) = \int_\mathcal{X} \tilde{f}\log\frac{\tilde{f}}{\tilde{g}}\mathrm{d}\mu \leq \int_\mathcal{X}\tilde{f}\left(\frac{\tilde{f}}{\tilde{g}} - 1\right)\mathrm{d}\mu = \int_\mathcal{X}\frac{(f - g)^2}{\tilde{g}}\mathrm{d}\mu \leq \gamma^{-1}L_2^2(f, g),
$$

The first inequality comes from the fundamental inequalities on logarithm $\log(x) \leq x - 1$ for all $x \geq 0$. Indeed, let $f(x) = \log(x) - x + 1$. We obtain $f'(x) = \frac{1}{x} - 1 = \frac{1-x}{x}$. Then $f'(x) < 0$ if $x > 1$ and $f'(x) \geq 0$ if $x \leq 1$. Therefore, $f$ is strictly decreasing on $(1, \infty)$ and $f$ is strictly increasing on $(0, 1]$. This leads to the desired inequality $f(x) \leq f(1) = 0$ for all $x \geq 0$.

The next equality comes from the following identities:

$$\int_{\mathcal{X}} \tilde{f}(\frac{\tilde{f}}{\tilde{g}} - 1)\mathrm{d}\mu = \int_{\mathcal{X}} \frac{\tilde{f}^2 - \tilde{f}\tilde{g}}{\tilde{g}}\mathrm{d}\mu = \int_{\mathcal{X}} \frac{\tilde{f}^2 - \tilde{f}\tilde{g} - \tilde{f}\tilde{g} + \tilde{g}\tilde{g}}{\tilde{g}}\mathrm{d}\mu = \int_{\mathcal{X}} \frac{(\tilde{f} - \tilde{g})^2}{\tilde{g}}\mathrm{d}\mu = \int_{\mathcal{X}} \frac{(f - g)^2}{\tilde{g}}\mathrm{d}\mu.$$

The last equality is followed from

$$\int_{\mathcal{X}} \frac{-\tilde{f}\tilde{g} + \tilde{g}\tilde{g}}{\tilde{g}}\mathrm{d}\mu = -\int_{\mathcal{X}} \tilde{f}\mathrm{d}\mu + \int_{\mathcal{X}} \tilde{g}\mathrm{d}\mu = -\int_{\mathcal{X}}(f + h)\mathrm{d}\mu + \int_{\mathcal{X}}(g + h)\mathrm{d}\mu = -\int_{\mathcal{X}} h\mathrm{d}\mu + \int_{\mathcal{X}} h\mathrm{d}\mu = 0.$$

In fact, the proof of Proposition 3 follows the standard technique in the derivation of the estimation error, see for example Meir & Zeevi (1997).

Additionally, we can show that the $h$-lifted KL divergence satisfies a Pinsker-like inequality, in the sense that

$$\sqrt{\mathrm{KL}_h(f \,\|\, g)} \geq \mathrm{TV}(f, g),$$

where TV represents the total variation distance between the densities $f$ and $g$. Indeed, this is easy to observe since

$$\mathrm{KL}_h(f \,\|\, g) = \int_{\mathcal{X}} \{f + h\} \log \frac{f + h}{g + h}\mathrm{d}\mu = 2\int \frac{f + h}{2} \log \frac{\left\{\frac{f+h}{2}\right\}}{\left\{\frac{g+h}{2}\right\}}\mathrm{d}\mu = 2\mathrm{KL}\left(\frac{f + h}{2}, \frac{g + h}{2}\right)$$

$$\geq 4\left\{\frac{1}{2}\int_{\mathcal{X}}\left|\frac{f + h}{2} - \frac{g + h}{2}\right|\mathrm{d}\mu\right\}^2 = \left\{\int_{\mathcal{X}}\left|\frac{f + h}{2} - \frac{g + h}{2}\right|\mathrm{d}\mu\right\}^2 = \left\{\frac{1}{2}\int_{\mathcal{X}}|f - g|\,\mathrm{d}\mu\right\}^2 = \mathrm{TV}^2(f, g),$$

where the inequality is due to Pinsker's inequality:

$$\sqrt{\frac{1}{2}\mathrm{KL}(f \,\|\, g)} \geq \mathrm{TV}(f, g).$$

### C.4 Proof of Proposition 9

For choice (11), by the dominated convergence theorem, we observe that

$$\frac{\mathrm{d}^2}{\mathrm{d}\pi^2}\kappa((1 - \pi)p + \pi q) = \mathbf{E}_f\left\{\frac{\mathrm{d}^2}{\mathrm{d}\pi^2}\log\frac{f + h}{(1 - \pi)p + \pi q + h}\right\} + \mathbf{E}_h\left\{\frac{\mathrm{d}^2}{\mathrm{d}\pi^2}\log\frac{f + h}{(1 - \pi)p + \pi q + h}\right\}$$

$$= \mathbf{E}_f\left\{\frac{(p - q)^2}{[(1 - \pi)p + \pi q + h]^2}\right\} + \mathbf{E}_h\left\{\frac{(p - q)^2}{[(1 - \pi)p + \pi q + h]^2}\right\}.$$

Suppose that each $\varphi(\cdot; \theta) \in \mathcal{P}$ is bounded from above by $c < \infty$. Then, since $p, q \in \mathcal{C}$ are non-negative functions, we have the fact that $(p - q)^2 \leq c^2$. If we further have $a \leq h$ for some $a > 0$, then $[(1 - \pi)p + \pi q + h]^2 \geq a^2$, which implies that

$$\frac{\mathrm{d}^2}{\mathrm{d}\pi^2}\kappa((1 - \pi)p + \pi q) \leq 2 \times \frac{c^2}{a^2}$$

for every $p, q \in \mathcal{C}$ and $\pi \in (0, 1)$, and thus

$$\sup_{p, q \in \mathcal{C}, \pi \in (0, 1)} \frac{\mathrm{d}^2}{\mathrm{d}\pi^2}\kappa((1 - \pi)p + \pi q) \leq \frac{2c^2}{a^2} < \infty.$$

Similarly, for case (12), we have

$$\frac{\mathrm{d}^2}{\mathrm{d}\pi^2}\kappa_n((1 - \pi)p + \pi q) = \frac{1}{n}\sum_{i=1}^{n}\frac{\mathrm{d}^2}{\mathrm{d}\pi^2}\log\frac{f(x_i) + h(x_i)}{(1 - \pi)p(x_i) + \pi q(x_i) + h(x_i)}$$

$$+ \frac{1}{n} \sum_{i=1}^{n} \frac{\mathrm{d}^2}{\mathrm{d}\pi^2} \log \frac{f(y_i) + h(y_i)}{(1-\pi) p(y_i) + \pi q(y_i) + h(y_i)}$$

$$= \frac{1}{n} \sum_{i=1}^{n} \frac{(p(x_i) - q(x_i))^2}{[(1-\pi) p(x_i) + \pi q(x_i) + h(x_i)]^2} + \frac{1}{n} \sum_{i=1}^{n} \frac{(p(y_i) - q(y_i))^2}{[(1-\pi) p(y_i) + \pi q(y_i) + h(y_i)]^2}.$$

By the same argument, as for $\kappa$, we have the fact that $(p(x) - q(x))^2 \le c^2$, for every $p, q \in \mathcal{C}$ and every $x \in \mathcal{X}$, and furthermore $[(1-\pi) p(x) + \pi q(x) + h(x)]^2 \ge a^2$, for any $\pi \in (0,1)$. Thus,

$$\sup_{p,q \in \mathcal{C}, \pi \in (0,1)} \frac{\mathrm{d}^2}{\mathrm{d}\pi^2} \kappa((1-\pi) p + \pi q) \le \frac{2c^2}{a^2} < \infty, \text{ as required.}$$

## D    Technical results

Here we collect some technical results that are required in our proofs but appear elsewhere in the literature. In some places, notation may be modified from the original text to keep with the established conventions herein.

**Lemma 18** (Kosorok, 2007. Lem 9.18). *Let $N(\mathscr{F}, \varepsilon, \|\cdot\|)$ denote the $\varepsilon$-covering number of $\mathscr{F}$, $N_{[]}(\mathscr{F}, \varepsilon, \|\cdot\|)$ the $\varepsilon$-bracketing number of $\mathscr{F}$, and $\|\cdot\|$ be any norm on $\mathscr{F}$. Then, for all $\varepsilon > 0$*

$$N(\mathscr{F}, \varepsilon, \|\cdot\|) \le N_{[]}(\mathscr{F}, \varepsilon, \|\cdot\|)$$

**Lemma 19** (Kosorok, 2007. Lem 9.22). *For any norm $\|\cdot\|$ dominated by $\|\cdot\|_\infty$ and any class of functions $\mathscr{F}$,*

$$\log N_{[]}(\mathscr{F}, 2\varepsilon, \|\cdot\|) \le \log N(\mathscr{F}, \varepsilon, \|\cdot\|_\infty), \text{ for all } \varepsilon > 0.$$

**Lemma 20** (Kosorok, 2007. Thm 9.23). *For some metric $d$ on $T$, let $\mathscr{F} = \{f_t : t \in T\}$ be a function class:*

$$|f_s(x) - f_t(x)| \le d(s,t) F(x),$$

*some fixed function $F$ on $\mathcal{X}$, and for all $x \in \mathcal{X}$ and $s, t \in T$. Then, for any norm $\|\cdot\|$,*

$$N_{[]}(\mathscr{F}, 2\varepsilon\|F\|, \|\cdot\|) \le N(T, \varepsilon, d).$$

**Lemma 21** (Shalev-Shwartz & Ben-David (2014), Lem 26.7). *Let $A$ be a subset of $\mathbb{R}^m$ and let*

$$A' = \left\{ \sum_{j=1}^{n} \alpha_j \mathbf{a}_j \mid n \in \mathbb{N}, \mathbf{a}_j \in A, \alpha_j \ge 0, \|\alpha\|_1 = 1 \right\}.$$

*Then, $\mathcal{R}_n(A') = \mathcal{R}_n(A)$, i.e., both $A$ and $A'$ have the same Rademacher complexity.*

**Lemma 22** (van de Geer, 2016, Thm. 16.2). *Let $(X_i)_{i \in [n]}$ be non-random elements of $\mathcal{X}$ and let $\mathscr{F}$ be a class of real-valued functions on $\mathcal{X}$. If $\varphi_i : \mathbb{R} \to \mathbb{R}$, $i \in [n]$, are functions vanishing at zero that satisfy for all $u, v \in \mathbb{R}$, $|\varphi_i(u) - \varphi_i(v)| \le |u - v|$, then we have*

$$\mathbf{E}\left\{ \left\| \sum_{i=1}^{n} \varphi_i(f(X_i)) \varepsilon_i \right\|_{\mathscr{F}} \right\} \le 2\mathbf{E}\left\{ \left\| \sum_{i=1}^{n} f(X_i) \varepsilon_i \right\|_{\mathscr{F}} \right\}.$$

**Lemma 23** (McDiarmid, 1998, Thm. 3.1 or McDiarmid, 1989). *Suppose $(X_i)_{i \in [n]}$ are independent random variables and let $Z = g(X_1, \ldots, X_n)$, for some function $g$. If $g$ satisfies the bounded difference condition, that is there exists constant $c_j$ such that for all $j \in [n]$ and all $x_1, \ldots, x_j, x'_j, \ldots, x_n$,*

$$|g(x_1, \ldots, x_{j-1}, x_j, x_{j+1}, \ldots, x_n) - g(x_1, \ldots, x_{j-1}, x'_j, x_{j+1}, \ldots, x_n)| \le c_j,$$

*then*

$$\mathbf{P}(Z - \mathbf{E}Z \ge t) \le \exp\left\{ \frac{-2t^2}{\sum_{j=1}^{n} c_j^2} \right\}.$$

**Lemma 24** (van der Vaart & Wellner, 1996, Lem. 2.3.1). *Let $\mathfrak{R}(f) = \mathbf{E}f$ and $\mathfrak{R}_n(f) = n^{-1}\sum_{i=1}^{n} f(X_i)$. If $\Phi: \mathbb{R}_{>0} \to \mathbb{R}_{>0}$ is a convex function, then the following inequality holds for any class of measurable functions $\mathscr{F}$:*

$$\mathbf{E}\Phi\left(\|\mathfrak{R}(f) - \mathfrak{R}_n(f)\|_{\mathscr{F}}\right) \leq \mathbf{E}\Phi\left(2\|R_n(f)\|_{\mathscr{F}}\right),$$

*where $R_n(f)$ is the Rademacher process indexed by $\mathscr{F}$. In particular, since the identity map is convex,*

$$\mathbf{E}\left\{\|\mathfrak{R}(f) - \mathfrak{R}_n(f)\|_{\mathscr{F}}\right\} \leq 2\mathbf{E}\left\{\|R_n(f)\|_{\mathscr{F}}\right\}.$$

**Lemma 25** (Koltchinskii, 2011, Thm. 3.11). *Let $d_n$ be the empirical distance*

$$d_n^2(f_1, f_2) = \frac{1}{n}\sum_{i=1}^{n}(f_1(X_i) - f_2(X_i))^2$$

*and denote by $N(\mathscr{F}, \varepsilon, d_n)$ the $\varepsilon$-covering number of $\mathscr{F}$. Let $\sigma_n^2 \coloneqq \sup_{f \in \mathscr{F}} P_n f^2$. Then the following inequality holds*

$$\mathbf{E}\left\{\|R_n(f)\|_{\mathscr{F}}\right\} \leq \frac{K}{\sqrt{n}}\mathbf{E}\int_0^{2\sigma_n} \log^{1/2} N(\mathscr{F}, \varepsilon, d_n)\mathrm{d}\varepsilon$$

*for some constant $K > 0$.*

