# OpenReview forum: "Risk Bounds for Mixture Density Estimation on Compact Domains via the h-Lifted Kullback–Leibler Divergence"
_TMLR — Accepted by TMLR_

### Review · Reviewer_u7g9 · 2024-07-20

**Summary Of Contributions:**

The authors introduce the h-lifted Kullback–Leibler (KL) divergence as a criterion for conducting risk minimization and prove an $O(1/\sqrt{n})$ bound on the expected estimation error under a compact support assumption when using this divergence.

**Audience:**

Yes

**Claims And Evidence:**

No

**Requested Changes:**

1. In Eq.(4), why this $Y_i$ variable is needed in the estimator? If it's just a random noise independent of $X_i$, it can't bring any extra information.  In a practical scenario, how to define it or obtain it?
2. In Sec. 1.3, the authors mentioned some other likelihood estimators, what are the pros and cons of the proposed estimator compared to those existing ones?
3. How to define the density function $h$ for a given dataset? Is it data-independent?
4. Proposition 3. The proof doesn't look correct. How was the first inequality obtained? And $\tilde{f}(\frac{\tilde{f}}{\tilde{g}}-1) = (f+h)\frac{f-g}{\tilde{g}}$, so where is the $(f-g)^2$ from?
5. What's $t$ in Theorem 11?
6. Are those theoretical results also valid in multivariate cases, such as $X_i$ is multidimensional? If yes, how does the data dimension affect the error rate?
7. A few practical experiments would be more convincing.
8. It could be helpful to explain how each theoretical result (e.g. a proposition, lemma, or remark) contributes to the main findings or why such a result is attractive.

**Strengths And Weaknesses:**

The authors propose a new extension of KL divergence to obtain a more sample-efficient estimator and provide a theoretical analysis of this divergence.
The paper is a bit hard to read through as there are many mathematical proofs included in the formal contents and there are no clear connections between those propositions, lemmas, and theorems. It is also not clear how those theoretical results can bring some benefits in practical scenarios. The experiments are too simplified to demonstrate any realistic applications.
As I'm not an expert in measure theory or a closely related field, I can not make a judgment about the novelty of this work. I'm not able to verify all the proofs either.
In my understanding, this work is more about statistics than machine learning and its contribution to the community of machine learning is dubious. Detailed questions are below.

---

> ### Author Response · Authors · 2024-08-13
> **Rebuttal by Authors**
>
> *Weaknesses: The paper is a bit hard to read through as there are many mathematical proofs included in the formal contents and there are no clear connections between those propositions, lemmas, and theorems.*
>
> *Requested Changes: 8. It could be helpful to explain how each theoretical result (e.g. a proposition, lemma, or remark) contributes to the main findings or why such a result is attractive.*
>
> We thank the Reviewer for the comment. The approach taken was that of a more journal-oriented style, as opposed to a conference-oriented style, with the provision of proof sketches in the main body of the text. When revising, we are happy to move all technical proofs to the appendix to improve the readability of the manuscript. Notably, we merge Section 3. Preliminaries and Section 5. Proofs and then move on to the Appendix. Furthermore, we would like to add a summary of the technical results at the end of the introduction. For example, Proposition 2 presents the advantages that the h-lifted KL divergence is bounded for all continuous densities in contrast to the standard KL divergence, while maintaining the good relationship between the KL divergence and the $L_2$ norm distance as in Proposition 3.
>
> *Weaknesses: In my understanding, this work is more about statistics than machine learning and its contribution to the community of machine learning is dubious.*
>
> We appreciate that the Reviewer's assessment that this is quite a statistical, and theory dense paper, whose contributions may be technical and abstract in nature. Nonetheless, we would like to point out that "TMLR emphasizes technical correctness over subjective significance, to ensure that we facilitate scientific discourse on topics that may not yet be accepted in mainstream venues but may be important in the future." We take this to mean that TMLR, or more generally other top machine learning conferences and journals, have also been a venue for highly statistical texts whose contributions are technical in nature. A quick search of the most recent top machine learning conferences and journals leads us to the following texts whose style of presentations and technical contributions are very similar to ours: [1,2,3,4,5,6,7,8,9,10,11,12,13]. In fact, it is the presence of these texts that motivated us to consider TMLR as the most appropriate venue for this work, since these articles align closely with our interests as researchers in Statistical Learning, within the discipline of Machine Learning. We feel that our risk bounds for mixture density estimation on compact domains via the $h$-Lifted Kullback–Leibler divergence is a very useful paradigm across many statistical learning scenarios, and thus, to bring it to the attention of the community who we think will most make use of its results, we believe that TMLR is in fact the most appropriate venue for our text. We thus believe that our work is well within the scope of the journal's audience.
>
> References
>
> [1] Ho, N., Yang, C. Y., & Jordan, M. I. (2022). Convergence rates for Gaussian mixtures of experts. Journal of Machine Learning Research, 23(323), 1-81.
>
> [2] Nguyen, H., Nguyen, T., & Ho, N. (2024). Demystifying softmax gating function in Gaussian mixture of experts. Advances in Neural Information Processing Systems, 36.
>
> [3] Varici, B., Katz, D., Wei, D., Sattigeri, P., & Tajer, A. (2024). Separability Analysis for Causal Discovery in Mixture of DAGs. Transactions on Machine Learning Research.
>
> [4] Nguyen, H., Nguyen, T., Nguyen, K., & Ho, N. (2024, April). Towards convergence rates for parameter estimation in Gaussian-gated mixture of experts. In International Conference on Artificial Intelligence and Statistics (pp. 2683-2691). PMLR.
>
> [5] Tai, W. M., & Aragam, B. (2023, July). Learning mixtures of Gaussians with censored data. In International Conference on Machine Learning (pp. 33396-33415). PMLR.
>
> [6] Diakonikolas, I., Kane, D., & Sun, Y. (2023). SQ lower bounds for learning mixtures of linear classifiers. Advances in Neural Information Processing Systems, 36, 27281-27293.
>
> [7] Fang, Z., & Cheng, G. (2023). Optimal Convergence Rates of Deep Convolutional Neural Networks: Additive Ridge Functions. Transactions on Machine Learning Research.
>
> [8] Arbas, J., Ashtiani, H., & Liaw, C. (2023, July). Polynomial time and private learning of unbounded Gaussian mixture models. In International Conference on Machine Learning (pp. 1018-1040). PMLR.
>
> [9] Guha, A., Ho, N., & Nguyen, X. (2023, July). On excess mass behavior in Gaussian mixture models with Orlicz-Wasserstein distances. In International Conference on Machine Learning (pp. 11847-11870). PMLR.
>
> [10] Nguyen, H., Akbarian, P., Yan, F., & Ho, N. Statistical Perspective of Top-K Sparse Softmax Gating Mixture of Experts. In The Twelfth International Conference on Learning Representations.

---

> > ### Author Response · Authors · 2024-08-13
> > **Rebuttal by Authors**
> >
> > [11] Manole, T., & Ho, N. (2022, June). Refined convergence rates for maximum likelihood estimation under finite mixture models. In International Conference on Machine Learning (pp. 14979-15006). PMLR.
> >
> > [12] Polaczyk, B., & Cyranka, J. (2022). Improved overparametrization bounds for global convergence of sgd for shallow neural networks. Transactions on Machine Learning Research.
> >
> > [13] Gassiat, É., & Corff, S. L. (2023). Variational excess risk bound for general state space models. Transactions on Machine Learning Research.
> >
> > *Requested Changes: 1. In Eq.(4), why this $Y_i$ variable is needed in the estimator? If it's just a random noise independent of $X_i$, it can't bring any extra information. In a practical scenario, how to define it or obtain it?*
> >
> > Thank you for this excellent question. From Equation (3), we notice
> > that if $X$ arises from a measure with density $f$, which we wish
> > to approximate $f$ via some density $g\in\text{co}\_{k}\left(\mathscr{P}\right)$
> > which minimizes the $h$-lifted KL divergence $\text{KL}\_{h}$ to
> > $f$, then we may define such an approximator (which we shall call
> > the minimum $h$-lifted KL approximator) as
> > \begin{align*}
> > f\_{k} & =\underset{g\in\text{co}\_{k}\left(\mathscr{P}\right)}{\arg\min}\left[\int\_{\mathscr{X}}\left\\{ f+h\right\\} \log\left\\{ f+h\right\\} \text{d}\mu-\int\_{\mathscr{X}}\left\\{ f+h\right\\} \log\left\\{ g+h\right\\} \text{d}\mu\right]\\
> >  & =\underset{g\in\text{co}\_{k}\left(\mathscr{P}\right)}{\arg\min}-\int\_{\mathscr{X}}\left\\{ f+h\right\\} \log\left\\{ g+h\right\\} \text{d}\mu\\
> >  & =\underset{g\in\text{co}\_{k}\left(\mathscr{P}\right)}{\arg\max}\int\_{\mathscr{X}}\left\\{ f+h\right\\} \log\left\\{ g+h\right\\} \text{d}\mu\text{,}
> > \end{align*}
> > by noting that $\int\_{\mathscr{X}}\left\\{ f+h\right\\} \log\left\\{\ f+h\right\\} \text{d}\mu$
> > is a constant that does not depend on the argument $g$. Now, notice
> > that
> > $$
> > \int\_{\mathscr{X}}f\log\left\\{ g+h\right\\} \text{d}\mu=\mathbf{E}\_{f}\log\left\\{ p+h\right\\}
> > $$
> > and
> > $$
> > \int\_{\mathscr{X}}h\log\left\\{ g+h\right\\} \text{d}\mu=\mathbf{E}\_{h}\log\left\\{ p+h\right\\} \text{,}
> > $$
> > since both $f$ and $h$ are density functions on $\mathscr{X}$ with
> > respect to the dominating measure $\mu$.
> >
> > If we have access to some sample $\mathbf{X}\_{n}=\left(X\_{i}\right)\_{i\in\left[n\right]}$,
> > then we may estimate the expectation $\mathbf{E}\_{f}\log\left\\{ g+h\right\\} $
> > by the sample average functional
> > $$
> > \frac{1}{n}\sum_{i=1}^{n}\log\left\\{ g\left(X_{i}\right)+h\left(X_{i}\right)\right\\} \text{,}
> > $$
> > leading to the sample estimator for $f_{k}$:
> > $$
> > f\_{k,n}^{\prime}=\underset{g\in\text{co}\_{k}\left(\mathscr{P}\right)}{\arg\max}\left[\frac{1}{n}\sum\_{i=1}^{n}\log\left\\{ g\left(X\_{i}\right)+h\left(X\_{i}\right)\right\\} +\mathbf{E}\_{h}\log\left\\{ g+h\right\\} \right]\text{,}
> > $$
> > which serves as an alternative to Equation (4). However, The expectation
> > $\mathbf{E}\_{h}\log\left\\{ p+h\right\\} $ in the expression is intractable,
> > making the optimization problem computationally infeasible, especially
> > when $\mathscr{X}$ is multivariate (i.e., $\mathscr{X}\subset\mathbb{R}^{d}$,
> > for $d>1$), since evaluations of the integral may be difficult to
> > accurately compute. Thus, using the principle of sample average approximation
> > (SAA) form stochastic programming (cf. [1] Chapter 5), whereupon
> > we approximate the intractable integral $\mathbf{E}\_{h}\log\left\\{ p+h\right\\} $
> > by its Monte Carlo approximation
> > $$
> > \frac{1}{n\_{1}}\sum\_{i=1}^{n\_{1}}\log\left\\{ p\left(Y\_{i}\right)+h\left(Y\_{i}\right)\right\\}
> > $$
> > for some sufficiently large $n_{1}\in\mathbb{N}$, where each $Y_{i}$
> > is an independent and identically distributed random variable from
> > the measure on $\mathscr{X}$ with density $h$. Thus, for any choice
> > $n_{1}$, we obtain an estimator for $f_{k}$ of the form
> > $$
> > f\_{k,n,n\_{1}}=\underset{g\in\text{co}\_{k}\left(\mathscr{P}\right)}{\arg\max}\left[\frac{1}{n}\sum\_{i=1}^{n}\log\left\\{ g\left(X\_{i}\right)+h\left(X\_{i}\right)\right\\} +\frac{1}{n\_{1}}\sum\_{i=1}^{n\_{1}}\log\left\\{ g\left(Y\_{i}\right)+h\left(Y\_{i}\right)\right\\} \right]
> > $$
> > which is equivalent to the maximum $h$-lifted likelihood estimator
> > ($h$-MLLE) defined by Equation (4) when we take $n\_{1}=n$. Indeed
> > the additional samples $\mathbf{Y}\_{n}=\left(Y\_{i}\right)\_{i\in\left[n\right]}$
> > provides no information regarding $\mathbf{E}\_{f}\log\left\\{ g+h\right\\} $,
> > which is the component of the object function which couples the estimator
> > $g$ with the target $f$. However, it provides a feasible mechanism
> > for computing the otherwise intractable integral $\mathbf{E}\_{h}\log\left\\{ g+h\right\\} $,
> > which cannot be ignored.

---

> > > ### Author Response · Authors · 2024-08-13
> > > **Rebuttal by Authors**
> > >
> > > We note that by choosing $n\_{1}=n$ for the SAA approximation of $\mathbf{E}\_{h}\log\left\\{ p+h\right\\} $,
> > > we do not compromise the rate of convergence in Theorem 9. This is
> > > because, for every $t>0$,
> > > $$
> > > \left\Vert \frac{1}{n}\sum\_{i=1}^{n}\log\frac{g\left(X\_{i}\right)+h\left(X\_{i}\right)}{f\left(X\_{i}\right)+h\left(X\_{i}\right)}-{\mathbf{E}}\_{f}\log\frac{g+h}{f+h}\right\Vert \_{\mathscr{C}}\le\frac{w\_{1}}{\sqrt{n}}\mathbf{E}\log^{1/2}N\left(\mathscr{P},\epsilon,d_{n,x}\right)\text{d}\epsilon+\frac{w\_{2}}{\sqrt{n}}+w\_{3}\sqrt{\frac{t}{n}}
> > > $$
> > > and
> > > $$
> > > \left\Vert \frac{1}{n}\sum\_{i=1}^{n}\log\frac{g\left(Y\_{i}\right)+h\left(Y\_{i}\right)}{f\left(Y\_{i}\right)+h\left(Y\_{i}\right)}-{\mathbf{E}}\_{h}\log\frac{g+h}{f+h}\right\Vert \_{\mathscr{C}}\le\frac{w_{1}}{\sqrt{n}}\mathbf{E}\log^{1/2}N\left(\mathscr{P},\epsilon,d_{n,y}\right)\text{d}\epsilon+\frac{w\_{2}}{\sqrt{n}}+w\_{3}\sqrt{\frac{t}{n}}\text{,}
> > > $$
> > > with probabilities at least $1-\text{e}^{-t}$, which we noted in
> > > Remark 12. Since the upper bounds are both of $O\left(1/\sqrt{n}\right)+O\left(\sqrt{t/n}\right)$,
> > > when we combine the bounds in the Proof of Theorem 9, the total bound
> > > is also of the same order $O\left(1/\sqrt{n}\right)+O\left(\sqrt{t/n}\right)$,
> > > as required.
> > >
> > > To answer the final part of the question, we simply simulate $\mathbf{Y}\_{n}$
> > > in practice, from the data generating process defined by $h$. Given
> > > that we are free to choose $h$, one should choose an $h$ such that
> > > the simulation of $\mathbf{Y}\_{n}$ is easy, such as $h$ uniform
> > > over $\mathscr{X}$, which is always bounded away from zero on a compact
> > > set, as required for our theorems.
> > >
> > > *Reference*
> > >
> > > [1] Shapiro, A., Dentcheva, D., \& Ruszczynski, A. (2021). Lectures
> > > on stochastic programming: modeling and theory. Society for Industrial
> > > and Applied Mathematics.
> > >
> > > *2. In Sec. 1.3, the authors mentioned some other likelihood estimators, what are the pros and cons of the proposed estimator compared to those existing ones?*
> > >
> > > We thank the Reviewer for this excellent question. Certainly, for
> > > any estimation problem, there are limitless numbers of potential risk
> > > functionals that could be considered, each with potential pros and
> > > cons. As we discuss at length in Sections 1 and 2, our two primary
> > > points of comparison are the maximum likelihood estimator (MLE) and
> > > least $L\_{2}$ estimator. In the case of the MLE, indeed the estimator
> > > is simpler than our $h$-MLLE estimator, since it has the reduced
> > > form
> > > $$
> > > \hat{f}\_{k,n}=\underset{g\in\text{co}\_{k}\left(\mathscr{P}\right)}{\arg\max}\frac{1}{n}\sum\_{i=1}^{n}\log g\left(X\_{i}\right)\text{,}
> > > $$
> > > and does not introduce the need for a sample average approximation
> > > of some intractable integral. It is well known that the MLE is an
> > > estimator of the minimum Kullback--Leibler divergence approximator
> > > to the target $f$:
> > > $$
> > > f\_{k}=\underset{g\in\text{co}\_{k}\left(\mathscr{P}\right)}{\arg\min}\int\_{\mathscr{X}}f\log\left\\{ f/g\right\\} \text{d}\mu=\text{KL}\left(f\Vert g\right)\text{.}
> > > $$
> > > As pointed out in the classic works of [1] and [2], we cannot
> > > control the expected risk
> > > $$
> > > \mathbf{E}\left\\{ \text{KL}\left(f\Vert\hat{f}\_{k,n}\right)\right\\} -\text{KL}\left(f\Vert\mathscr{C}\right)
> > > $$
> > > without requiring that $f\ge\gamma$ for some strictly positive constant
> > > $\gamma>0$. This requirement excludes vast families of interesting
> > > density functions as targets, including any that vanish at the boundary
> > > of the space $\mathscr{X}$, for example the $\text{Beta}\left(2,2\right)$
> > > distribution, or any that vanish in the interior of $\mathscr{X}$,
> > > as per our examples in Section 6.2: $f_{1}$ and $f\_{2}$. Thus $f\ge\gamma$
> > > is a very restrictive assumption that cannot be assumed to be satisfied
> > > in practice for many data analytic settings.
> > >
> > > As an alternative, one may consider targeting the minimum $L\_{2}$
> > > estimator:
> > > \begin{align*}
> > > f\_{k} & =\underset{g\in\text{co}\_{k}\left(\mathscr{P}\right)}{\arg\min}\int\_{\mathscr{X}}\left(f-g\right)^{2}\text{d}\mu\\
> > >  & =\underset{g\in\text{co}\_{k}\left(\mathscr{P}\right)}{\arg\min}\int\_{\mathscr{X}}f^{2}-2fg+g^{2}\text{d}\mu\\
> > >  & =\underset{g\in\text{co}\_{k}\left(\mathscr{P}\right)}{\arg\min}\left[-\int\_{\mathscr{X}}fg\text{d}\mu+\int\_{\mathscr{X}}g^{2}\text{d}\mu\right]\text{.}
> > > \end{align*}
> > > Using a sample $\mathbf{X}\_{n}$ generated from distribution given
> > > by $f$, we can certainly approximate the first term of the optimization
> > > objective by
> > > $$
> > > -\frac{1}{n}\sum\_{i=1}^{n}g\left(X\_{i}\right)\text{,}
> > > $$
> > > which is indeed simple. However, the second term, an intractable integral,
> > > cannot be Monte Carlo estimated by a sample from a fixed generative
> > > distribution, since it is dependent on our optimization argument $g$.
> > > Thus, we cannot reduce this intractable integral to a sample average
> > > like for the $h$-MLLE, which implies that one requires a numerical
> > > approximation to this integral in practice, which may be computationally
> > > highly complex when $g$ is complicated and when the domain $\mathscr{X}$
> > > is high dimensional.

---

> > > > ### Author Response · Authors · 2024-08-13
> > > > **Rebuttal by Authors**
> > > >
> > > > Thus, the estimator minimum $L_{2}$-norm estimator
> > > > of form
> > > > $$
> > > > \hat{f}\_{k,n}=\underset{g\in\text{co}\_{k}\left(\mathscr{P}\right)}{\arg\min}\left[-\frac{2}{n}\sum\_{i=1}^{n}g\left(X\_{i}\right)+\int\_{\mathscr{X}}g^{2}\text{d}\mu\right]
> > > > $$
> > > > is too computationally infeasible in practice, although one can easily
> > > > bound the risk of
> > > > $$
> > > > \mathbf{E}\left\Vert f-\hat{f}_{k,n}\right\Vert \_{2}-\inf\_{g\in\mathscr{C}}\left\Vert f-g\right\Vert \_{2}
> > > > $$
> > > > as described in the works of [3] and [4], even when $\min\_{\mathscr{X}}f=0$.
> > > > Thus when compared to the minimum $L\_{2}$ estimator and the MLE,
> > > > we can see that the $h$-MLLE allows for bounding of risk for targets
> > > > $f$ that are not necessarily bounded from below (i.e. $\min\_{\mathscr{X}}f=0$),
> > > > whilst also not requiring any intractable and infeasible integral
> > > > expression. The fact that the $h$-MLLE satisfies the two positive
> > > > properties of the MLE and minimum $L\_{2}$ estimators is the entire
> > > > purpose of our work.
> > > >
> > > > Indeed, one can further consider other divergence and hence other
> > > > risk minimization schemes for estimating $f$, such as the so called
> > > > $\beta$-likelihoods and $L_{q}$ likelihood, among many other potential
> > > > estimators that we cannot hope to exhaustively discuss. In the case
> > > > of the $L_{q}$ likelihood, the maximizing estimator has a simple
> > > > sample average expression like the MLE and the $h$-MLLE. However,
> > > > the estimator does not characterize the minimization with respect
> > > > to a proper divergence function like the KL divergence, $h$-lifted
> > > > KL divergence, or $L_{2}$ norm between densities. As such, the maximizing
> > > > estimator is typically inconsistent, in that even if $f\in\text{co}\_{k}\left(\mathscr{P}\right)$,
> > > > it may happen that for any choice of $q\in\mathbb{R}\_{>0}\backslash\left\\{ 1\right\\} $,
> > > > $\hat{f}\_{k,n}$ obtained by maximizing the $L\_{q}$ likelihood does
> > > > not converge in any meaningful way to $f$. This is observed in the
> > > > works of [5] and [6], where the authors consider a sequence
> > > > of minimum $L\_{q}$ estimators where $q$ depends on $n$ and converges
> > > > to $1$ as to recover the MLE. Thus, the maximum $L\_{q}$ likelihood estimator does not permit a risk bound of the type that we seek.
> > > >
> > > > For the $\beta$-likelihood (also referred to as the density power
> > > > divergence) the situation is akin to that of the minimum $L\_{2}$
> > > > norm estimator, where the sample based estimator must contain an intractable
> > > > integral that cannot be approximated via SAA. In particular, the minimum
> > > > $\beta$-likelihood estimator is characterized as (cf. [7]):
> > > > $$
> > > > \hat{f}\_{k,n}=\underset{g\in\text{co}\_{k}\left(\mathscr{P}\right)}{\arg\min}\left[-\frac{1}{n}\left(1+\frac{1}{\beta}\right)\sum\_{i=1}^{n}g^{\beta}\left(X\_{i}\right)+\int\_{\mathscr{X}}g^{1+\beta}\text{d}\mu\right]
> > > > $$
> > > > for $\beta>0$, which we can see strongly resembles that of the minimum
> > > > $L\_{2}$ estimator. Thus, our discussions regarding the shortcomings
> > > > of the minimum $L\_{2}$ estimator applies here as well, although we
> > > > note that in principle, if one ignores the computational infeasibility,
> > > > it is possible to obtain a risk bound with respect the corresponding
> > > > divergence of the $\beta$-likelihood estimator. In Section 1.3, we
> > > > cite many other potential estimators based on divergences and modified
> > > > likelihoods. However, we note that in each case, one of the shortcomings
> > > > that we have discussed will apply.
> > > >
> > > > References
> > > >
> > > > [1] Jonathan Li and Andrew Barron. Mixture Density Estimation.
> > > > In Advances in Neural Information
> > > > Processing Systems, volume 12. MIT Press, 1999
> > > >
> > > > [2] Alexander Rakhlin, Dmitry Panchenko, and Sayan Mukherjee.
> > > > Risk bounds for mixture density estimation. ESAIM: PS, 9:220--229,
> > > > 2005.
> > > >
> > > > [3] Jussi S Klemela. Density estimation with stagewise optimization
> > > > of the empirical risk. Machine Learning, 67:169--195, 2007.
> > > >
> > > > [4] Jussi S Klemela. Smoothing of multivariate data: density estimation
> > > > and visualization. John Wiley \& Sons, 2009.
> > > >
> > > > [5] Ferrari, D., \& Yang, Y. (2010). Maximum lq-likelihood method.
> > > > Annals of Statistics, 38, 573-583.
> > > >
> > > > [6] Qin, Y., \& Priebe, C. E. (2013). Maximum L q-Likelihood estimation
> > > > via the Expectation-Maximization algorithm: a robust estimation of
> > > > mixture models. Journal of the American Statistical Association, 108(503),
> > > > 914-928.
> > > >
> > > > [7] Basu, A., Harris, I. R., Hjort, N. L., \& Jones, M. C. (1998).
> > > > Robust and efficient estimation by minimising a density power divergence.
> > > > Biometrika, 85(3), 549-559.

---

> > > > ### Author Response · Authors · 2024-08-13
> > > > **Rebuttal by Authors**
> > > >
> > > > *3. How to define the density function $\mathbf{h}$ for a given dataset? Is it data-independent?*
> > > >
> > > > The choice of $h$ is entirely independent of the data. Indeed $h$
> > > > can be chosen to be any density with respect to $\mu$, such that
> > > > $0< a\le h\left(x\right)\le b<\infty$, for every $x\in\mathscr{X}$.
> > > > Beyond this consideration, our theory is independent of the choice
> > > > of $h$. For practical purposes, we recall the response to Question
> > > > 1, where we note that the role that $h$ plays is as the sampling
> > > > distribution for the SAA of the intractable integral $\mathbf{E}\_{h}\log\left\\{ g+h\right\\} $.
> > > > Since it is used as a sampling distribution, one should consider making
> > > > the distribution characterized by $h$ easy to sample from. We believe
> > > > that in many situations, the best choice for $h$ is to take the uniform
> > > > distribution over $\mathscr{X}$, which obviously satisfies the bounding
> > > > requirements. Such a choice also has the added bonus that for any
> > > > $x\in\mathscr{X}$, $h\left(x\right)=z$ for some constant $z$, such
> > > > that $z=1/\int_{\mathscr{X}}\text{d}\mu$, and thus the $h$-lifted
> > > > likelihood terms $\log\left\\{ g\left(X_{i}\right)+h\left(X_{i}\right)\right\\} $
> > > > and $\log\left\\{ g\left(Y\_{i}\right)+h\left(Y\_{i}\right)\right\\} $
> > > > can be computed more computationally efficiently. In light of the
> > > > constants with respect to $k$ provided by Proposition 6, we also
> > > > note that one should seek to minimize the constant $c^{2}/a^{2}$,
> > > > where $c$ is the upper bound of $f$. Since one cannot control $c$,
> > > > one would wish to make $a$ as large as possible, uniformly over $\mathscr{X}$.
> > > > Again, this is achieved by the uniform distribution on $\mathscr{X}$
> > > > since
> > > > if $h$ is not uniform then there must exist some region $\mathscr{Z}$
> > > > of positive measure such that $h\left(x\right)>z$, but on this region
> > > > $\int_{\mathscr{Z}}h\text{d}\mu>\mu\left(\mathscr{Z}\right)/\mu\left(\mathscr{X}\right)$,
> > > > which would imply that $h\left(x\right)<z$ for some $x\in\mathscr{X}\backslash\mathscr{Z}$,
> > > > else
> > > > $$
> > > > \int_{\mathscr{X}}h\text{d}\mu=\int_{\mathscr{Z}}h\text{d}\mu+\int_{\mathscr{X}\backslash\mathscr{Z}}h\text{d}\mu>\frac{\mu\left(\mathscr{Z}\right)}{\mu\left(\mathscr{X}\right)}+\frac{\mu\left(\mathscr{X}\backslash\mathscr{Z}\right)}{\mu\left(\mathscr{X}\right)}=1\text{,}
> > > > $$
> > > > hence making $h$ not a density function.
> > > > We note that the same argument holds when we consider the constants
> > > > with respect to $n$ that appear in the proof of Theorem 11. For instance,
> > > > the constant with respect to $t/\sqrt{n}$ is
> > > > $$
> > > > w_{3}=\sqrt{2}\log\left(\frac{c+b}{a}\right)\text{.}
> > > > $$
> > > > When $h$ is uniform, $a=b=z$, yields the constant
> > > > $$
> > > > w_{3}=\sqrt{2}\log\left(\frac{c+z}{z}\right)\text{.}
> > > > $$
> > > > If we deviate from $h$ being uniform, then necessarily we will get
> > > > $b>z$ and $a< z$, which implies that
> > > > $$
> > > > \sqrt{2}\log\left(\frac{c+z}{z}\right)<\sqrt{2}\log\left(\frac{c+b}{a}\right)
> > > > $$
> > > > since $w_{3}$ is increasing in $b$ but decreasing in $a$.
> > > >
> > > > In Section 6, we experiment with a case where $h$ is uniform and
> > > > one where it is not, and show that in both situations, one obtains
> > > > convergence in both $k$ and $n$ at a sufficiently fast rate that
> > > > corroborate with the predictions of Theorem 9.
> > > >
> > > > *4. Proposition 3. The proof doesn't look correct. How was the first inequality obtained? And $\tilde{f}(\frac{\tilde{f}}{\tilde{g}}-1) = (f+h)\frac{f-g}{\tilde{g}}$, so where is the $(f-g)^2$from?*
> > > >
> > > > The first inequality comes from the fundamental inequalities on logarithm: $\log(x) \le x-1$ for all $x\ge 0$. Indeed, let $f(x) = \log(x) - x +1$. We obtain $f'(x) = \frac{1}{x} -1 = \frac{1-x}{x}$. Then $f'(x)<0$ if $x>1$ and $f'(x) \ge 0$ if $x \le 1$. Therefore, $f$ is strictly decreasing on $(1, ∞)$ and $f$ is strictly increasing on $(0,1]$. This leads to the desired inequality $f(x) \le f(1) = 0$ for all $x \ge 0$.
> > > >
> > > > The next equality comes from the following identities:
> > > > $$\int_{\mathscr{X}}\tilde{f}(\frac{\tilde{f}}{\tilde{g}}-1) d \mu= \int_{\mathscr{X}}\frac{\tilde{f}^2-\tilde{f}\tilde{g}}{\tilde{g}} d \mu = \int_{\mathscr{X}}\frac{\tilde{f}^2-\tilde{f}\tilde{g} -\tilde{f}\tilde{g}+\tilde{g}\tilde{g}}{\tilde{g}} d \mu = \int_{\mathscr{X}}\frac{(\tilde{f} -\tilde{g})^2}{\tilde{g}} d \mu = \int_{\mathscr{X}}\frac{(f -g)^2}{\tilde{g}} d \mu.$$
> > > > The last equality is followed from
> > > > $$\int_{\mathscr{X}}\frac{-\tilde{f}\tilde{g}+\tilde{g}\tilde{g}}{\tilde{g}} d \mu = -\int_{\mathscr{X}} \tilde{f} d \mu + \int_{\mathscr{X}} \tilde{g} d \mu = -\int_{\mathscr{X}} (f+h) d \mu + \int_{\mathscr{X}} (g+h) d \mu = -\int_{\mathscr{X}} h d \mu + \int_{\mathscr{X}} h d \mu = 0.$$
> > > > In fact, the proof of Proposition 3 follows the standard technique in the derivation of the estimation error, see for example [1].

---

> > > > > ### Author Response · Authors · 2024-08-13
> > > > > **Rebuttal by Authors**
> > > > >
> > > > > We further note that whilst addressing the comments of Reviewer Wawz, we had discovered
> > > > > an additional relationship of the $h$-lifted Kullback--Leibler divergence
> > > > > with traditional measures of divergence. Namely, in our response to
> > > > > Reviewer Wawz, we demonstrate that the $h$-lifted KL divergence satisfies
> > > > > a Pinsker-like inequality in the sense that
> > > > > $$
> > > > > \sqrt{\text{KL}_{h}\left(f\Vert g\right)}\ge\text{TV}\left(f,g\right)\text{,}
> > > > > $$
> > > > > Where $\text{TV}$ denotes the total variation distance between the
> > > > > two densities $f$ and $g$.
> > > > >
> > > > > *References*
> > > > >
> > > > > [1] Zeevi, A. J., & Meir, R. (1997). Density estimation through convex combinations of densities: approximation and estimation bounds. Neural Networks, 10(1), 99-109.
> > > > >
> > > > > *5. What's  𝐭  in Theorem 11?*
> > > > >
> > > > > We thank the Reviewer for identifying this typo. Here, $t>0$ determines
> > > > > the upper bounding probability $1-\text{e}^{-t}$ for the event
> > > > > $$
> > > > > \left\Vert \frac{1}{n}\sum\_{i=1}^{n}\log\frac{g\left(X\_{i}\right)+h\left(X\_{i}\right)}{f\left(X\_{i}\right)+h\left(X\_{i}\right)}-\mathbf{E}\_{f}\log\frac{g+h}{f+h}\right\Vert \_{\mathscr{C}}\le\frac{w\_{1}}{\sqrt{n}}\mathbf{E}\log^{1/2}N\left(\mathscr{P},\epsilon,d\_{n,x}\right)\text{d}\epsilon+\frac{w\_{2}}{\sqrt{n}}+w\_{3}\sqrt{\frac{t}{n}}\text{.}
> > > > > $$
> > > > > That is to say, the second sentence of Theorem 11 should read: ``Then,
> > > > > for each $t>0$, with probability at least $1-\text{e}^{-t}$, $\dots$''
> > > > >
> > > > > *6. Are those theoretical results also valid in multivariate cases, such as $\mathbf{X}\_i$ is multidimensional? If yes, how does the data dimension affect the error rate?*
> > > > >
> > > > > Indeed our theoretical results are valid for any compact metric space
> > > > > $\mathscr{X}$, with the space $\left[0,1\right]$ being archetypal
> > > > > and providing a simple and tractable example for us to demonstrate
> > > > > the key features of our theory. There is no problem in considering
> > > > > $\mathscr{X}=\left[-m,m\right]^{d}$ for $m>0$ and $p\in\mathbb{N}$,
> > > > > or abstractly $\mathscr{X}\subset\mathbb{R}^{d}$ compact. We alluded
> > > > > to this fact in the second paragraph of our conclusion where we suggested
> > > > > that our methods are applicable for mixtures of Dirichlet distributions,
> > > > > and mixture models on Euclidean manifolds of $\mathbb{R}^{d}$, such
> > > > > as the unit sphere, as in the case of mixtures of Kent and von Mises--Fisher
> > > > > distributions. Furthermore, we may even consider $\mathscr{X}$ to
> > > > > be functional compact spaces, although verifying that such a space
> > > > > is compact and then defining suitable component classes $\mathscr{P}$
> > > > > over such spaces that yield small approximation errors $\text{KL}\_{h}\left(f\Vert\mathscr{C}\right)$
> > > > > is a nontrivial exercise in approximation theory.
> > > > >
> > > > > From the proof of Theorem 9, we observe that the dimensionality of
> > > > > $\mathscr{X}$ only enters into our bound via the complexity of the
> > > > > class $\mathscr{P}$, i.e., the constant $\int\_{0}^{c}\log^{1/2}N\left(\mathscr{P},\epsilon/2,\left\Vert \cdot\right\Vert \_{\infty}\right)\text{d}\epsilon$,
> > > > > which is independent of both $n$ and $k$. In fact, the constant
> > > > > with respect to $k$ ($u\_{1}$ in Theorem 9) is entirely unaffected
> > > > > by the dimensionality of $\mathscr{X}$. Thus, the rates of our bound
> > > > > on the expected $h$-lifted KL divergence are dimension independent,
> > > > > and in fact valid even when the space $\mathscr{X}$ is infinite dimensional,
> > > > > provided that some class $\mathscr{P}$ can be defined so that $\int\_{0}^{c}\log^{1/2}N\left(\mathscr{P},\epsilon/2,\left\Vert \cdot\right\Vert \_{\infty}\right)\text{d}\epsilon$
> > > > > is finite. Corollary 10 demonstrates a method under which such a bound
> > > > > can be obtained when elements of $\mathscr{P}$ satisfy a Lipschitz
> > > > > condition.

---

> > > > > > ### Author Response · Authors · 2024-08-13
> > > > > > **Rebuttal by Authors**
> > > > > >
> > > > > > *7. A few practical experiments would be more convincing.*
> > > > > >
> > > > > > We thank the Reviewer for this suggestion. We believe that the contribution
> > > > > > of our work is theoretical, whereupon we describe our novel $h$-MLLE
> > > > > > estimator as a sample risk minimizer of the novel $h$-lifted Kullback--Leibler
> > > > > > divergence, and prove important properties of these objects in the
> > > > > > context of mixture model estimation. In particular, in Theorem 9,
> > > > > > we demonstrate that one can bound the expected $h$-lifted Kullback--Leibler
> > > > > > divergence between any target distribution $f$ and the $h$-MLLE
> > > > > > of the $k$ component density approximation class $\text{co}_{k}\left(\mathscr{P}\right)$,
> > > > > > where the expected divergence minus its minimum with respect to the
> > > > > > closure class $\mathscr{C}$ decreases at order $O\left(1/k\right)$
> > > > > > and $O\left(1/\sqrt{n}\right)$ with respect to the component number
> > > > > > and sample size of the data, respectively. We further develop some
> > > > > > interesting tools and techniques for analyzing these objects, including
> > > > > > a feasible minorization--majorization algorithm for computing the
> > > > > > $h$-MLLE of a $k$ component density. We assess both the performance
> > > > > > of our MM algorithm and our risk bound in Theorem 9 as part of the
> > > > > > outcome of our simulations in Section 6.3. These simulations demonstrate
> > > > > > all that we wanted to demonstrate, which is that our results faithfully
> > > > > > bound the error of the expected $h$-lifted KL divergence risk, in
> > > > > > the sense that the risk decreases at rate at least $O\left(1/k\right)$
> > > > > > in terms of $k$, and rate $O\left(1/\sqrt{n}\right)$, in terms of
> > > > > > $n$. Furthermore, these bounds are faithful for both targets that
> > > > > > are continuous and discontinuous on $\mathscr{X}$, and for different
> > > > > > choices of $h$ satisfying our requirements.
> > > > > >
> > > > > > Since our numerical experiments exactly match the kind of practical density estimation scenario that we anticipate for our methodology, we see our simulations as a complete
> > > > > > validation of what we wanted to numerically show and thus do not feel
> > > > > > that there is much utility in further simulations. For instance, the
> > > > > > question of whether the mixture of beta distributions that we assessed
> > > > > > are good density estimators on the unit interval is not interesting
> > > > > > as the approximation capability has been established in works such
> > > > > > as [1], [2] and [3], for instance.
> > > > > >
> > > > > > We see our work as a continuation of the works of [4] and [5],
> > > > > > who each provide analogous risk bound constructions in the context
> > > > > > of the MLE, and who like us purely concentrate on the theoretical
> > > > > > bounds and statements. In these works, the authors do not provide
> > > > > > any numerical verification, thus we make note that we have already
> > > > > > gone beyond the usual expectation in the literature by providing our
> > > > > > numerical experiments.
> > > > > >
> > > > > > References
> > > > > >
> > > > > > [1] Ghosal, S. (2001). Convergence rates for density estimation
> > > > > > with Bernstein polynomials. The Annals of Statistics, 29(5), 1264-1280.
> > > > > >
> > > > > > [2] Petrone, S., \& Wasserman, L. (2002). Consistency of Bernstein
> > > > > > polynomial posteriors. Journal of the Royal Statistical Society Series
> > > > > > B: Statistical Methodology, 64(1), 79-100.
> > > > > >
> > > > > > [3] Petrone, S. (1999). Random Bernstein polynomials. Scandinavian
> > > > > > Journal of Statistics, 26(3), 373-393.
> > > > > >
> > > > > > [4] Li, J., & Barron, A. (1999). Mixture density estimation. Advances in neural information processing systems, 12.
> > > > > >
> > > > > > [5] Rakhlin, A., Panchenko, D., & Mukherjee, S. (2005). Risk bounds for mixture density estimation. ESAIM: Probability and Statistics, 9, 220-229.

---

### Review · Reviewer_Wawz · 2024-07-21

**Summary Of Contributions:**

This paper presents a new (theoretical) risk bound for risk density estimation using a so-called $h$-lifted Kullback-Leibler divergence.
The main contribution is the proof that the new bound based on this divergence is correct.
The literature on previous bounds, using different divergences, seems complete and updated.
The authors present some computational results to show that the proposed $h$-lifted Kullback-Leibler divergence can be computed in practice and used to estimate the density of function given a finite sample of observed data.

**Disclaimer:** This paper falls out of my main research interests. Hence, I might have misunderstood some points.

**Audience:**

Yes

**Broader Impact Concerns:**

This is mainly a theoretical result without any direct ethical implications.

**Claims And Evidence:**

Yes

**Requested Changes:**

Since this paper is out of my main research competences, I rely on the requests of the other reviews.

In the previous paragraphs, I have suggested some possible computational test, but they are not mandatory for acceptance.

**Strengths And Weaknesses:**

**Strengths:** The strength of this paper is the theoretical result on the risk bound, which complements previous results known in the literature.
We like the link with the more general class of Bregman divergences.

**Weaknesses:** Even if the main contribution is theoretical, we find that the computational tests rather weak.
We would have expected some empirical studies on mixtures of known densities (e.g., a mixture of 3,4,5 Gaussians) with an increasing number of sample points and then empirical studies of how this theoretical bound meets practice.
This type of test (by considering a mixture of other known probability densities) could help establish whether the risk bound is tight or not.

---

> ### Author Response · Authors · 2024-08-13
> **Rebuttal by Authors**
>
> We thank the Reviewer for the nice assessment and breakdown of our
> contributions. Indeed, we contribute a largely theoretical result
> regarding the new maximum $h$-lifted likelihood estimator, and study
> its associated Bregman divergence, the $h$-lifted Kullback--Leibler
> (KL) divergence.
>
> The Reviewer suggested that we should conduct a study regarding estimation
> via known densities, e.g., a fixed sequence of mixture models, and
> in particular, a fixed sequence of Gaussian distributions. In fact,
> the Gaussian scenario falls outside of our paper's scope, since we
> are concerned entirely with the bounding the density estimation risk
> (namely, the expected $h$-lifted KL divergence; $\text{KL}_{h}$)
> for target densities whose support is a compact set. The Gaussian
> distributions are supported on unbounded sets and are thus outside
> the scope of our work.
>
> Similar to the status of the mixture of Gaussians is the archetypical
> class of mixture models on Euclidean spaces, the archetypical class
> of mixture models on the compact interval $\left[0,1\right]$ is the
> mixture of beta distributions, as per the studies of [1], [2],
> [3]. This is because, like the fact that the mixture of Gaussians
> can approximate arbitrarily closely any continuous density on $\mathscr{X}=\mathbb{R}^{d}$
> with respect to an $L_{p}$-norm, the mixture of beta distributions
> can arbitrarily well approximate any continuous density on $\mathscr{X}=\left[0,1\right]$,
> with respect to the supremum norm. We shall demonstrate a use of this
> fact in the sequel.
>
> Thus, the experiments that we conducted in Section 6.2 are exactly
> the suggested experiments that the Reviewer has asked for. Namely,
> we assess the quality of the expected $h$-lifted KL divergence as
> sample size increases ($n$ varies from $2^{10}$ to $2^{15}$), and
> as the number of beta distribution components increases ($k$ varies
> from $2$ to $8$). We made exactly the practical assessments that
> the Reviewer has asked for. Namely, we found that across the range
> of values we assessed in terms of $n$ and $k$, and across our two
> simulation scenarios, the rate of convergence with respect to both
> $n$ and $k$ are faster than our theoretical rates of $1/k$, and
> $1/\sqrt{n}$ (cf. Table 1). We argue that there is an elbow phenomenon,
> as described by [4], whereupon there are rapid decreases in loss
> when complexity ($k$) of models are small, which will slow down as
> the loss approaches its limit when models become more complex.
>
> Suppose that the target $f$ is in the closure of our mixture class
> $\mathscr{C}$ (i.e., $\text{KL}\_{h}\left(f\Vert\mathscr{C}\right)=0$).
> Then, by setting $k\_{n}=O\left(\sqrt{n}\right)$, we obtain a rate
> of convergence in expected $\text{KL}\_{h}$ of rate $O\left(1/\sqrt{n}\right)$
> for the mixture maximum $h$-lifted likelihood estimator ($h$-MLLE)
> $f\_{k\_{n},n}$. The Reviewer asks whether we can establish that our
> rate is tight and is in fact not overly conservative given the observed
> rates in Table 1. We seek to address this problem theoretically by
> discussing a lower bound for our estimation problem.
>
> To do so we make a novel observation that $\text{KL}\_{h}$ satisfies
> a Pinsker inequality in the sense that
> $$
> \sqrt{\text{KL}\_{h}\left(f\Vert g\right)}\ge\text{TV}\left(f,g\right)\text{,}
> $$
> where $\text{TV}\left(f,g\right)=\left(1/2\right)\int\_{\mathscr{X}}\left|f-g\right|\text{d}\mu$.
> Indeed, this is easy to observe since
> \begin{align*}
> \text{KL}\_{h}\left(f\Vert g\right) & =\int\_{\mathscr{X}}\left\\{ f+h\right\\} \log\frac{f+h}{g+h}\text{d}\mu\\
>  & =2\int\frac{f+h}{2}\log\frac{\left\\{ \frac{f+h}{2}\right\\} }{\left\\{ \frac{g+h}{2}\right\\} }\text{d}\mu\\
>  & =2\text{KL}\left(\frac{f+h}{2},\frac{g+h}{2}\right)\\
> & \ge4\left\\{ \left(1/2\right)\int\_{\mathscr{X}}\left|\frac{f+h}{2}-\frac{g+h}{2}\right|\text{d}\mu\right\\} ^{2}\\
>  & =\left\\{ \int\_{\mathscr{X}}\left|\frac{f+h}{2}-\frac{g+h}{2}\right|\text{d}\mu\right\\} ^{2}\\
>  & =\left\\{ \frac{1}{2}\int\_{\mathscr{X}}\left|f-g\right|\text{d}\mu\right\\} ^{2}\\
>  & =\text{TV}^{2}\left(f,g\right)\text{,}
> \end{align*}
> where the inequality is due to Pinsker's inequality:
> $$
> \sqrt{\frac{1}{2}\text{KL}\left(f\Vert g\right)}\ge\text{TV}\left(f,g\right)\text{.}
> $$
> Using this inequality, together with Corollary 10 and the fact that $f\mapsto f^2$ is convex, we have the fact
> that the $h$-MLLE satisfies the total variation bound:
> \begin{align*}
> \mathbf{E}\left\\{ \text{TV}\left(f,f_{k_{n},n}\right)\right\\}  & \le\sqrt{\frac{w\_{1,f}}{k\_{n}}+\frac{w\_{2,f}}{\sqrt{n}}}\le\frac{w\_{1,f}}{k\_{n}^{1/2}}+\frac{w\_{2,f}}{n^{1/4}}
>   \le\frac{w\_{f}}{n^{1/4}}\text{,}
> \end{align*}
> with positive constants depending on $f$, and taking $k\_{n}=\sqrt{n}$.

---

> > ### Author Response · Authors · 2024-08-13
> > **Rebuttal by Authors**
> >
> > Let us now specialize to the case when $\mathscr{X}=\left[0,1\right]$,
> > and lets consider that we take the component class $\mathscr{P}$
> > to be the beta distributions. In such a case, we have the fact that
> > (cf. [2], Eq. 5), for any continuous density function $f:\left[0,1\right]\to\mathbb{R}\_{\ge0}$:
> > $$
> > \inf\_{g\in\mathscr{C}}\sup\_{x\in\left[0,1\right]}\left|f\left(x\right)-g\left(x\right)\right|=0\text{,}
> > $$
> > which implies that
> > $$
> > \inf\_{g\in\mathscr{C}}\text{KL}\_{h}\left(f\Vert g\right)=0\text{,}
> > $$
> > since we have
> > $$
> > \sup\_{x\in\left[0,1\right]}\left|f\left(x\right)-g\left(x\right)\right|\ge L\_{2}\left(f,g\right)\ge\sqrt{\gamma}\text{KL}\_{h}\left(f\Vert g\right)\text{,}
> > $$
> > for any $0<\gamma\le h$, where second inequality is due to Proposition 3. Thus, let us assume that for some compact parameter space $\Theta$
> > determining $\mathscr{P}$, $\text{KL}\_{h}\left(f\Vert\mathscr{C}\right)=0$.
> > Thus, the $O\left(n^{-1/4}\right)$ rate for the expected total variation
> > distance is achieved in the beta mixture model setting. In fact, this
> > convergence is uniform in the sense that
> > $$
> > \mathbf{E}\left\\{ \text{TV}\left(f,f\_{k\_{n},n}\right)\right\\} \le\frac{w}{n^{1/4}}\text{,}
> > $$
> > for some $w$ depending only on the maximum $c\ge f$, the diameter
> > of $\Theta$, and the condition that $\text{KL}\_{h}\left(f\Vert\mathscr{C}\right)=0$,
> > where the component distributions in $\mathscr{P}$ are restricted
> > to taking parameter values in $\Theta$.
> >
> > In the setting of minimum total variation density estimation on $\left[0,1\right]$,
> > Exercise 15.14 of [5] states that for every estimator $\hat{f}$
> > and every Lipschitz continuous density $f$ (with sufficiently large
> > Lipschitz constant),
> > $$
> > \sup\_{f\in\mathrm{Lip}}\text{E}\_{f}\left\\{ \text{TV}\left(\hat{f},f\right)\right\\} \ge\frac{W}{n^{1/3}}\text{,}
> > $$
> > for some universal constant $W$ depending only on the large Lipschitz
> > constant. This lower bound is indeed faster than our achieved rate
> > of $O\left(n^{-1/4}\right)$, but it is achieved by restricting to
> > the smaller class of Lipschitz targets which are a subset of the continuous
> > targets which satisfy $\text{KL}_{h}\left(f\Vert\mathscr{C}\right)=0$.
> >
> > We notice that the target $f\_{2}$ from our simulations in Section
> > 6 corresponds to an element in the class of Lipschitz targets and
> > thus the rate of improvement the lower bound of $O\left(n^{-1/3}\right)$
> > of [5] should apply. Here, we can compare this rate with $\sqrt{n^{-b_{2}}}$
> > for Experiment 2 in Table 1, which yields the experimental rate in
> > $n$ of $O\left(n^{-1.03}\right)$, where the exponent is between
> > $-1.07$ and $-0.98$, with $95\%$ confidence, over the range of
> > $n\in\left\\{ 2^{10},\dots,2^{15}\right\\} $ that we assessed. Clearly
> > this is much faster than the lower bound rate of $O\left(n^{-1/3}\right)$,
> > and thus we must conclude that the rapidity of the observed rates
> > in Table 1 is a small $n$ and $k$, phenomenon, and at least the
> > rate with respect to $n$ must eventually slow down to at least $O\left(n^{-1/3}\right)$,
> > in the case when the target $f$ is Lipschitz on $\mathscr{X}$, which
> > is only slightly faster than our guaranteed rate of $O\left(n^{-1/4}\right)$.
> > Proving that $O\left(n^{-1/4}\right)$ is minimax optimal for some
> > class of targets $f$ is highly nontrivial and we leave such exploration
> > to future work.
> >
> > We shall include a version of this discussion in our revision, along with the novel Pinsker's inequality result.
> >
> > *References*
> >
> > [1] Ghosal, S. (2001). Convergence rates for density estimation
> > with Bernstein polynomials. The Annals of Statistics, 29(5), 1264-1280.
> >
> > [2] Petrone, S., \& Wasserman, L. (2002). Consistency of Bernstein
> > polynomial posteriors. Journal of the Royal Statistical Society Series
> > B: Statistical Methodology, 64(1), 79-100.
> >
> > [3] Petrone, S. (1999). Random Bernstein polynomials. Scandinavian
> > Journal of Statistics, 26(3), 373-393.
> >
> > [4] Cadez, I., \& Smyth, P. (2000). Model complexity, goodness
> > of fit and diminishing returns. Advances in Neural Information Processing
> > Systems, 13.
> >
> > [5] Devroye, L., \& Lugosi, G. (2001). Combinatorial methods in
> > density estimation. Springer Science \& Business Media.

---

### Review · Reviewer_gqi7 · 2024-08-10

**Summary Of Contributions:**

The authors proposed a special divergence so called the h-lifted KL divergence and proved the convergence rate
of the excess risk of the MLE corresponding to the h-lifted KL divergence. The advantage the authors claim is that their theoretical
results are valid for densities not lower bounded.

**Audience:**

Yes

**Broader Impact Concerns:**

I do not think any ethical concerns exist.

**Claims And Evidence:**

No

**Requested Changes:**

(1) There must be results about the convergence rate of the density estimation of $f$ rather than $f+h.$
(2) I want to hear why existing theoretical results for lower bounded density and KL divergence cannot be applied for the estimation
of $f+h$ that is lower bounded.
(3) Discussions about the choice of $h$ must be given for practical applicability of the proposed algorithm.
(4) Some analysis for the efficiency of the MM algorithm compared with existing algorithms should be done.

**Strengths And Weaknesses:**

Strength: Adding a lower bounded density function to a given model is interesting. In addition, the paper is well written.

Weakness
(1) My main concern is that all the theoretical results are derived in terms of the excess risk of the h-lifted KL diverges. It is not clear what are the relations between the h-lifted KL divergence and the original KL divergence (or the original L2 distance). To me, all of the results are about the $f+h$ instead of $f.$ Since $f+h$ is lower bounded, I think that existing results can be applied.

(2) I could not find discussions about the choice of $h$ in practice.

(3) Computation is another concern. A standard EM may not be easily applicable and so the authors developed an MM algorithm. I wonder how efficient is the proposed MM algorithm compared to the computation algorithm for the minimum L2 distance estimator.

---

> ### Author Response · Authors · 2024-08-13
> **Rebuttal by Authors**
>
> *Weakness: (1) My main concern is that all the theoretical results are derived in terms of the excess risk of the h-lifted KL diverges. It is not clear what are the relations between the h-lifted KL divergence and the original KL divergence (or the original L2 distance). To me, all of the results are about the  𝑓+ℎ  instead of  𝑓 . Since  𝑓+ℎ  is lower bounded, I think that existing results can be applied.*
>
> *Requested change: (1) There must be results about the convergence rate of the density estimation of $f$ rather than $f+h$.*
>
> We thank the Reviewer for the comment. We note that for any $f,g$
> that are probability densities with respect to the dominant measure
> $\mu$ on $\mathscr{X}$, the $h$-lifted KL divergence
> $$
> \text{KL}\_{h}\left(f||g\right)=\int\_{\mathscr{X}} \left\\{ f+h\right\\} \log\left(\frac{f+h}{g+h}\right)\text{d}\mu
> $$
> is a Bregman divergence on the space of probability densities dominated
> by $\mu$ on $\mathscr{X}$, as we prove in Appendix A. That is to
> point out that $h$ is part of the divergence function, not part of
> the function inputs, and thus all bounds and results that we produce
> are with respect to bounding the divergence between some target $f$
> and some approximator $f_{k}$ or estimator $f_{k,n}$, with the fact
> that it is a Bregman divergence implying that $\text{KL}\_{h}\left(f\Vert f\_{k,n}\right)=0$
> if and only if $f=f\_{k,n}$ $\mu$-almost everywhere, for instance.
> Thus, one should think of $h$ as being part of the definition of
> the loss and not part of the target or estimands.
>
> We note that we had already established the relationship between $\text{KL}\_{h}$
> and the $L\_{2}$ distance in our text (i.e., Proposition 3), where
> we have the result that if $h\left(x\right)\ge\gamma>0$ for every
> $x\in\mathscr{X}$, then
> $$
> \text{KL}\_{h}\left(f\Vert g\right)\le\frac{1}{\gamma}L\_{2}^{2}\left(f,g\right)\text{,}
> $$
> where
> $$
> L\_{2}^{2}\left(f,g\right)=\left\Vert f-g\right\Vert\_{2}^{2}=\int\_{\mathscr{X}}\left(f-g\right)^{2}\text{d}\mu
> $$
> is the square of the usual $L_{2}$ distance between density functions.
> Since we can always choose $h\left(x\right)\ge\gamma$, we can always
> enforce this bound. This is a stronger property result than for the
> relationship between the $\text{KL}$ divergence and the $L_{2}$
> distance, which has the same form (see [1], Lemma 3), i.e.,
> $$
> \text{KL}\left(f\Vert g\right)\le\frac{1}{\gamma}L_{2}^{2}\left(f,g\right)\text{,}
> $$
> but with the stipulation that $f\left(x\right)\ge\gamma>0$, for every
> $x\in\mathscr{X}$, which of course excludes application to any $f$
> which is not bounded away from zero, which limits the applicability
> of this relationship.
>
> In answering a question of Reviewer Wawz, we also made the important
> observation that, like the KL divergence, $\text{KL}_{h}$ is also
> bounded from below by the total variation distance:
> $$
> \text{TV}\left(f,g\right)=\left(1/2\right)\int\_{\mathscr{X}}\left|f-g\right|\text{d}\mu
> $$
> via the relationship
> $$
> \sqrt{\text{KL}\_{h}\left(f\Vert g\right)}\ge\text{TV}\left(f,g\right)\text{,}
> $$
> which can be viewed as a version of Pinsker's inequality. But
> the total variation distance, via Le Cam's inequality, also bounds
> the Hellinger divergence
> $$
> \text{H}\left(f,g\right)=\left\\{ \int\left(f^{1/2}-g^{1/2}\right)\text{d}\mu\right\\} ^{1/2}\text{,}
> $$
> that is
> $$
> \text{TV}\left(f,g\right)\ge\frac{1}{2}\text{H}^{2}\left(f,g\right)\text{.}
> $$
>
> Thus, our bound implies not only bounds with respect to the $\text{KL}\_{h}$
> divergence, which yields our sharpest result, but also bounds with
> respect to the total variation distance and the Hellinger divergence
> between targets $f$ and the $h$-MLLE $f\_{k,n}$. That is, we have
> the following chain of inequalities form the bound from Corollary
> 10 (and similarly for Theorem 9):
> $$
> \frac{1}{4}\mathbf{E}\left\\{ \text{H}^{4}\left(f,f\_{k,n}\right)\right\\} \le\mathbf{E}\left\\{\text{TV}^{2}\left(f,f\_{k,n}\right)\right\\}\le\mathbf{E}\left\\{\text{KL}\_{h}\left(f\Vert f\_{k,n}\right)\right\\}\le\text{KL}\_{h}\left(f\Vert\mathscr{C}\right)+\frac{c\_{1}}{k+2}+\frac{c\_{2}}{\sqrt{n}}\text{,}
> $$
> where the upper bound can be further bounded above by
> $$
> \cdots\le\frac{1}{a}L_{2}^{2}\left(f,\mathscr{C}\right)+\frac{c_{1}}{k+2}+\frac{c_{2}}{\sqrt{n}}\text{,}
> $$
> where $h\left(x\right)\ge a>0$, for each $x\in\mathscr{X}$. We believe
> that these relationships provide a comprehensive view as to the bounding
> and rates of convergence of $f_{k,n}$ to $f$, and never with respect
> to $f+h$, since $h$ is part of the divergence. We also note that
> except for the constants, the bounds are satisfied for any $h$, such
> that $h\left(x\right)\ge a>0$, for each $x\in\mathscr{X}$.

---

> > ### Author Response · Authors · 2024-08-13
> > **Rebuttal by Authors**
> >
> > Unfortunately, we do not have a bound for $\text{KL}\_{h}$ being less
> > than $\text{KL}$, since we can observe that
> > $$
> > \text{KL}\_{h}\left(f,g\right)-\text{KL}\left(f,g\right)
> > $$
> > equals:
> > $$
> > \int\_{\mathscr{X}}\left(f+h\right)\log\left(f+h\right)-f\log f\text{d}\mu-\int\_{\mathscr{X}}\left(f+h\right)\log\left(g+h\right)-f\log g\text{d}\mu\text{,}
> > $$
> > where both the first term and the term subtracted are positive, and
> > their relative sizes are difficult to deduce. However, we at least
> > know that $\text{KL}\_{h}$ is bounded but $\text{KL}$ is not, which
> > are both demonstrated in Section 2.2, which means at least that $\text{KL}$
> > is not bounded above by $\text{KL}\_{h}$.
> >
> > *References*
> >
> > [1] Zeevi, A. J., \& Meir, R. (1997). Density estimation through
> > convex combinations of densities: approximation and estimation bounds.
> > Neural Networks, 10(1), 99-109.
> >
> > *Requested changes: (2) I want to hear why existing theoretical results for lower bounded density and KL divergence cannot be applied for the estimation of $f+h$ that is lower bounded.*
> >
> > Thank you for the question. We note that the two best results for
> > this problem are the ones that we reported in the Introduction. Namely,
> > from [1], it is obtained that when $f\left(x\right)\ge\gamma>0$
> > for all $x\in\mathscr{X}$, then
> > $$
> > \mathbf{E}\left\\{ \text{KL}\left(f\Vert\hat{f}\_{k,n}\right)\right\\} -\text{KL}\left(f\Vert\mathscr{C}\right)\le\frac{c\_{1}}{k}+\frac{c\_{2}}{\sqrt{n}}
> > $$
> > where $\hat{f}\_{k,n}$ is the maximum likelihood estimator, and from
> > [2],
> > $$
> > \mathbf{E}\left\\{ L\_{2}^{2}\left(f,f\_{k,n}^{\prime}\right)\right\\} -L\_{2}^{2}\left(f,\mathscr{C}\right)\le\frac{c\_{1}}{k}+\frac{c\_{2}}{\sqrt{n}}\text{,}
> > $$
> > where $f\_{k,n}^{\prime}$ is the minimum $L\_{2}$ distance estimator.
> > We note that in both cases, we observe the same rates of convergence
> > in $k$ and $n$ as what we achieved. The author asks why one cannot
> > simply insert $f+h$ into the bound of [1] and this is simply
> > because $f+h$ is not a probability density, since $\int\_{\mathscr{X}}f+h\text{d}\mu=2$.
> > We note in the proof of the Pinsker inequality in our response to
> > Reviewer Wawz that one can write
> > $$
> > \text{KL}\_{h}\left(f\Vert g\right)=2\text{KL}\left(\frac{f+h}{2},\frac{g+h}{2}\right)
> > $$
> > which yields a form that can be used with the result of [1] if
> > we take the target to be the mixture density $\left\\{ f+h\right\\} /2$,
> > and we take the approximand to be $g+h$ where $g\in\text{co}\_{k}\left(\mathscr{P}\right)$.
> > However, we notice that the maximum likelihood estimator in this case
> > takes the form
> > $$
> > f\_{k,n}\in\underset{g\in\text{co}\_{k}\left(\mathscr{P}\right)}{\arg\min}-\frac{1}{n}\sum\_{i=1}^{n}\log\left(\frac{g\left(Z\_{i}\right)+h\left(Z\_{i}\right)}{2}\right)\text{,}
> > $$
> > where $Z\_{1},\dots,Z\_{n}$ are independent and identically distributed
> > samples from the distribution with density $\left\\{ f+h\right\\} /2$.
> > We can sample from such a distribution by taking $X\_{i}$ with probability
> > $1/2$ or $Y\_{i}$ with probability $1/2$, for each $i\in\left[n\right]$,
> > where $X\_{i}$ is our $i$th observation from our generative model
> > $f$ and $Y\_{i}$ is an independent sample from the lifting density
> > $h$. Then, the bounds used by [1] are largely satisfied with
> > minor modifications, although we note that the sampling process for
> > computing the estimator throws out $50\\%$ of our observed data and
> > replaces it with the simulated data $Y\_{i}$. Thus, although this
> > MLE approach together with the bound from [1], which can be seen
> > as an alternative estimator to the $h$-MLLE and our theory, yields
> > the same rate bounds, it is wasteful in its use of the observed data.
> > Thus, our $h$-MLLE approach differs to the modified MLE approach
> > in the fact that we can provide the same bound, but with an estimator
> > that makes full use of the available data.
> >
> > We note from our response to the previous question that
> > $$
> > \text{KL}\_{h}\left(f\Vert g\right)\le\frac{1}{\gamma}L\_{2}^{2}\left(f,g\right)
> > $$
> > and thus the bound from [2] can be used to obtain the inequality
> > of the form
> > $$
> > \gamma\mathbf{E}\left\\{ \text{KL}\_{h}\left(f\Vert f\_{k,n}^{\prime}\right)\right\\} -L\_{2}^{2}\left(f,\mathscr{C}\right)\le\frac{c\_{1}}{k}+\frac{c\_{2}}{\sqrt{n}}\text{,}
> > $$
> > which implies that the average risk $\text{KL}\_{h}$ risk obtained
> > by using the minimum $L\_{2}$ estimator diminishes at the same rate
> > as for the $h$-MLLE estimator. However, simply using this result
> > has two drawback. Firstly, we do not want to obtain a risk bound for
> > the minimum $L\_{2}$ estimator since, as we have discussed in Section
> > 1.1, the estimator (see Equation (2)) is highly computationally complex
> > due to the presence of an intractable integral of the functional whose
> > parameters are the subject of the risk minimization. Furthermore,
> > the bias term $L\_{2}^{2}\left(f,\mathscr{C}\right)$ cannot be made
> > to match the expected risk being controlled since we do not know of
> > an upper bound for the $L\_{2}$ distance in terms of the KL or the
> > $\text{KL}\_{h}$ divergence.

---

> > > ### Author Response · Authors · 2024-08-13
> > > **Rebuttal by Authors**
> > >
> > > References
> > >
> > > [1] Alexander Rakhlin, Dmitry Panchenko, and Sayan Mukherjee.
> > > Risk bounds for mixture density estimation. ESAIM: PS, 9:220--229,
> > > 2005.
> > >
> > > [2] Jussi S Klemela. Density estimation with stagewise optimization
> > > of the empirical risk. Machine Learning, 67:169--195, 2007.
> > >
> > > *Weaknesses: (2) I could not find discussions about the choice of $h$ in practice.*
> > >
> > > *Requested Changes: (3) Discussions about the choice of $h$ must be given for practical applicability of the proposed algorithm.*
> > >
> > > We thank the Reviewer for this excellent suggestion. Indeed we did
> > > not provide any advice regarding the choice of $h$ as our bound is
> > > valid for any $h$ satisfying the bound: $0< a\le h\left(x\right)\le b<\infty$,
> > > for every $x\in\mathscr{X}$. Our initial thought was that $h$ did
> > > not influence the rate of convergence in either $k$ or $n$ and thus
> > > we were not concerned with its calibration. However, upon reflection
> > > on the Reviewer's question and the similar question of Reviewer u7g9,
> > > we noticed that in fact although one cannot improve the rate by calibrating
> > > $h$, it is possible to improve the constants of the upper bound of
> > > the final equation on Page 14 (immediately after Equation (14)), which
> > > states for each $t> 0$, with probability at least $1-\text{e}^{-t}$,
> > > $$
> > > \text{KL}\_{h}\left(f\Vert f\_{k,n}\right)-\text{KL}\_{h}\left(f\Vert\mathscr{C}\right)\le\frac{w\_{1}}{\sqrt{n}}\int\_{0}^{c}\log^{1/2}N\left(\mathscr{P},\epsilon/2,\left\Vert \cdot\right\Vert \_{\infty}\right)\text{d}\epsilon+\frac{w\_{2}}{\sqrt{n}}+w\_{3}\sqrt{\frac{t}{n}}+\frac{w\_{4}}{k+2}\text{,}
> > > $$
> > > which contribute to the constants from Theorem 9. Letting $c$ be
> > > the upper bound of the target $f$ (i.e., $f\left(x\right)\le c<\infty$,
> > > for every $x\in\mathscr{X}$), we make the following observations
> > > regarding the constants
> > > $$
> > > w\_{1}=\text{const.}\times\frac{c+b}{a^{2}}\text{,}
> > > $$
> > > $$
> > > w\_{2}=\text{const.}\times\frac{\left(8b+4a\right)\left(c+b\right)}{a^{2}}\text{,}
> > > $$
> > > $$
> > > w\_{3}=\text{const.}\times\log\left(\frac{c+b}{a}\right)\text{,}
> > > $$
> > > $$
> > > w\_{4}=\text{const.}\times\frac{c^{2}}{a^{2}}\text{.}
> > > $$
> > > Here, $w\_{1},w\_{2},w\_{3}$ take their form because of the final bound
> > > in the proof of Theorem 11 (immediately after Equation (13)), and
> > > $w\_{4}$ takes its form because of the bound from Equation (11).
> > >
> > > Let us consider that although $h$ can take any form, the most practical
> > > form for $h$ is to take it as the uniform distribution on $\mathscr{X}$,
> > > since sampling from $h$ for the purpose of computing the $h$-MLLE
> > > is typically easiest under this choice. Furthermore, $h\left(x\right)=z$,
> > > where $z=1/\int_{\mathscr{X}}\text{d}\mu$, and thus $a=z=b$. We
> > > notice that if $h$ is not uniform, then necessarily $a< z < b$, since
> > > there must exist some region $\mathscr{Z}$ of positive measure such
> > > that $h\left(x\right) > z$, but on this region $\int_{\mathscr{Z}}h\text{d}\mu>\mu\left(\mathscr{Z}\right)/\mu\left(\mathscr{X}\right)$,
> > > which would imply that $h\left(x\right)< z$ for some $x\in\mathscr{X}\backslash\mathscr{Z}$,
> > > else
> > > $$
> > > \int_{\mathscr{X}}h\text{d}\mu=\int_{\mathscr{Z}}h\text{d}\mu+\int_{\mathscr{X}\backslash\mathscr{Z}}h\text{d}\mu>\frac{\mu\left(\mathscr{Z}\right)}{\mu\left(\mathscr{X}\right)}+\frac{\mu\left(\mathscr{X}\backslash\mathscr{Z}\right)}{\mu\left(\mathscr{X}\right)}=1\text{,}
> > > $$
> > > hence making $h$ not a density function. Let us recall that we cannot
> > > control $c$ but can control $a$ and $b$ by choosing $h$. Now,
> > > suppose that we take $h=z$, then we notice that the numerators of
> > > $w_{1}$ would increase if $h$ deviates from $z$ and the denominator
> > > would decrease since $a< z< b$ when $h$ is not uniform, causing the
> > > constants to increase. The same argument can be made regarding the
> > > denominator of $w_{4}$, which would decrease if $h$ deviated from
> > > uniformity and thus $w_{4}$ is smallest when $h$ is uniform. For
> > > constant $w_{2}$, we can write
> > > \begin{align*}
> > > w\_{2} & =\text{const.}\times\frac{\left(8b+4a\right)\left(c+b\right)}{a^{2}}\\
> > >  & =\text{const.}\times\left\\{ \frac{8bc}{a^{2}}+\frac{4c}{a}+\frac{8b^{2}}{a^{2}}+\frac{4b}{a}\right\\} \text{.}
> > > \end{align*}
> > > But since $c>0$, the numerator in each term either increases or stays
> > > the same if $h$ deviates from uniformity, while every denominator
> > > decreases, hence $w_{2}$ is also made smallest when $h$ is taken
> > > to be uniform. Lastly, the same argument regarding $w_{1}$ holds
> > > for $w_{3}$, since the logarithm is increasing, thus $w_{3}$ is
> > > also minimized when $h$ is uniform. We are thus left with the conclusion
> > > that the smallest constants in Theorem 9 are achieved when we take
> > > $h$ to be the uniform distribution on $\mathscr{X}$. This then implies
> > > that $h$ being uniform is the optimal choice for both practical and
> > > theoretical considerations.

---

> > > > ### Author Response · Authors · 2024-08-13
> > > > **Rebuttal by Authors**
> > > >
> > > > *Weakness: (3) Computation is another concern. A standard EM may not be easily applicable and so the authors developed an MM algorithm. I wonder how efficient is the proposed MM algorithm compared to the computation algorithm for the minimum L2 distance estimator.*
> > > >
> > > > We thank the Reviewer for this comment regarding the MM algorithm.
> > > > Indeed, since the risk functional is not a log-likelihood function,
> > > > one cannot apply an EM algorithm to compute the $h$-MLLE. But as
> > > > we've discussed above, if one instead interprets $\text{KL}\_{h}$
> > > > as a loss between the target mixture $\frac{1}{2}f+\frac{1}{2}h$
> > > > to the mixture estimator $\frac{1}{2}f\_{k,n}+\frac{1}{2}h$, then
> > > > one can construct an EM algorithm using the standard admixture EM
> > > > algorithm approach (e.g., [1], Section 9.5). Remarkably the EM
> > > > algorithm obtained for computing the MLE $\frac{1}{2}f\_{k,n}+\frac{1}{2}h$
> > > > for the target $\frac{1}{2}f+\frac{1}{2}h$ has exactly the same form
> > > > as that of our MM algorithm, which uses the Jensen's inequality majorizer
> > > > (as per [1], Section 8.3). In fact, one can show that the implied
> > > > majorizer of any EM algorithm is a direct consequence of Jensen's
> > > > inequality (cf. [1], Section 9.2). Thus, our MM algorithm presented
> > > > in Section 6.1 is in fact no more complicated than an EM algorithm
> > > > for estimating mixture models.
> > > >
> > > > We cannot however compare our MM algorithm to any algorithm for computing
> > > > the minimum $L\_{2}$ estimator because there is no standard method
> > > > for computing the minimum $L\_{2}$ estimator. Further, the lack of
> > > > standard methods for computing the minimum $L\_{2}$ estimator is because
> > > > such an estimator is typically computationally intractable for all
> > > > but the simplest situations. As we have presented in the introduction,
> > > > the minimum $L\_{2}$ estimator has the form:
> > > > $$
> > > > f\_{k,n}^{\prime}\in\underset{g\in\text{co}\_{k}\left(\mathscr{P}\right)}{\arg\min}\ -\frac{2}{n}\sum\_{i=1}^{n}g\left(X\_{i}\right)+\int\_{\mathscr{X}}g^{2}\text{d}\mu\text{,}
> > > > $$
> > > > where the integral $\int_{\mathscr{X}}g^{2}\text{d}\mu$ is dependent
> > > > on the optimization variable $g$ and cannot be approximated by a
> > > > sample average Monte Carlo approximation as per our $h$-MLLE. To
> > > > note how such an integral can be highly complex, one need only consider
> > > > the situation of a Dirichlet mixture class on simplex
> > > > $$
> > > > \mathscr{X}=\left\\{ x\in\mathbb{R}^{d}:\sum_{j=1}^{d}x\_{j}=1,x\_{j}\in\left[0,1\right]\text{,}j\in\left[d\right]\right\\}
> > > > $$
> > > > defined by
> > > > $$
> > > > \mathscr{P}=\left\\{ p\left(x\right):p\left(x\right)=\frac{\prod_{j=1}^{d}\Gamma\left(\alpha_{j}\right)}{\Gamma\left(\sum\_{j=1}^{d}\alpha\_{j}\right)}\prod\_{j=1}^{d}x\_{j}^{\alpha\_{j}-1},\psi=\left(\alpha\_{1},\dots,\alpha\_{d}\right)\in\mathbb{R}\_{\ge0}^{d}\right\\} \text{,}
> > > > $$
> > > > whereupon $g\in\text{co}\_{k}\left(\mathscr{P}\right)$, and thus
> > > > $$
> > > > \int\_{\mathscr{X}}g^{2}\text{d}\mu=\int\_{\mathscr{X}}\left\\{ \sum\_{\xi=1}^{k}\pi_{\xi}\frac{\prod\_{j=1}^{d}\Gamma\left(\alpha\_{\xi,j}\right)}{\Gamma\left(\sum\_{j=1}^{d}\alpha\_{\xi,j}\right)}\prod\_{j=1}^{d}x\_{j}^{\alpha\_{\xi,j}-1}\right\\} ^{2}\text{d}x,
> > > > $$
> > > > which is a highly intractable integral with no closed form that would
> > > > require numerical integration over $d$ dimensions to approximate,
> > > > and whose approximation is required for each iteration of any algorithm
> > > > that is used to compute the minimum $L_{2}$ parameter estimator for
> > > > the parameter vector of $g$: $\psi_{k}=\left(\pi_{1},\dots,\pi_{k},\theta_{1},\dots,\theta_{k}\right)$,
> > > > where $\theta_{\xi}=\left(\alpha_{\xi,1},\dots,\alpha_{\xi,d}\right)$.
> > > > Furthermore, one neither knows that $\int_{\mathscr{X}}g^{2}\text{d}\mu$
> > > > is a convex function of $\psi_{k}$ nor the sample average expression
> > > > $-\frac{2}{n}\sum_{i=1}^{n}g\left(X_{i}\right)$, which implies that
> > > > computing the minimum $L_{2}$ parameter estimator is both a non-convex
> > > > optimization problem whose objective value whose accurate approximation
> > > > (i.e., numerical integration of $\int_{\mathscr{X}}g^{2}\text{d}\mu$)
> > > > can be highly computationally infeasible, particularly when dimension
> > > > $d$ and number of components $k$ are large.

---

> > > > > ### Author Response · Authors · 2024-08-13
> > > > > **Rebuttal by Authors**
> > > > >
> > > > > Because of the difficulty of computing $L_{2}$ objective, approximations
> > > > > to the minimum $L_{2}$ estimator are often considered, such as in
> > > > > [2] and [3], where the author considered the so-called greedy
> > > > > algorithm, based on the procedure of [4], which is generalized
> > > > > by [5], whose results we used to prove the convergence rate in
> > > > > $k$ in our main result: Theorem 9. We note that these greedy algorithms
> > > > > reduce the complexity of the stagewise optimization problem by
> > > > > considering only one mixture component at a time, i.e., reducing the
> > > > > optimization over $k$ set of parameters to that over only one set
> > > > > of parameters $\psi_{k}$ and $\pi_{k}$, for each stage $k$. However,
> > > > > this stagewise optimization does not reduce the complexity of the
> > > > > integral $\int_{\mathscr{X}}g^{2}\text{d}\mu$ which one still needs
> > > > > to compute each time an iteration of any optimization algorithm is
> > > > > run. Thus, such a stagewise algorithm does not reduce the primary
> > > > > complexity of the minimum $L_{2}$ problem. We note that in [2],
> > > > > the author only considered the mixture of $d=1$-dimensional Gaussians
> > > > > which is the simplest situation that one may consider. Furthermore,
> > > > > no details are given regarding the actual numerical optimization at
> > > > > each stage of the stagewise process, largely because this was not
> > > > > of interest and because any $L_{2}$ optimization typically simply
> > > > > combines some kind of non-convex optimization routine with a numerical
> > > > > integration routine. Other than [2] and [3], in the mixture
> > > > > model setting, we only know of two recent works [6] who consider
> > > > > the minimum $\beta$-power divergence estimator, which can be seen
> > > > > as a generalization of the minimum $L_{2}$ problem whose objective
> > > > > shares the form of a sample average expression plus an intractable
> > > > > integral, and [7] who consider an approximate $L_{2}$ minimization
> > > > > problem that uses a so-called density, like [8], instead of the
> > > > > proper mixture expression for the likelihood function.
> > > > >
> > > > > In [6], the authors also consider a stagewise estimator and provide
> > > > > no advice as to how they are solving each of the greedy stagewise
> > > > > problems, including not giving advice regarding how they are computing
> > > > > the required numerical integrals. In [7], the author considers
> > > > > the use of a genetic algorithm, which if properly set up to explore
> > > > > the parameter space, can converge eventually to the global minimum
> > > > > at great computational cost, as per any zeroth order metaheuristic
> > > > > procedure. Here, again and unfortunately, no advice is provided regarding
> > > > > the computation of the required numerical integrals.
> > > > >
> > > > > We thus do not think that there is any reasonable target for comparison
> > > > > to an algorithm for the $L_{2}$ minimization problem, as $L_{2}$
> > > > > minimization procedures for conducting finite mixture model estimation
> > > > > are typically ad hoc or not well documented.
> > > > >
> > > > > *References*
> > > > >
> > > > > [1] Lange, K. (2013). Optimization (Vol. 95). Springer Science
> > > > > \& Business Media.
> > > > >
> > > > > [2] Jussi S Klemela. Density estimation with stagewise optimization
> > > > > of the empirical risk. Machine Learning, 67:169--195, 2007.
> > > > >
> > > > > [3] Jussi S Klemela. Smoothing of multivariate data: density estimation
> > > > > and visualization. John Wiley \& Sons, 2009.
> > > > >
> > > > > [4] Barron, A. R. (1993). Universal approximation bounds for superpositions
> > > > > of a sigmoidal function. IEEE Transactions on Information theory,
> > > > > 39(3), 930-945.
> > > > >
> > > > > [5] Zhang, T. (2003). Sequential greedy approximation for certain
> > > > > convex optimization problems. IEEE Transactions on Information Theory,
> > > > > 49(3), 682-691.
> > > > >
> > > > > [6] Nishida, K., \& Naito, K. (2024). Kernel density estimation
> > > > > by stagewise algorithm with a simple dictionary. Computational Statistics,
> > > > > 39(2), 523-560.
> > > > >
> > > > > [7] Nishida, K. (2023). Kernel density estimation by genetic algorithm.
> > > > > Journal of Statistical Computation and Simulation, 93(8), 1263-1281.
> > > > >
> > > > > [8] Celeux, G., \& Govaert, G. (1992). A classification EM algorithm
> > > > > for clustering and two stochastic versions. Computational statistics
> > > > > \& Data analysis, 14(3), 315-332.
> > > > >
> > > > > *Requested Changes: (4) Some analysis for the efficiency of the MM algorithm compared with existing algorithms should be done.*
> > > > >
> > > > > Thank you for this suggestion. As above, we note that, other than
> > > > > the EM algorithm, which the MM algorithm is effectively equivalent
> > > > > to, there are no classes of algorithms that are typically implemented
> > > > > for the problem of generic estimation of a $k$ component mixture
> > > > > model in the class $\text{co}_{k}\left(\mathscr{P}\right)$ for some
> > > > > fixed class of parametric distribution $\mathscr{P}$. Thus, since
> > > > > our MM algorithm for computing the $h$-MLLE has almost identical
> > > > > iterations to the EM algorithm for computing the MLE for solving the
> > > > > same problem, we conclude that our algorithm has the same iterative
> > > > > complexity as the EM algorithm.

---

> > > > > > ### Author Response · Authors · 2024-08-13
> > > > > > **Rebuttal by Authors**
> > > > > >
> > > > > > We note that this statement is about
> > > > > > rate and not constants since obviously, per iteration, since we require
> > > > > > evaluations of objectives at both $\mathbf{X}\_{n}$ and $\mathbf{Y}\_{n}$,
> > > > > > and since we evaluate $g\left(X\_{i}\right)$ and $h\left(X\_{i}\right)$
> > > > > > at each step, our MM algorithm requires a constant multiple of evaluations
> > > > > > at each step compared to the EM algorithm. We note that this constant
> > > > > > would be $2$ or $4$ depending on whether one takes $h$ to be the
> > > > > > uniform distribution over $\mathscr{X}$ or something else. Thus,
> > > > > > in terms of number of iterations required for convergence, the MM
> > > > > > and EM algorithms should behave equivalently.
> > > > > >
> > > > > > Next, we note that the $h$-MLLE problem, like the MLE problem and
> > > > > > the $L\_{2}$ minimization problem are non-convex optimization problems,
> > > > > > thus any iterative algorithm other than global optimization algorithms
> > > > > > will not be guaranteed to converge to a global optimizer of the problem,
> > > > > > including conventional methods such as gradient based methods, e.g.,
> > > > > > gradient descent, coordinate descent, mirror descent, and variants
> > > > > > such as momentum-based methods. Nor can we expect second order methods
> > > > > > such as Newton algorithms, and quasi-Newton algorithms to locate the
> > > > > > global optimizer. In non-convex settings, the most important guarantee
> > > > > > that one can provide is that that the iterations of the algorithm
> > > > > > converges to a critical point of the objective function. In our case,
> > > > > > this guarantee is provided via the direct application of Corollary
> > > > > > 1 of [1], which we state in the conclusion of Section 6.1. We
> > > > > > note that this is the same guarantee that would be provided by the
> > > > > > EM algorithm, gradient descent, or Newton's algorithm.
> > > > > >
> > > > > > Next, one may wish to know whether one can guarantee certain rates
> > > > > > of convergence when the algorithm's iterates are in some neighborhood
> > > > > > of a critical value. The answer regarding the MM algorithm is again
> > > > > > affirmative to this question, since the $h$-MLLE objective is twice
> > > > > > continuously differentiable in the parameter $\psi\_{k}$, it admits
> > > > > > the local convergence result of [2] (Proposition 7.2.2) which
> > > > > > states that if $\psi\_{k}^{\left(s\right)}$ lands in a sufficiently
> > > > > > small neighborhood of a local minimizer $\psi\_{k}^{\*}$, then it will
> > > > > > converge at linear rate to $\psi\_{k}^{\*}$. This is the same guarantee
> > > > > > as provided by most iterative algorithms such as gradient descent
> > > > > > or line search-based quasi-Newton algorithms. We can obtain quadratic
> > > > > > rates of convergence in a neighborhood of $\psi\_{k}^{*}$ if one instead
> > > > > > uses a Newton algorithm. However, this sacrifices the monotonicity
> > > > > > (stability) property of the MM algorithm (i.e., the objective function
> > > > > > improves at each iteration), since it is known that even for convex
> > > > > > problems, the Newton algorithm may generate a divergent sequence of
> > > > > > objectives if initialized incorrectly.
> > > > > >
> > > > > > Another good reason as to why one may wish to use an MM algorithm
> > > > > > instead of a Newton algorithm is that the MM algorithm allows us to
> > > > > > decompose the original objective into a sum of objectives where the
> > > > > > components of $\psi\_{k}=\left(\pi\_{1},\dots,\pi\_{k},\theta\_{1},\dots,\theta\_{k}\right)$
> > > > > > are separable with respect to the sum. That is, we can optimize a
> > > > > > sum of functions that are each only dependent on parameters$\left(\pi\_{1},\dots,\pi\_{k}\right)$
> > > > > > or one of $\theta\_{1},\dots,\theta\_{k}$, thus separating the overall
> > > > > > difficult of the iterative task. We note this fact after Equation
> > > > > > (15) in the text. This iterative simplification can often yield computationally
> > > > > > simpler evaluations than the need to compute the Hessian for Newton's
> > > > > > method, or Hessian estimators for quasi-Newton methods. Lastly, in
> > > > > > the case of mixtures of exponential family distributions, like the
> > > > > > beta distributions studied in Section 6.2, each of the parameter separated
> > > > > > problems is in fact a strictly concave maximization problem and thus
> > > > > > can be solved very efficiently (cf. Proposition 3.10 of [3]).
> > > > > > Thus, to conclude our discussion, the MM algorithm is comparable to
> > > > > > the EM algorithm and other first-order methods such as gradient descent,
> > > > > > line search, and mirror descent algorithms, with respect to its ability
> > > > > > to find critical points and its linear convergence rate close to critical
> > > > > > points. Furthermore, it is stable, since it generates monotonic sequences
> > > > > > of objective evaluations, it generates iterations that can be viewed
> > > > > > as separable optimization sub-problems, and these sub-problems can
> > > > > > be strictly convex in many situations of interest. Thus, the MM algorithm
> > > > > > has numerous appealing characteristics as an optimization routine
> > > > > > for $h$-MLLE for mixture models.
> > > > > >
> > > > > > *References*
> > > > > >
> > > > > > [1] Meisam Razaviyayn, Mingyi Hong, and Zhi-Quan Luo. A unified
> > > > > > convergence analysis of block successive minimization methods for
> > > > > > nonsmooth optimization. SIAM Journal on Optimization, 23(2):1126--1153,
> > > > > > 2013.
> > > > > >
> > > > > > [2] Kenneth Lange. MM optimization algorithms. SIAM, 2016.
> > > > > >
> > > > > > [3] Sundberg, R. (2019). Statistical modelling by exponential
> > > > > > families (Vol. 12). Cambridge University Press.

---

> > > > > ### Comment · Reviewer_gqi7 · 2024-08-19
> > > > > **Question**
> > > > >
> > > > > I wonder why not using the EM algorithm instead of MM. As the authors wrote, $f+h$ is still a mixture. What can we more by considering MM algorithm?

---

> > ### Comment · Reviewer_gqi7 · 2024-08-19
> > **Concern**
> >
> > Thank you for the clarification. But, I still have a concern about the novelty of the proposed method.
> > I think that $f+h$ is still a mixture model (after the normalization) and the h-lifted KL divergence is nothing but the KL divergence
> > of $f+h$ and $f^*+h.$ Since $f+h$ is lower bounded, existing theories for mixture models could be applied.
> > I do not clearly see what new problems arise when we consider $f+h$ and how the authors resolve such problems.

---

> ### Author Response · Authors · 2024-09-01
> **Addressing further Question and Concern**
>
> We apologize for the late response to this question. We had mistakenly
> interpreted the 7th of September deadline as being the deadline for
> our response, rather than the 24th of August deadline that was listed.
> We hope that our subsequent statement can be taken into consideration
> when considering the publication of our manuscript.
>
> We do not disagree with the reviewer that our approach of minimizing
> the $h$-lifted KL divergence between our estimator $f_{n,k}$ (the
> $h$-MLLE) and the target $f$ is strongly related to the idea of
> computing the maximum likelihood estimator $\hat{f}_{n,k}$ in the
> class of mixtures of the form $\left\\{ g+h\right\\} /2$, where $g\in\text{co}\_{k}\left(\mathscr{P}\right)$
> is a $k$-component mixture distribution with component densities
> in $\mathscr{P}$, which minimizes the KL distance to the target $\left\\{ f+h\right\\} /2$,
> where $f$ is density on the compact set $\mathscr{X}$.
>
> Indeed we have already shown that $\text{KL}\_{h}\left(f\Vert g\right)=2\text{KL}\left(\frac{f+h}{2}\Vert\frac{f+g}{2}\right)$,
> which we use to prove that $\text{KL}\_{h}$ is a Bregman divergence
> and thus properly quantifies differences between densities $f$ and
> $g$ on $\mathscr{X}$. However, the key property that makes the analysis
> of $\text{KL}\_{h}$ fundamentally different to that of $\text{KL}$
> is the fact that when one seeks to compute an estimator for a target
> $f$, one typically observes a sample $\mathbf{X}\_{n}=\left(X\_{1},\dots,X\_{n}\right)$
> on $\mathscr{X}$ in order to compute the estimator. As we have noted,
> when one observes a sample of such a form, we can further simulate
> a sample $\mathbf{Y}\_{n}=\left(Y_{1},\dots,Y\_{n}\right)$ from the
> lifting density $h$ from which we can compute the $h$-MLLE
> $$
> f\_{n,k}=\underset{g\in\text{co}\_{k}\left(\mathscr{P}\right)}{\arg\max}\frac{1}{n}\sum\_{i=1}^{n}\log\left(g\left(X\_{i}\right)+h\left(X\_{i}\right)\right)+\frac{1}{n}\sum\_{i=1}^{n}\log\left(g\left(Y\_{i}\right)+h\left(Y\_{i}\right)\right)\text{.}
> $$
> Notice that this estimator makes full use of the sample $\mathbf{X}\_{n}$,
> with only the need of conducting a simple simulation of $\mathbf{Y}\_{n}$
> to supplement the computation.
>
> Now, if we wish to compute the MLE in the class of functions $\left\\{ g+h\right\\} /2$
> with $g\in\text{co}\_{k}\left(\mathscr{P}\right)$ targeting $\left\\{ f+h\right\\} /2$,
> we need a sample from this target. However, as we have already noted,
> our sample $\mathbf{X}\_{n}$ that we observe is naturally from $f$.
> To use $\mathbf{X}\_{n}$ to construct a sample from $\left\\{ f+h\right\\} /2$,
> we would need to conduct a simulation whereupon we simulate the sample
> $\mathbf{Y}\_{n}$, as above, but combine the sample $\mathbf{X}\_{n}$
> and $\mathbf{Y}\_{n}$ to produce a sample $\mathbf{Z}\_{n}=\left(Z\_{1},\dots,Z\_{n}\right)$,
> where $Z_{i}=X_{i}$ with probability $1/2$ and $Z_{i}=Y_{i}$ with
> probability $1/2$, independently for each $i\in\left[n\right]$.
> We can then target $\left\\{ f+h\right\\} /2$ by the MLE
> $$
> \hat{f}\_{n,k}=\underset{g\in\text{co}\_{k}\left(\mathscr{P}\right)}{\arg\max}\frac{1}{n}\sum_{i=1}^{n}\log\left(g\left(Z\_{i}\right)+h\left(Z\_{i}\right)\right)\text{,}
> $$
> which throws away, on average $1/2$ of the original data from $f$.
> This MLE can now be analyzed using the theory of [1]. And as the
> Reviewer rightly notices, the theory of [1] then provides the
> corresponding bounds for the MLE rather than the $h$-MLLE. Furthermore,
> the MLE targeting $\left\\{ f+h\right\\} /2$ can then be computed using
> the corresponding EM algorithm which would then resemble exactly that
> presented in Section 6.1, except that the $X\_{i}$s are replaced by
> $Z\_{i}$s, and the elements corresponding to $Y\_{i}$ can be removed.
> Thus, if one wants to analyze the MLE targeting $\left\\{ f+h\right\\} /2$,
> then the theory of [1] is entirely sufficient for this task, although
> this MLE does not make proper use of the full sample $\mathbf{X}_{n}$
> and thus we feel that it is less desirable than our $h$-MLLE, which
> corresponds to the adjacent theory regarding the $h$-KL divergence.

---

> > ### Comment · Reviewer_gqi7 · 2024-09-15
> > **Comment of the second rebuttal**
> >
> > I agree that the idea of using the simulated data to compute the MLE of (f+h)/2 is interesting, but this idea itself is not novel enough. It is a technical coincidence but I cannot see what new insights this interesting idea possibly implies. Hence, I think
> > that this paper should be evaluated by the derived theoretical results.

---

> > > ### Author Response · Authors · 2024-09-17
> > > **Response to Further Comment**
> > >
> > > We thank the Reviewer for the continued interest in our manuscript.
> > > We are glad that the Reviewer appreciates the use of a simulated sample
> > > as a means of computation as we see this as being a very clever method
> > > for estimating our quantity of interest. However, we must continue
> > > to disagree with the reviewer and insist that the MLE and the $h$-MLLE
> > > are not the same stochastic objects. They are just objects that have
> > > the same stochastic limit, and that can be used to estimate the same
> > > quantity, i.e.,
> > > $$
> > > \underset{f\_{k}\in\text{co}\_{k}\left(\mathscr{P}\right)}{\arg\min}\text{KL}\_{h}\left(f\Vert f\_{k}\right)=\underset{f\_{k}\in\text{co}\_{k}\left(\mathscr{P}\right)}{\arg\min}\text{KL}\left(\frac{f+h}{2}\Vert\frac{f+f\_{k}}{2}\right)\text{.}
> > > $$
> > > In fact, the $h$-MLLE estimator is the sample average approximation
> > > (SAA) estimator for the ``infeasible estimator''
> > > $$
> > > \bar{f}\_{k,n}=\underset{f\_{k}\in\text{co}\_{k}\left(\mathscr{P}\right)}{\arg\min}-\frac{1}{n}\sum\_{i=1}^{n}\log\left(f\_{k}\left(X\_{i}\right)+h\left(X\_{i}\right)\right)-\int\_{\mathscr{X}}\log\left(f\_{k}\left(y\right)+h\left(y\right)\right)h\left(y\right)\text{d}y\text{,}
> > > $$
> > > where we approximate the second term as:
> > > $$
> > > \int\_{\mathscr{X}}\log\left(f\_{k}\left(y\right)+h\left(y\right)\right)h\left(y\right)\text{d}y\approx\sum\_{i=1}^{n}\log\left(f\_{k}\left(Y\_{i}\right)+h\left(Y\_{i}\right)\right)\text{.}
> > > $$
> > > This is a fundamentally different stochastic object from the MLE which
> > > we had described in the previous response to the reviewer. As such,
> > > although following a similar process, the derivation of the bounds
> > > is different to that of the MLE, as obtained in [1]. Given that
> > > we may be at am impasse on this point, we wish to shift the matter
> > > of discussion with the Reviewer. We believe that although the observation
> > > to replace the target $f$, whose KL divergence
> > > $$
> > > \text{KL}\left(f\Vert\cdot\right)=\int\_{\mathscr{X}}f\log\frac{f}{g}\text{d}\mu
> > > $$
> > > cannot be bounded from above, via the $h$-lifted KL divergence
> > > $$
> > > \text{KL}\_{h}\left(f\Vert\cdot\right)=2\text{KL}\left(\frac{f+h}{2}\Vert\frac{\cdot+h}{2}\right)
> > > $$
> > > is a novel and useful device for constructing meaning oracle inequalities
> > > that bound the generalization loss for estimators that can be computed,
> > > i.e. via an EM (as in the case of the MLE) or MM algorithm (as in
> > > the case of the $h$-MLLE). In fact, what motivated us to write the
> > > paper was the acknowledgement that we had not come across the use
> > > of such a simple device for constructing oracle inequalities that
> > > so closely resemble already existing inequalities in the literature
> > > with such a low technical overhead. We find this technique to be elegant
> > > in its simplicity, akin to observing that if one replaces $X\ge0$
> > > in Markov's inequality with $\left|X-\text{E}X\right|$, then one
> > > obtains Chebyshev's inequality that expands the domain of application
> > > of the inequality from positive random variables with finite mean
> > > to random variables taking any real value with finite variance. Although
> > > we do not consider our work as impactful as Chebyshev's inequality,
> > > we nevertheless feel that the expansion of application via a simple
> > > observation is similar in spirit and demonstrably useful.
> > >
> > > We feel that, even if the Reviewer does not acknowledge that the $h$-MLLE
> > > and MLE are fundamentally different objects and thus demand different
> > > proofs for their oracle bounds, that at least the fact that such oracle
> > > bounds are even available using our technique is a useful contribution
> > > of interest, in particular, given its simplicity and ease of analysis.
> > > We wish to reiterate that to the best of our knowledge, our technique
> > > appears to be novel for the purpose of constructing oracle bounds
> > > and our $h$-MLLE estimator is novel as a computable estimator for
> > > either the $\text{KL}\_{h}$ or the $\text{KL}\left(\left\\{ f+h\right\\} /2\Vert\cdot/2+h/2\right)$
> > > divergences.
> > >
> > > *References*
> > >
> > > [1] Rakhlin, A., Panchenko, D., \& Mukherjee, S. (2005). Risk
> > > bounds for mixture density estimation. ESAIM: Probability and Statistics,
> > > 9, 220-229.

---

> ### Author Response · Authors · 2024-09-01
>
> We lastly acknowledge that we can understand why the Reviewer sees
> our theory as being not too different form the MLE approach, as we
> can too acknowledge that the differences are subtle, and the preference
> for approach may even be up to an opinion. However, what we feel is unfair to dispute is the fact that our observation that one can usefully
> lift the density function on a compact set $f$ by a density-valued
> factor $h$ is a novel innovation that provides the ability to obtain
> useful bounds regarding sequences of estimators $f_{n,k}$ whose distance
> to $f$ can be properly quantified by a Bregman divergence, and whose
> estimation and approximation rate to the target $f$ in the Bregman
> divergence can be bounded. Furthermore, via our novelly revealed relationship
> between the $h$-KL divergence and the total variation distance, our
> approach provides a novel, computationally feasible and implementable
> method for minimizing the total variation distance between an estimator
> and target $f$, which is known to be a very difficult task, as discussed
> at length in the work of [2, Chapter 6]. Not only that, but our
> method obtains reasonable approximation and estimation bounds in the
> total variation distance, which has a major practical implication.
>
> References
>
> [1] Rakhlin, A., Panchenko, D., \& Mukherjee, S. (2005). Risk
> bounds for mixture density estimation. ESAIM: Probability and Statistics,
> 9, 220-229.
>
> [2] Devroye, L., \& Lugosi, G. (2001). Combinatorial methods in
> density estimation. Springer Science \& Business Media.

---

> > ### Comment · Reviewer_gqi7 · 2024-09-15
> > **Question**
> >
> > Do you mean that the h-lifted KL divergence is an upper bound of the total variation distance?
> > Only thing I can find in the paper is that $L_2$ distance is an upper bound of the h-lifted KL divergence.

---

> > > ### Author Response · Authors · 2024-09-17
> > > **Response to Question**
> > >
> > > Indeed the relationship between $\text{KL}\_{h}$ and the total variation
> > > distance is a novel observation that arose from the review process.
> > > We made this observation as a response to Wawz, and we will include
> > > this fact, along with the other relationships we considered during
> > > review in the revision of the manuscript once this review process
> > > concludes. For the sake of completeness, we provide the derivation
> > > below.
> > >
> > > Let us first acknowledge the classical Pinsker's inequality [1,
> > > Lem. 2.5]:
> > > $$
> > > \text{TV}\left(f,g\right)=\frac{1}{2}\int\_{\mathscr{X}}\left|f-g\right|\text{d}\mu\le\sqrt{\frac{1}{2}\text{KL}\left(f\Vert g\right)}\text{.}
> > > $$
> > > Observe that
> > > $$
> > > \text{TV}\left(\frac{f+h}{2},\frac{g+h}{2}\right)=\frac{1}{2}\int\_{\mathscr{X}}\left|\frac{f+h}{2}-\frac{g+h}{2}\right|\text{d}\mu=\frac{1}{2}\times\frac{1}{2}\int\_{\mathscr{X}}\left|f-g\right|\text{d}\mu=\frac{1}{2}\text{TV}\left(f,g\right)\text{.}
> > > $$
> > > Pinsker's inequality then implies that
> > > $$
> > > \frac{1}{2}\text{TV}\left(f,g\right)=\text{TV}\left(\frac{f+h}{2},\frac{g+h}{2}\right)\le\sqrt{\frac{1}{2}\text{KL}\left(\frac{f+h}{2}\Vert\frac{g+h}{2}\right)}\text{.}
> > > $$
> > > As we have noted throughout, we have the relationships that
> > > $$
> > > \text{KL}\_{h}\left(f\Vert g\right)=2\text{KL}\left(\frac{f+h}{2}\Vert\frac{g+h}{2}\right)\text{,}
> > > $$
> > > thus
> > > $$
> > > \text{KL}\left(\frac{f+h}{2}\Vert\frac{g+h}{2}\right)=\frac{1}{2}\text{KL}\_{h}\left(f\Vert g\right)
> > > $$
> > > and hence
> > > $$
> > > \frac{1}{2}\text{TV}\left(f,g\right)\le\sqrt{\frac{1}{2}\times\frac{1}{2}\text{KL}\_{h}\left(f\Vert g\right)}=\frac{1}{2}\sqrt{\text{KL}\_{h}\left(f\Vert g\right)}
> > > $$
> > > yielding the desired conclusion that
> > > $$
> > > \text{TV}\left(f,g\right)\le\sqrt{\text{KL}\_{h}\left(f\Vert g\right)}\text{.}
> > > $$
> > >
> > > References
> > >
> > > [1] Tsybakov, A. B. (2009). Introduction to Nonparametric Estimation.
> > > Springer Series in Statistics.

---

> > > > ### Comment · Reviewer_gqi7 · 2024-09-17
> > > > **Still concern**
> > > >
> > > > Thank you for the detailed proof about the relation of h-lifted KL divergence and total variation.
> > > > But, I have still a concern about the message of the the theoretical results. For me, the convergence rate of the h-lifted KL divergence is not much interesting since the choice of $h$ is not easy in practice. Instead, I am interested in what  the convergence rate of the h-lifted KL divergence implies the convergence rate of $f_n$ to $f$ in a certain sense. The relation of the total variation and h-lifted KL divergence implies that the convergence rate of $f_n$ to $f$ in view of the total variation is the square root of the convergence rate of the h-lifted KL divergence.
> > > >
> > > > The convergence rate with respect to the total variation norm seems to be new (there is no literature review for this point).
> > > > I wonder why there is no study about the convergence rate of mixture model in terms of the total variation norm.
> > > > Is the derived convergence rate optimal? It would be very interesting to see whether the total variation norm convergence rate
> > > > could imply the convergence rate in $L_2$ norm since there exist convergence rates for the $L_2$ norm with densities not lower bounded.
> > > >
> > > > To sum up, my understanding is that the new contribution of this manuscript is to derive the convergence rate with respect to
> > > > the total variation norm when the density is not lower bounded. But, there is no discussion about the implications of the derived
> > > > convergence rates and no comparison with the existing results.
> > > >
> > > > Suggestion: If the authors think that the convergence rate with respect to the total variation norm is an important
> > > > contribution, I recommend the authors to focus more on the total variation norm and to use the h-lifted KL divergence as a technical tool.

---

> > > > > ### Author Response · Authors · 2024-09-17
> > > > > **Further Address of Concerns**
> > > > >
> > > > > We thank the Reviewer for the continued interest in discussing our
> > > > > paper. We are thankful for the productive discussions and suggestions.
> > > > >
> > > > > We would firstly like to point out that we had already addressed the
> > > > > choice of $h$ in our initial responses to the Reviewer. Indeed, we
> > > > > argued that although the choice of $h$ does not influence the rate
> > > > > of convergence of the approximation and estimation schemes, it does
> > > > > in fact influence the constant factors which can be made small by
> > > > > appropriately choosing $h$. Furthermore, we argued with technical
> > > > > proof that the optimal choice of $h$ is to take $h$ to be the density
> > > > > of the uniform distribution over $\mathscr{X}$. This choice provides
> > > > > the smallest constants for our obtained bounds. Thus, in fact the
> > > > > choice of $h$ is easy, it is simply to set
> > > > > $$
> > > > > h=\frac{1}{\mu\left(\mathscr{X}\right)}\text{,}
> > > > > $$
> > > > > where $h$ is a density with respect to the measure $\mu$ on $\mathscr{X}$
> > > > > that dominates the measure of $X$.
> > > > >
> > > > > Next, we agree that the convergence rate with respect to the total
> > > > > variation distance is indeed very interesting as we had implied in
> > > > > our initial discussions and explicitly described to Reviewer Wawz.
> > > > > For the sake of completeness, we explicitly describe the oracle bound,
> > > > > whose rate the Reviewer correctly inferred. That is, we have the result
> > > > > that
> > > > > $$
> > > > > \mathbf{E}\left[\text{TV}\left(f,f\_{k\_{n},n}\right)\right]\le\sqrt{\frac{w\_{1,f}}{k\_{n}}+\frac{w\_{2,f}}{\sqrt{n}}}\le\frac{w\_{1,f}^{1/2}}{\sqrt{k\_{n}}}+\frac{w\_{2,f}^{1/2}}{n^{1/4}}\le\frac{w\_{f}}{n^{1/4}}\text{,}
> > > > > $$
> > > > > where $w\_{1,f},w\_{2,f},w\_{f}>0$ are constants that depend on $f$,
> > > > > where the last inequality is obtained by setting $k\_{n}=\sqrt{n}$,
> > > > > which is the optimal choice for the number of mixture components.
> > > > >
> > > > > We note that there is no previous literature that obtains a convergence
> > > > > rate for the $\text{TV}$ distance via a sample average estimator.
> > > > > Indeed the TV distance has been explored extensively in the works
> > > > > of [1] and [2]. The topic of the two manuscripts is the TV
> > > > > distance properties of classes of density estimators. Of particular
> > > > > note is the fact that the two books focus on adaptive density estimators
> > > > > such as kernel density estimators, histogram estimators, and wavelet
> > > > > estimators. The reason why minimum risk estimators (and thus minimum
> > > > > risk mixture model estimators) are not considered is because there
> > > > > is no loss whose sample average converges to the TV distance to an
> > > > > arbitrary target density $f$. In fact, to the best of our knowledge,
> > > > > the only $\text{L}\_{p}$ norm whose minimum can be estimated via risk
> > > > > minimization is the $\text{L}\_{2}$ norm. This is because we can write
> > > > > $$
> > > > > \int\_{\mathscr{X}}\left(f-f\_{k}\right)^{2}\text{d}\mu=\int\_{\mathscr{X}}f^{2}\text{d}\mu-2\int\_{\mathscr{X}}f\_{k}f\text{d}\mu+\int\_{\mathscr{X}}f\_{k}^{2}\text{d}\mu\text{.}
> > > > > $$
> > > > > Clearly the first term is a constant and given a sample from the distribution
> > > > > whose density is the target $f$, we can approximate the distance
> > > > > by
> > > > > $$
> > > > > \int f^{2}\text{d}\mu-2\frac{1}{n}\sum\_{i=1}^{n}f\_{k}\left(X\_{i}\right)+\int\_{\mathscr{X}}f\_{k}^{2}\text{d}\mu
> > > > > $$
> > > > > and thus we have the risk minimizer estimator of the $\text{L}\_{2}$
> > > > > norm of the form
> > > > > $$
> > > > > f\_{k,n}^{\prime}\in\underset{f\_{k}\in\text{co}\_{k}\left(\mathscr{P}\right)}{\arg\min}-2\frac{1}{n}\sum\_{i=1}^{n}f\_{k}\left(X\_{i}\right)+\int\_{\mathscr{X}}f\_{k}^{2}\text{d}\mu\text{.}
> > > > > $$
> > > > > This makes the minimization of the $\text{L}\_{2}$ norm at least feasible,
> > > > > conditioned on the difficulty of computing or approximating the integral
> > > > > $\int\_{\mathscr{X}}f\_{k}^{2}\text{d}\mu$. We do not know of a similar
> > > > > method for minimizing any other $\text{L}\_{p}$ norm and in particular,
> > > > > the $\text{L}\_{1}$ norm (i.e., the TV) has no such risk minimization
> > > > > estimator.

---

> > > > > > ### Author Response · Authors · 2024-09-17
> > > > > >
> > > > > > Regarding the optimal rate, we note that our estimator obtains the
> > > > > > TV rate of $O\left(n^{-1/4}\right)$ for any continuous $f$ on compact
> > > > > > set $\mathscr{X}$. The best known rate that we can compare our result
> > > > > > to are the minimax rates for Lipschitz $f$, which is of order $O\left(n^{-1/3}\right)$
> > > > > > (cf. [1, Exercise 15.14]). Of course these rates are faster than
> > > > > > ours but require quite a strong restriction. In fact, the specific
> > > > > > result from [1] shows that for $\alpha$-Holder continuous $f$,
> > > > > > one has a lower-bounding uniform rate of $O\left(n^{-\alpha/\left(1+2\alpha\right)}\right)$
> > > > > > and thus our rate of $O\left(n^{-1/4}\right)$ is approximately as
> > > > > > can be uniformly expected of a $\left(1/2\right)$-Holder continuous
> > > > > > function and is typically faster than the worst case for any $\alpha<1/2$.
> > > > > > Indeed, under other restrictions one can obtain different estimators
> > > > > > with different rates that vary from our rate of $O\left(n^{-1/4}\right)$.
> > > > > > For instance, for concave densities the minimax rate is $O\left(n^{-2/5}\right)$
> > > > > > (cf. [1]), which is achieved by Kernel density estimators. However,
> > > > > > we again note that our target class is all bounded continuous densities
> > > > > > on $\mathscr{X}$, which is much larger than the Lipschitz or the
> > > > > > concave densities and thus the optimal rate can be expected to be
> > > > > > quite low. It is well-known that obtaining minimax rates for the class
> > > > > > of continuous densities is typically not possible and thus we do not
> > > > > > know whether our rate is optimal or not, but given that it is not
> > > > > > much slower than for concave densities (a much more restrictive class),
> > > > > > we anticipate that we are fairly close to being optimal.
> > > > > >
> > > > > > Next, we wish to address the question of whether our $\text{KL}\_{h}$
> > > > > > and $\text{TV}$ rates can be used to obtain rates in the $\text{L}\_{2}$.
> > > > > > This is possible if $f$ is supported on a finite number of points
> > > > > > on $\mathscr{X}$ (e.g. $\mu$ is a discrete measure). Then, the $\text{L}\_{1}$
> > > > > > and $\text{L}\_{2}$ norms will take the form
> > > > > > $$
> > > > > > \sum\_{j=1}^{m}\left|f\_{j}-g\_{j}\right|\text{ and }\left[\sum\_{j}^{m}\left|f\_{j}-g\_{j}\right|^{2}\right]^{1/2}
> > > > > > $$
> > > > > > where $f$ and $g$ are supported on $m$ points whose probabilities
> > > > > > are $f\_{j}$ and $g\_{j}$ for $j\in\left[m\right]$. Since the two
> > > > > > norms are just finite dimensional norms by norm equivalence, we immediately
> > > > > > obtain the $\text{L}\_{2}$ norm oracle bound of the same order (i.e.,
> > > > > > $O\left(n^{-1/4}\right)$). However, much better bounds can be obtained
> > > > > > directly via the theory of [3]. Unfortunately, when $\mathscr{X}$
> > > > > > is infinite dimensional, we no longer have norm equivalence. To the
> > > > > > best of our knowledge, there is not another method for upper bounding
> > > > > > the $\text{L}\_{2}$ norm by the $\text{L}\_{1}$ norm.
> > > > > >
> > > > > > Lastly, we agree with the Reviewer that we should include details
> > > > > > regarding the contribution towards obtaining a TV estimation rate
> > > > > > using the $h$-lifted KL divergence. We shall keep the detailed proofs
> > > > > > regarding the $\text{KL}\_{h}$ since they have some details that are
> > > > > > found nowhere else, and we will provide additional details regarding
> > > > > > the ability for the $h$-MLLE to produce a TV divergence bound. We
> > > > > > will also provide the discussion regarding our obtained rates in comparison
> > > > > > to other minimum TV divergence results. We are apprehensive in rewriting
> > > > > > our entire manuscript in the perspective of minimum TV estimation,
> > > > > > since this is a complex topic which should be treated with care, as
> > > > > > evidenced by the works of [1] and [2], and our result regarding
> > > > > > the minimum TV divergence is merely a corollary of our $\text{KL}\_{h}$
> > > > > > theory. We are still of the opinion that the $\text{KL}\_{h}$ divergence
> > > > > > is interesting technical device on its own and may serve as a good
> > > > > > starting point for other interesting derivations that we have not
> > > > > > considered, beyond the TV divergence results we have discussed.
> > > > > >
> > > > > > References
> > > > > >
> > > > > > [1] Devroye,L., Gyorfi,L.( 1985). Nonparametric density estimation:
> > > > > > the L\textsubscript{1} view.?United Kingdom: Wiley.
> > > > > >
> > > > > > [2] Devroye, L., \& Lugosi, G. (2001). Combinatorial methods in
> > > > > > density estimation. Springer Science \& Business Media.
> > > > > >
> > > > > > [3] Klemela, J. (2007). Density estimation with stagewise optimization
> > > > > > of the empirical risk. Machine Learning, 67, 169-195.

---

> > > > > > > ### Comment · Reviewer_gqi7 · 2024-09-17
> > > > > > > **Thank you**
> > > > > > >
> > > > > > > Thank you for clarification.
> > > > > > > Now, I am satisfied with the replies.
> > > > > > > But, I think the reorganization of the paper might be needed before publication.

---

> > > > > > > > ### Author Response · Authors · 2024-09-18
> > > > > > > > **Thank you**
> > > > > > > >
> > > > > > > > We thank the Reviewer for the very productive discussion and for the careful assessment of our work! Of course we agree that given the suggestions from the Reviewers, that some revision is needed to include the insights that were discussed in these reviews.

---

### Decision · Action_Editor_g7SX · 2024-11-01

**Recommendation:** Accept with minor revision

**Comment:**

When preparing the camera ready revision, please reorganize the paper as suggested by Reivewer gqi7.

**Audience:**

The paper deals with theory of statistics, which is a central foundation of machine learning. As such, some individuals in TMLR's audience will likely be interested in its finding.

**Claims And Evidence:**

Following lengthy discussion, the reviewers were satisfied with the accuracy of the claims in the paper and quality of evidence.

---

> ### Author Response · Authors · 2024-12-08
> **Revisions and Thanks**
>
> Dear Professor Carmon,
>
> Thank you for recommending our manuscript for publication.
>
> We have revised the manuscript according to your instructions and the comments from the Reviewers. Specifically, we have made the following changes:
>
> - Reorganized the Preliminary Results section into Appendix C: Auxiliary Proofs for better flow. This appendix now also contains proofs of results presented in Section 2.
>
> - For better exposition, we have moved the proofs of the main results (previously Section 5) to Appendix A: Proofs of Main Results.
> Modified Proposition 3 to include a new result from discussions regarding the lower bounding of the lifted KL divergence by the total variation distance.
>
> - Included a new remark (Remark 7) that addresses and elaborates on the scope of our results to general compact spaces, not just compact subsets of $\mathbb{R}^d$.
>
> - To address all discussion points with the reviewers, we have included Appendix B: Discussions and Remarks Regarding h-MLLEs. These discussions are paraphrased and refined versions of our conversations with the reviewers.
>
> Appendix B includes the following subsections:
>
> - B.1: Elementary Derivations - Fundamental derivations of the minimum risk estimators associated with the h-MLLE construction.
>
> - B.2: Advantages and Limitations - In-depth discussions and comparisons of the h-MLLE approach versus alternatives in the literature, including the $L^2$ distance, the maximum likelihood method, and related divergence-based loss minimizing techniques.
>
> - B.3: Selection of the Lifting Density Function h - Discussion on the problem of optimally choosing the lifting density h.
>
> - B.4: Discussions Regarding the Sharpness of the Obtained Risk Bound - Comparisons of our achieved convergence rates to minimax rates and those obtained in other works. We also discuss the implied total variation rate obtained by our estimator and provide references regarding total variation estimation and how our method can be viewed from the perspective of the ($L^1$ literature.
>
> - B.5: The KL Divergence and the MLE - Comparison of our lifted-KL divergence to an alternative construction using the KL divergence of mixtures. We compare the sample estimator constructions arising from the two approaches and their merits.
>
> - B.6: Comparison of the MM Algorithm and the EM Algorithm - Discussion of the MM algorithm proposed in our numerical experiments compared to the equivalent EM algorithm for finite mixture models under the mixture KL divergence construction.
>
> - B.7: Non-Convex Optimization - Discussion of the numerical problem of computing the h-MLLE as a non-convex optimization problem, and why the MM algorithm approach provides advantages compared to other solution methods.
>
> - We have included our author information as well as directions to the paper's github repository.
>
> - We have comprehensively proofread our manuscript, fixing typos identified by the reviewers and others we found.
>
> We are sincerely thankful for your time and effort, and the time and effort of the reviewers, to help us improve our work.
>
> Regards,
>
> The Authors